# Heterogeneous multicopy of *bla*CTX-M variants on the same plasmid enhances evolutionary adaptability in clinical *Klebsiella pneumoniae*

Rui Weng[1,2,3,11], Jingyi Zhu[1,2,3,11], Xueqing Wu[1,2,3], Qiucheng Shi [1,2,3], Yue Li[1,2,3], Junxin Zhou[1,2,3], Yanfei Wang[1,2,3], Yinping Wang[1,2,3], Weiyi Huang[1,2,3], Haiyang Liu[4], Sai Qiao[5], Ying Chen[6,7], Jinzheng Ren [8,9,10], Ping Zhang[1,2,3], Jingjing Quan[1,2,3], Dongdong Zhao[1,2,3], Xiaoting Hua [1,2,3], Xiaoxing Du[1,2,3], Jiawei Wang [8,9] ✉, Yunsong Yu [1,2,3] ✉ & Yan Jiang [1,2,3,7] ✉

Pathogenic bacteria continually evolve under antimicrobial pressure through acquired resistance genes, making it crucial to understand their evolutionary strategies. We identify a clinical *Klebsiella pneumoniae* isolate resistant to ceftazidime/avibactam (CZA), harboring heterogeneous multicopy *bla*CTX-M, among which a *bla*CTX-M-249 variant mediates CZA resistance. Both *bla*CTX-M-249 and its closely related allele *bla*CTX-M-65 are dominant within the clonal population and are located at two loci on the same plasmid, with their proportions shifting under antibiotic pressure. Using experimental and mathematical models, we demonstrate that the heterogeneous arrangement of *bla*CTX-M variants on the same plasmid confers greater stability and competitive advantage than that across separate plasmids, particularly during drug switching. Re-analysis of large genomic datasets supports the universality of this phenomenon. Our findings reveal an evolutionary strategy in which β-lactamase genes, through multicopy heterogeneity on a single plasmid, ensure stable inheritance of resistance and enhance bacterial adaptability under fluctuating clinical antibiotic pressures.

Humans have combated pathogenic bacteria by discovering naturally occurring antibiotics or developing synthetic antimicrobial agents, yet bacteria have continuously evolved new resistance mechanisms[1,2]. Meanwhile, the pace of developing new antibiotics has lagged far behind the rate at which resistance arises, becoming the biggest challenge in the treatment of clinical infections[3]. β-lactam antibiotics were once the most important class of antibiotics in clinical use, with carbapenems in particular regarded as the last resort against Gram-negative bacterial infections[4]. However, the widespread prevalence of β-lactamase- or carbapenemase-producing strains in recent decades has driven development of alternative therapeutic approaches[5],

notably β-lactam/β-lactamase inhibitor combinations such as ceftazidime/avibactam (CZA), approved by the FDA in 2015[6,7]. Nevertheless, the use of these combination drugs means that new resistance genes or mechanisms may continually be selected.

The introduction of CZA has provided an effective treatment option for infections caused by multidrug-resistant *Klebsiella pneumoniae*, particularly those caused by the widely prevalent KPC-producing strains[7]. However, the widespread use of CZA in the therapy of clinical infections has been accompanied by the emergence of various resistance mechanisms[8]. For instance, *K. pneumoniae* strains have acquired metallo-β-lactamases (MBLs), or their *bla*KPC genes have

undergone point mutations mostly within the omega loop, or increased expression via multicopy $bla_{KPC}$, all of which contribute to resistance[6,9–11]. Beyond KPC carbapenemase, mutations within the omega loop of extended-spectrum β-lactamases (ESBLs) can also confer CZA resistance, as seen in the rare CZA-non-susceptible CMY-172 and CTX-M-219, the latter closely related to the widespread CTX-M-14[12,13].

Under the selective pressure of CZA, *K. pneumoniae* can develop resistance through multiple evolutionary pathways. Therefore, understanding and predicting the evolution trajectory of bacterial resistance are crucial and significant questions. Generally, mutation is a decisive factor in bacterial evolution under selective pressure, yet this process is often hindered by "trade-offs", as newly acquired mutations, while conferring a novel gene function, may simultaneously result in the loss or weakening of the original gene activity[14]. However, gene duplications can alleviate trade-offs because they allow the coexistence of different alleles of the same gene[15,16]. In this way, extra gene copies might acquire new functions while others retain their original role[15]. Compared with single-nucleotide mutations, genetic amplifications can occur at higher rates. For example, duplications can occur at rates ranging from $10^{-4}$ to $3 \times 10^{-2}$ per cell per generation[17], whereas the typical rate for single-nucleotide mutations is $\leq 10^{-10}$ per generation[18].

A widespread phenomenon of resistance gene duplications has been observed in the genomes of clinically resistant bacteria[19]. A significant number of pathogenic bacteria, such as *K. pneumoniae*, disseminate resistance through the transfer of multicopy plasmids, although typical large conjugative plasmids usually exist at very low-copy numbers[20]. For the evolutionary process of antibiotic resistance genes under selective pressure, whether it is gene duplications or multiple copies of a resistance plasmid, they provide the potential for genetic heterogeneity, which refers to inconsistencies among the copies within a whole. This heterogeneity enables bacteria to retain the resistance profile of the pre-mutation gene while also acquiring a new resistance spectrum from the mutated gene[21–23]. This is also referred to as the "bet-hedging" strategy in evolutionary research, meaning that both functions are retained simultaneously, or in other words, it is a way of maintaining responses to multiple possible outcomes[20].

In this study, we identified a CZA-resistant *K. pneumoniae* strain from a clinical patient and found that its resistance was mediated by a novel $bla_{CTX-M}$ variant closely related to $bla_{CTX-M-65}$, a widely prevalent ESBL gene in recent years[24,25]. CTX-M-type β-lactamase genes comprise numerous variants that are classified into several groups based on sequence similarities, with the two major groups being the $bla_{CTX-M-1}$-group and the $bla_{CTX-M-9}$-group. $bla_{CTX-M-65}$ belongs to the $bla_{CTX-M-9}$-group and represents a variant derived from the ancestral member $bla_{CTX-M-14}$. Similar to CTX-M-14, CTX-M-65 exhibits strong hydrolytic activity against cefotaxime (CTX)[26]. Other members of the CTX-M-9 group, such as CTX-M-27, possess an Asp240Gly substitution, which confers enhanced resistance to ceftazidime (CAZ)[27]. Furthermore, we found that the plasmid-borne $bla_{CTX-M-65}$ and three closely related alleles in this strain appeared in a multicopy heterogeneous state. The aim of this study was to explore how variants coexist, plasmid location and multicopy heterogeneity support bacterial resistance and promote an evolutionary bet-hedging strategy under multidrug pressure.

## Results

### Clinically isolated CZA-resistant *K. pneumoniae* mediated by CTX-M variant

In a genomic sequencing-based surveillance project of resistant strains in the ICU of a tertiary hospital, we discovered two closely related bloodstream infection isolates collected from separate patients (Fig. 1, part I). One isolate from a patient with a history of CZA treatment, named ICU-3, exhibited resistance to CZA with a minimum inhibitory

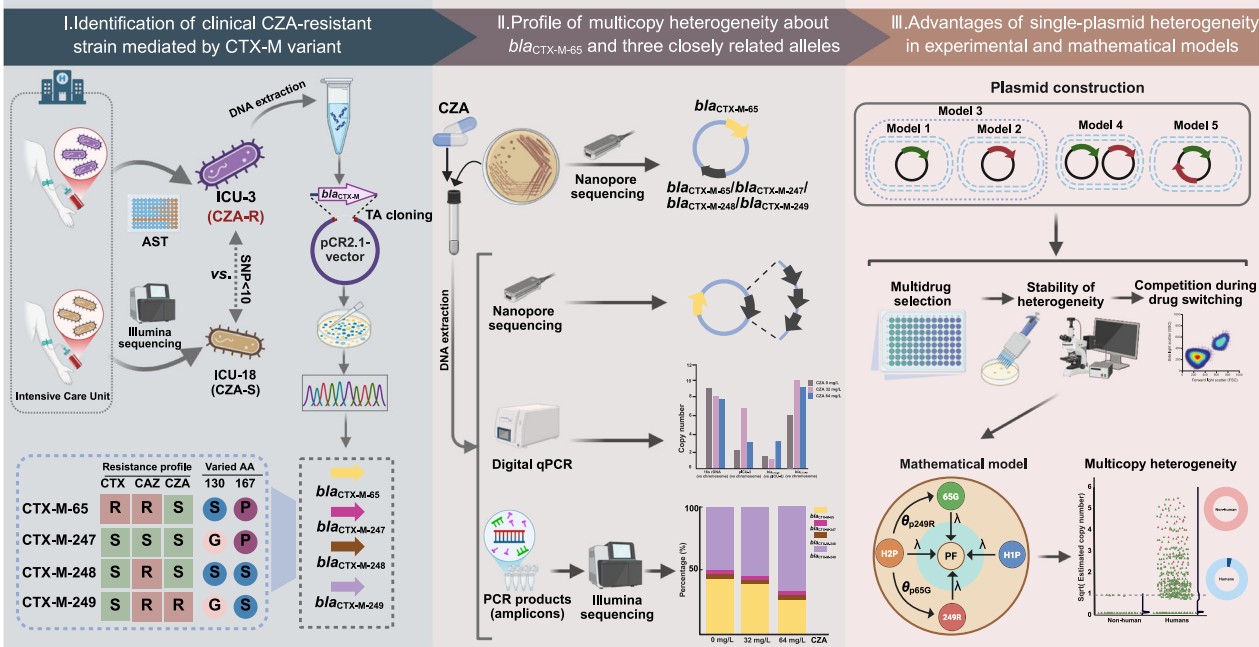

**Fig. 1 | Strain origin and workflow of this study. Part I:** Clinical identification of a CZA-resistant *K. pneumoniae* strain; genomic comparison with a susceptible strain revealed a $bla_{CTX-M}$ variant mediating resistance, and cloning confirmed distinct variants and their resistance phenotypes. **Part II:** Plasmid location and multicopy heterogeneity states of $bla_{CTX-M}$ variants were characterized, and their proportions under varying drug pressures were assessed. **Part III:** Plasmid models carrying $bla_{CTX-M}$ variants were constructed to simulate and compare the advantages of multicopy heterogeneity, particularly under multiple antibiotic combination pressures, plasmid stability under drug pressure, and competitive ability during drug switching. All findings were further validated using mathematical modeling and large-scale genomic data. AST antimicrobial susceptibility testing, CZA cefta-zidime/avibactam, CTX cefotaxime, CAZ ceftazidime, SNP single-nucleotide poly-morphisms, AA amino acid, R resistant, S in a square, susceptible; S in a circle, Serine; P proline, G glycine. Created in BioRender. Wu, X. (2026) https://BioRender.com/9m21zef.

**Table 1 | MICs of antimicrobial agents for *K. pneumoniae* clinical isolates and clones of β-lactamase gene based on *E. coli* DH5α recipient**

| Antimicrobial agent[a] | MIC (mg/L) | | | | | | | | |
|---|---|---|---|---|---|---|---|---|---|
| | *K. pneumoniae* | | *E. coli* | | | | | | ATCC 25922 |
| | | | DH5α | | | | | | |
| | ICU-3 | ICU-18 | pCR2.1-KPC-157 | pCR2.1-CTX-M-65 | pCR2.1-CTX-M-247 | pCR2.1-CTX-M-248 | pCR2.1-CTX-M-249 | pCR2.1 | |
| IMP | 64 | 64 | 16 | 0.25 | 0.25 | 0.125 | 0.125 | 0.25 | 0.25 |
| MEM | >256 | >256 | 32 | 0.03 | 0.03 | 0.03 | 0.03 | 0.03 | 0.06 |
| ETP | >256 | >256 | 32 | 0.03 | 0.06 | 0.016 | 0.016 | 0.016 | 0.016 |
| CZA | 128 | 4 | 0.5 | 0.25 | 0.25 | 8 | 64 | 0.25 | 0.25 |
| CAZ | >256 | >256 | 2 | 16 | 1 | 256 | 64 | 0.25 | 0.25 |
| CTX | >256 | >256 | 0.06 | >256 | 0.125 | 8 | 0.5 | 0.06 | 0.125 |

[a]*IMP* imipenem, *MEM* meropenem, *ETP* ertapenem, *CZA* ceftazidime/avibactam, *CAZ* ceftazidime, *CTX* cefotaxime.

concentration (MIC) of 128 mg/L, while the other isolate from a patient who had not undergone CZA treatment, named ICU-18, was susceptible (Table 1). Both isolates belong to ST11 and differ by <10 single-nucleotide polymorphisms (SNPs). The most noteworthy CZA resistance-related differences included the nucleotide substitution A392G in $bla_{KPC-2}$ (changed to $bla_{KPC-157}$ in ICU-3) and two substitutions (A397G, C508T) in $bla_{CTX-M-65}$. By testing the resistance conferred by $bla_{KPC-157}$ cloned into pCR2.1, we found that this change did not cause an increase in CZA resistance, as recently demonstrated[28] (Table 1). We then cloned the $bla_{CTX-M-65}$ mutant from ICU-3 into pCR2.1, but this resulted in colonies exhibiting heterogeneous genotypes. Among these, two predominant variants encoded the widely disseminated CTX-M-65 β-lactamase and a novel enzyme carrying Ser130Gly and Pro167Ser (based on the ABL scheme)[29], designated CTX-M-249. Two additional rare variants, carried only one of the two substitutions and were designated CTX-M-247 and CTX-M-248, respectively (Fig. 1, part I and Fig. S1a). Among clones carrying $bla_{CTX-M-65}$ or three novel alleles, only the CTX-M-249-producing clone exhibited a CZA MIC of up to 64 mg/L, indicating that this novel variant could mediate CZA resistance (Table 1). Sequencing chromatograms of directly amplified $bla_{CTX-M}$ from strain ICU-3 showed A/G and C/T double peaks at nucleic acid positions 397 and 508, confirming gene heterogeneity in this clinical strain (Fig. S1b).

### Resistance profiles and hydrolytic activity of $bla_{CTX-M-65}$ and three closely related alleles

The clones carrying $bla_{CTX-M-65}$ and three closely related alleles exhibited different MICs of CAZ, CTX and CZA, indicating that distinct combinations of mutations at amino acid positions 130 and 167 result in varied resistance profiles (Fig. 1, part I). The clone expressing CTX-M-65 β-lactamase exhibited high-level resistance to CTX (>256 mg/L) and low-level resistance to CAZ (16 mg/L), but was susceptible to CZA (0.25 mg/L). In contrast, the CTX-M-249-producing clone was resistant to both CAZ (64 mg/L) and CZA (64 mg/L), but highly susceptible to CTX (0.5 mg/L). The CTX-M-247-producing clone was susceptible to both cephalosporins (CAZ: 1 mg/L, CTX: 0.125 mg/L) and to CZA (0.25 mg/L), whereas the CTX-M-248-producing clone exhibited resistance to CAZ (256 mg/L) and CTX (8 mg/L) but remained susceptible to CZA (8 mg/L) (Table 1 and Fig. S1c).

The distinct resistance profiles of CTX-M-65 and three closely related alleles displayed a trade-off pattern, indicating that point mutations at different sites of the β-lactamase altered its hydrolytic activity, as confirmed by enzymatic kinetic assays (Table S1). Both CTX-M-248 and CTX-M-249 exhibit enhanced CAZ hydrolysis compared to CTX-M-65. CTX-M-247 and CTX-M-249 showed markedly higher $K_i$ values for avibactam and other common inhibitors, indicating reduced affinity. Thus, two point mutations in CTX-M-65 can shape interactions

with substrates or inhibitors. Specifically, the Ser130Gly in CTX-M-247 diminishes enzyme affinity for inhibitors, while the Pro167Ser in CTX-M-248 enhances enzyme affinity for CAZ, and their combination in CTX-M-249 confers CZA resistance.

### Proportion of $bla_{CTX-M-65}$ and three closely related alleles within a clonal population

In the clinical *K. pneumoniae* isolate ICU-3, long-read sequencing revealed that $bla_{KPC}$ and two copies of $bla_{CTX-M}$ are co-localized on the same plasmid, pICU-3. Notably, the two copies of $bla_{CTX-M}$ are positioned at two loci approximately 54 kb apart. One locus consistently carries $bla_{CTX-M-65}$, whereas the other locus, located near $bla_{KPC-157}$, can harbor any of four $bla_{CTX-M}$ variants, with $bla_{CTX-M-249}$ being the most frequently recovered (Fig. 2a). Plasmid pICU-18 carried by ICU-18 shares an identical backbone with pICU-3, differing only by five SNPs, including two within $bla_{CTX-M}$, one within $bla_{KPC}$, and two within a truncated IS26 element. Interestingly, under antibiotic pressure, both the copy number and the relative proportions of these variants shifted in ICU-3 bacterial cultures, as measured by digital qPCR and amplicon sequencing, respectively. Digital qPCR results showed that, compared with drug-free culture, exposure to 32 mg/L CZA increased $bla_{CTX-M}$ copy number to >10 per chromosome equivalent ($p < 0.0001$), primarily driven by a rise in plasmid copy number (from ~2 to > 7, $p = 0.0013$). At a higher 64 mg/L CZA, $bla_{CTX-M}$ copy number also approached 10 ($p = 0.0001$), but without a marked plasmid copy number increase ($p = 0.9944$); instead, the number of $bla_{CTX-M}$ copies per plasmid rose from 2 to ~3 ($p = 0.0006$, Fig. 2b). These findings suggest that amplification of resistance genes enables adaptation to drug pressure, with distinct strategy depending on pressure intensity. Next, we applied amplicon sequencing to quantify variant proportions within the clonal population. Without the CZA drug, $bla_{CTX-M-249}$ accounted for 44%, $bla_{CTX-M-65}$ for 51%, and $bla_{CTX-M-247/248}$ for ~2% each. Under CZA pressure, the proportion of $bla_{CTX-M-249}$ increased while $bla_{CTX-M-65}$ declined: at 32 mg/L, 54 vs. 41%, and at 64 mg/L, 77 vs. 19%. The frequencies of $bla_{CTX-M-247/248}$ remained ~2% regardless of drug exposure (Fig. 2c). At 64 mg/L CZA, analysis of raw long reads revealed tandem multicopy formation (1–7 copies) of the $bla_{CTX-M}$ gene near $bla_{KPC-157}$, likely explaining the increased proportions of $bla_{CTX-M-249}$ at this drug concentration. This tandem amplification is probably mediated by IS26 elements bracketing the resistance gene to form a translocatable unit (TU). In contrast, no tandem copies were detected at the other $bla_{CTX-M}$ locus, likely because its flanking IS26 are in opposite orientations (Fig. 2a). Based on above evidence, we speculate that in the vast majority of pICU-3, $bla_{CTX-M-65}$ and $bla_{CTX-M-249}$ coexist in a multicopy heterogeneous state, with their copy numbers varying in response to drug selective pressure (Fig. 1, part II).

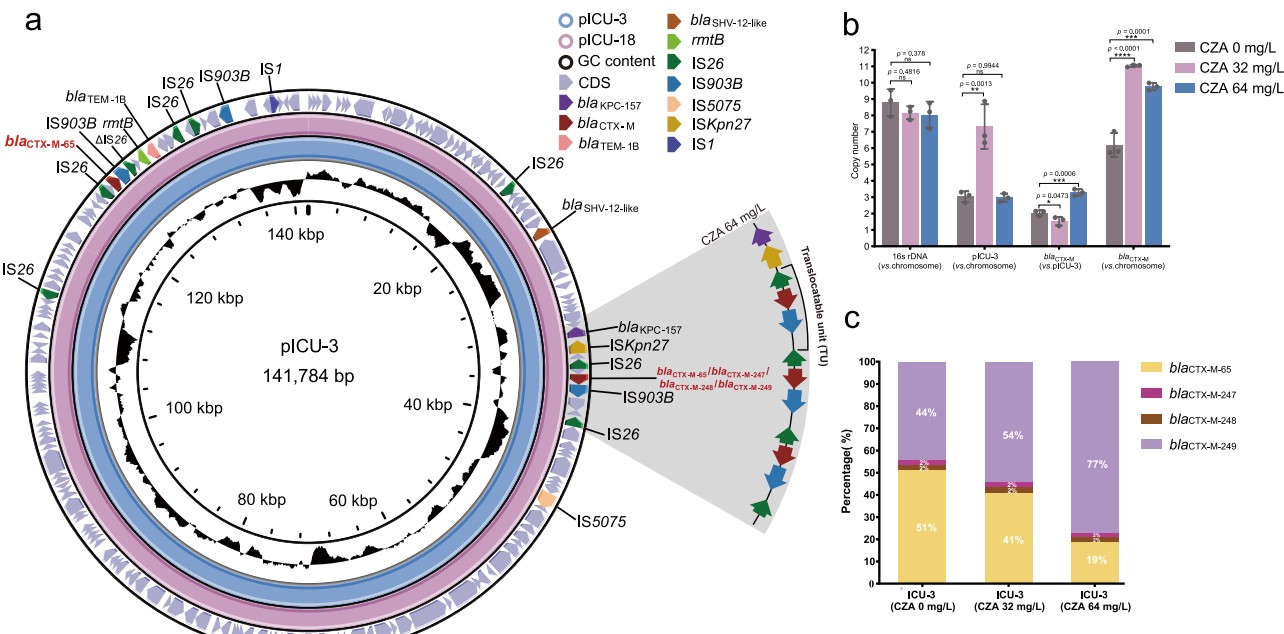

**Fig. 2 | Plasmid comparison and multicopy heterogeneity of $bla_{CTX-M}$ variants. a** Circular comparison of pICU-3 and pICU-18 with annotated resistance genes and IS; an enlarged view shows a $bla_{CTX-M}$ locus forming tandem TUs based on a representative Nanopore-based raw read. **b** Relative copy numbers of 16S rDNA, pICU-3, and $bla_{CTX-M}$ (all alleles) under different CZA concentrations (0, 32, and 64 mg/L), measured by digital qPCR. Data were mean ± SD ($n = 3$ biological independent replicates). Statistical significance was determined by two-sided one-way ANOVA with Dunnett's multiple comparisons test. **c** Proportional distribution of $bla_{CTX-M-65}$ and three closely related alleles within a single clonal population under different CZA concentrations (0, 32, and 64 mg/L), assessed by amplicon sequencing. The changes in the proportion of each allele arise from variations in both the plasmid copy number they reside in and the number of copies within tandem TUs. Data were presented as the mean of three biological independent replicates ($n = 3$). CDS coding sequence, CZA ceftazidime/avibactam. Source data are provided as a Source Data file.

## Advantages of plasmid-mediated multicopy heterogeneity

We constructed five models of plasmids carrying $bla_{CTX-M}$ variants (Fig. S2 and Table S2) to simulate and compare the advantages of multicopy heterogeneity at the population-level and single-cell level (Fig. 1, part III). Model 1 is DH5α with p65G (strain 65G, carrying $bla_{CTX-M-65}$), and Model 2 is DH5α with p249R (strain 249R, carrying $bla_{CTX-M-249}$), these two represent non-heterogeneity state. Model 3 is a 1:1 mixture of 65G and 249R (Heterogeneity in two cells, H2C), representing population-level heterogeneity. Model 4 is DH5α with both p65G and p249R, termed "Heterogeneity in two plasmids" (H2P), where heterogeneous $bla_{CTX-M}$ copies are duplicated on separate plasmids. Model 5 is DH5α with the single plasmid p65G-249R (carrying both $bla_{CTX-M-65}$ and $bla_{CTX-M-249}$), termed "Heterogeneity in one plasmid" (H1P), carrying heterogeneous $bla_{CTX-M}$ copies on the same plasmid (Fig. 3a). Then we evaluated the bacterial survival capability for these five models when facing two antibiotics (CAZ and CTX) simultaneously using a checkerboard susceptibility assay (Fig. 3b). Strain 65G grew up to high concentrations of CTX, but only at low concentrations of CAZ, while 249R grew up to high concentrations of CAZ, but not in CTX, both as expected. Models 3, 4, and 5, which express both CTX-M-65 and CTX-M-249 in different configurations, all showed growth up to relatively high levels of both CTX and CAZ. Among these, the single-cell level heterogeneity models (H1P and H2P) demonstrated stronger resistance to both drugs than the population-level heterogeneity model (H2C).

## Stability of heterogeneity in H1P and H2P models

Next, we focused on the two single-cell level heterogeneity models with dual resistance genes, H1P and H2P (Fig. 1, part III). Both models enable bacteria to acquire a bet-hedging ability, allowing them to tackle environmental stress perturbations through genetic diversification, even with the short-term fitness cost. The existence of multiple variants of the same gene that confer distinct resistance profiles

enables bacteria to withstand the combined effects of multiple drugs. However, the adaptability of bacteria in these two models still shows differences. Using fluorescence microscopy (Fig. S3a), bacteria cultured for one day under antibiotic pressure already showed distinct changes: in the H2P model, the initial orange fluorescence reverted to green in the presence of CTX (resistance conferred by CTX-M-65) and to red in the presence of CZA (resistance conferred by CTX-M-249). In the H1P model, however, orange fluorescence was consistently maintained under both CTX and CZA pressure, because CTX-M-65 and CTX-M-249 are encoded on the same plasmid. We further monitored the dynamics of the H1P and H2P models during six days of serial passage under CTX or CZA pressure using flow cytometry and colony counting. Flow cytometry revealed that, under continuous exposure to CTX or CZA, the proportion of H2P clones (dual-fluorescent cells) dropped from 80% to ~19 and ~40%, respectively. In the H2P strains, p65G was gradually lost under CZA selection (when p249R conferred resistance), whereas p249R was lost under CTX selection (when p65G was effective). By contrast, H1P strains retained >96% dual-fluorescent cells under both antibiotics, indicating stable maintenance of $bla_{CTX-M-65}$, $bla_{CTX-M-249}$, and p65G-249R (Fig. 4a and Fig. S3b). Bacterial counts further corroborated these results. After 6 days of serial passage in Lysogeny Broth (LB) with or without antibiotics, both models showed fitness costs with reduced dual-resistant populations. However, the H2P model exhibited a decline in dual-resistant bacteria relative to total cells under CTX or CZA selection, while nearly all surviving H1P cells maintained dual resistance (Fig. 4b and Fig. S3c). Collectively, these findings suggest that heterogeneity is more stable when both copies of $bla_{CTX-M}$ are on the same plasmid than distributed across separate plasmids.

## H1P gains a competitive advantage when drug pressure switches

We conducted a competition assay by co-culturing H1P and H2P strains to better understand how these two models of heterogeneity balance

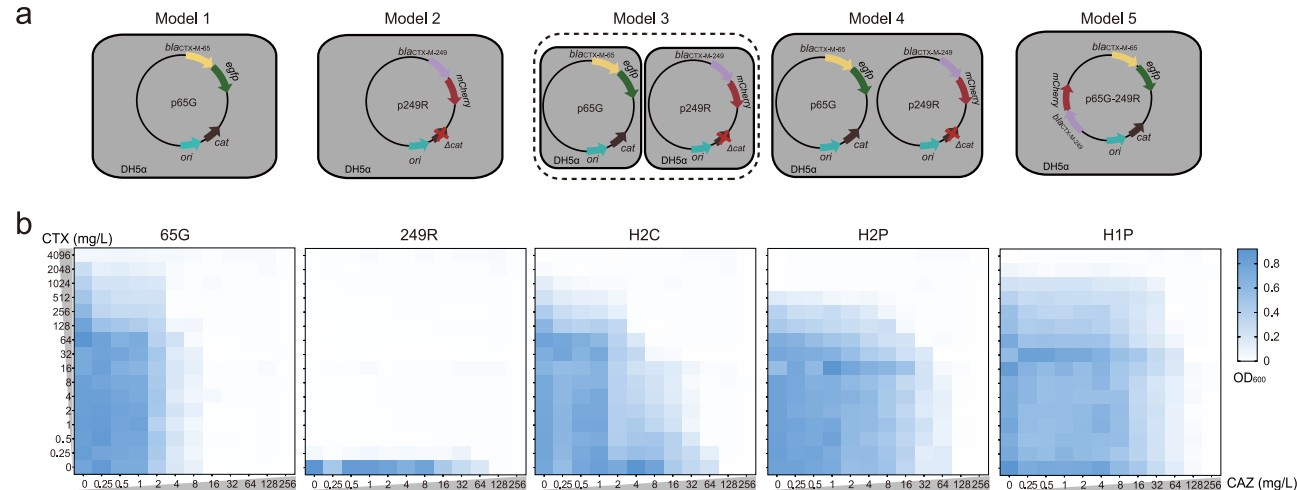

**Fig. 3 | Construction of heterogeneity models and selection under multidrug combinations. a** Model 1 (strain 65G) has p65G ($bla_{CTX-M-65}$-eGFP) only, Model 2 (strain 249R) has p249R ($bla_{CTX-M-249}$-mCherry), Model 3 is a 1:1 mixture of 65G and 249R (Heterogeneity in Two Cells, H2C). Model 4 has p65G and p249R in the same cell (Heterogeneity in Two Plasmids, H2P) and Model 5 has p65G-249R ($bla_{CTX-M-65}$-eGFP + $bla_{CTX-M-249}$-mCherry) in the cell (Heterogeneity in One Plasmid, H1P). **b** Checkerboard susceptibility assay results for the five models, evaluating the effects of co-selection by CAZ and CTX combinations. The x-axis of the checkerboard represents a gradient of increasing CAZ concentrations (from 0 to 256 mg/L), while the y-axis represents a gradient of increasing CTX concentrations (from 0 to 4096 mg/L). The gradient of blue from light to dark represents the $OD_{600}$ growth at the corresponding drug combination. Mean values of three biological independent replicates are shown ($n = 3$). CTX cefotaxime, CAZ ceftazidime. Source data are provided as a Source Data file.

their fitness under alternating antibiotics pressure (Fig. 1, part III). For differentiation in mixed population, the H1P strain was supplemented to express blue fluorescence (H1P-B), enabling identification by flow cytometry (Fig. S4a–c). Under antibiotic selection, H2P cells frequently lost one plasmid, giving rise to single-plasmid derivatives (65G or 249R), which were also distinguishable by flow cytometry. In a 5-day competition assay, H1P-B and H2P strains were cocultured at an initial 1:1 ratio under increasing CTX concentrations for four days, followed by CZA on the fifth day. Flow cytometry revealed that under CTX pressure, H1P-B remained relatively stable for the first 3 days but dropped to <1% on day 4 at high CTX concentrations due to its fitness cost. In contrast, H2P rapidly diversified into three subpopulations: itself and the plasmid-loss derivatives 65G and 249R (Fig. 5a). Both H2P and 249R quickly decreased to <1% with rising CTX pressure, while 65G expanded over 98% owing to its CTX resistance and higher fitness. However, once the antibiotic pressure switched to CZA, H1P-B increased sharply to 74%, while 65G rapidly fell to 1%, and both H2P and 249R remained low (Fig. 5b and Fig. S4d). A parallel trend was observed when CZA was applied first: H2P and its derivative 65G declined rapidly over four days, while the CZA-resistant 249R expanded to 79%. H1P-B remained initially stable but then declined. On the final day, when CTX was introduced, H1P-B rapidly dominated the population at 90%, while H2P and both derivatives dropped to 1, 7, and 1%, respectively (Fig. 5c and Fig. S4e). Overall, despite its fitness cost (Fig. S5), the H1P model stably maintains both resistance genes and rapidly gains advantage upon drug switching, whereas the H2P strain suffers from plasmid loss and remains competitively disadvantaged.

### Mathematical modeling of stability and competitive advantage
We developed an ordinary differential equation (ODE) model to simulate plasmid population dynamics, plasmid stability, and competitive advantage (Fig. 1, part III). The stability of plasmid heterogeneity in H1P and H2P subpopulations under exposure to CTX and CZA was simulated, with detailed predictions presented in Supplementary Fig. 6a, b. The simulations indicate that, in the presence of antibiotics, H1P exhibits significantly higher stability of plasmid heterogeneity compared to H2P. This finding is consistent with stability assay, as demonstrated in Fig. 4, where experimental observations align with the model's predictions, confirming its reliability in assessing differential stability under antibiotic stress. Simulations of competitive interactions between H1P and H2P under alternating antibiotic exposures are shown in Fig. 6a, b and are consistent with experimental results. Under escalating CTX concentrations followed by a switch to CZA (16 mg/L) on Day 5, H2P remained stable at low CTX levels, but with increasing concentrations and continued passaging, the 65G subpopulation (one of the H2P plasmid-loss derivatives) expanded sharply and became dominant at 256 mg/L, while both H1P and H2P declined. After the switch to CZA, 65G dropped precipitously under drug pressure, and H1P emerged as the dominant subpopulation due to its broader resistance. A parallel trend was observed with escalating CZA concentrations followed by a switch to CTX (16 mg/L). H1P dominated early, but at higher concentrations the 249R subpopulation (another H2P plasmid-loss derivative) expanded substantially. Once switched to CTX, 249R rapidly declined, and H1P regained dominance owing to its enhanced resistance.

The constructed mathematical model allowed us to estimate the antibiotic pressure thresholds at which different resistant subpopulations gain dominance. Growth density comparisons across a range of CTX (0.125–4096 mg/L) and CZA (0.0625–256 mg/L) concentrations over 24 h revealed the boundary concentrations at which H1P and H2P-derived subpopulations became dominant (Fig. 6c, d). At low CTX levels, the combined density of H2P and its derivatives (65G, 249R) exceeded that of H1P, reflecting greater adaptability. However, H1P surpassed H2P and its derivatives at 37.09 mg/L, marking its competitive advantage (Fig. 6c). A similar trend was observed under CZA exposure. H2P and its derivatives predominated at low concentrations, but H1P overtook H2P at 1.25 mg/L and maintained dominance at higher levels (Fig. 6d).

### Universality of resistance gene heterogeneity
To examine whether enhanced multidrug resistance adaptability from such heterogeneity represents a shared bacterial mechanism, we analyzed raw genomic data from additional *K. pneumoniae* isolates (derived from both humans and nonhuman sources) for evidence of multicopy heterogeneity in β-lactamase genes, such as some prevalent alleles belonging to $bla_{SHV}$, $bla_{TEM}$, $bla_{CTX-M}$, $bla_{NDM}$, $bla_{KPC}$, and $bla_{OXA}$ (Fig. 1, part III). We used the raw genomic data of *K. pneumoniae* from our own surveillance projects (1580 isolates) and downloaded the raw

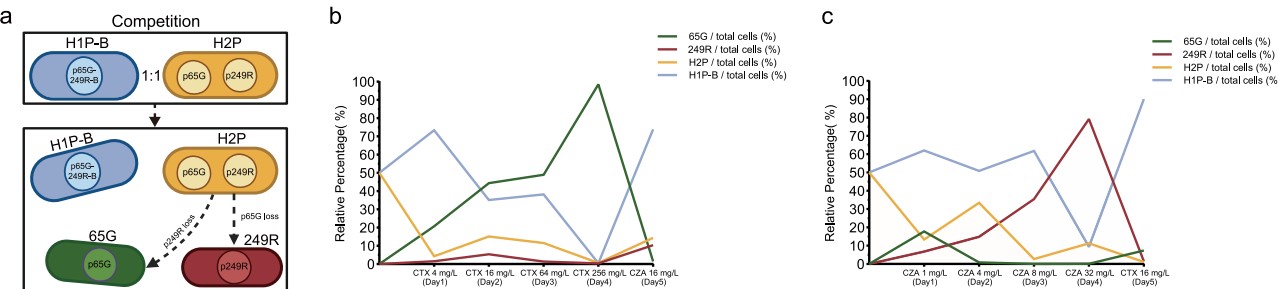

**Fig. 4 | Stability of heterogeneity in the H1P and H2P models under CTX or CZA pressure. a** H1P and H2P strains were continuously passaged under CTX or CZA pressure and analyzed by flow cytometry. The percentage of dual-fluorescent cells among total cells is plotted over passages at days 0, 1, 3, and 6. Data were presented as mean ± SD ($n$ = 3 biological independent replicates). Statistical significance was determined by two-sided repeated measures two-way ANOVA with Geisser-Greenhouse correction, followed by Tukey's post hoc test. **b** H1P and H2P strains

models were continuously passaged by plating serial dilutions on CTX or CAZ and counting CFU at days 0, 3, and 6. Data were presented as mean ± SD ($n$ = 3 biological independent replicates, each averaged from 3 technical replicates). Significance was assessed by two-tailed paired $t$-tests at each time point. CTX cefotaxime, CZA ceftazidime/avibactam, H2P heterogeneity in two plasmids, H1P heterogeneity in one Plasmid. Source data are provided as a Source Data file.

**Fig. 5 | Dynamics of subpopulations in competition assays under drug switching. a** Schematic of competition outcome. A 1:1 co-culture was initiated with H1P-B (additional blue fluorescent) and H2P. During competition under alternating drug pressure, segregational loss of one plasmid gave rise to single-plasmid derivatives: 65G (retaining p65G only) and 249R (retaining p249R only). Created in BioRender. Wu, X. (2026) https://BioRender.com/auf14hj. **b** H1P-B and H2P strains were mixed 1:1, cocultured, and serially passaged. After 4 days of increasing CTX exposure, the drug was switched to CZA on day 5. Flow cytometry tracked H1P-B,

H2P, and derivative subpopulations (65G or 249R) arising from loss of one of the two plasmids in H2P. Mean values of three biological independent replicates are shown ($n$ = 3). **c** The same setup with 4 days of increasing CZA followed by a switch to CTX on day 5. Flow cytometry quantified the proportions of H1P-B, H2P, and derivative subpopulations 65G and 249R. Mean values of three biological independent replicates are shown ($n$ = 3). CTX cefotaxime, CZA ceftazidime/avibactam, H1P heterogeneity in one plasmid, H2P heterogeneity in two plasmids. Source data are provided as a Source Data file.

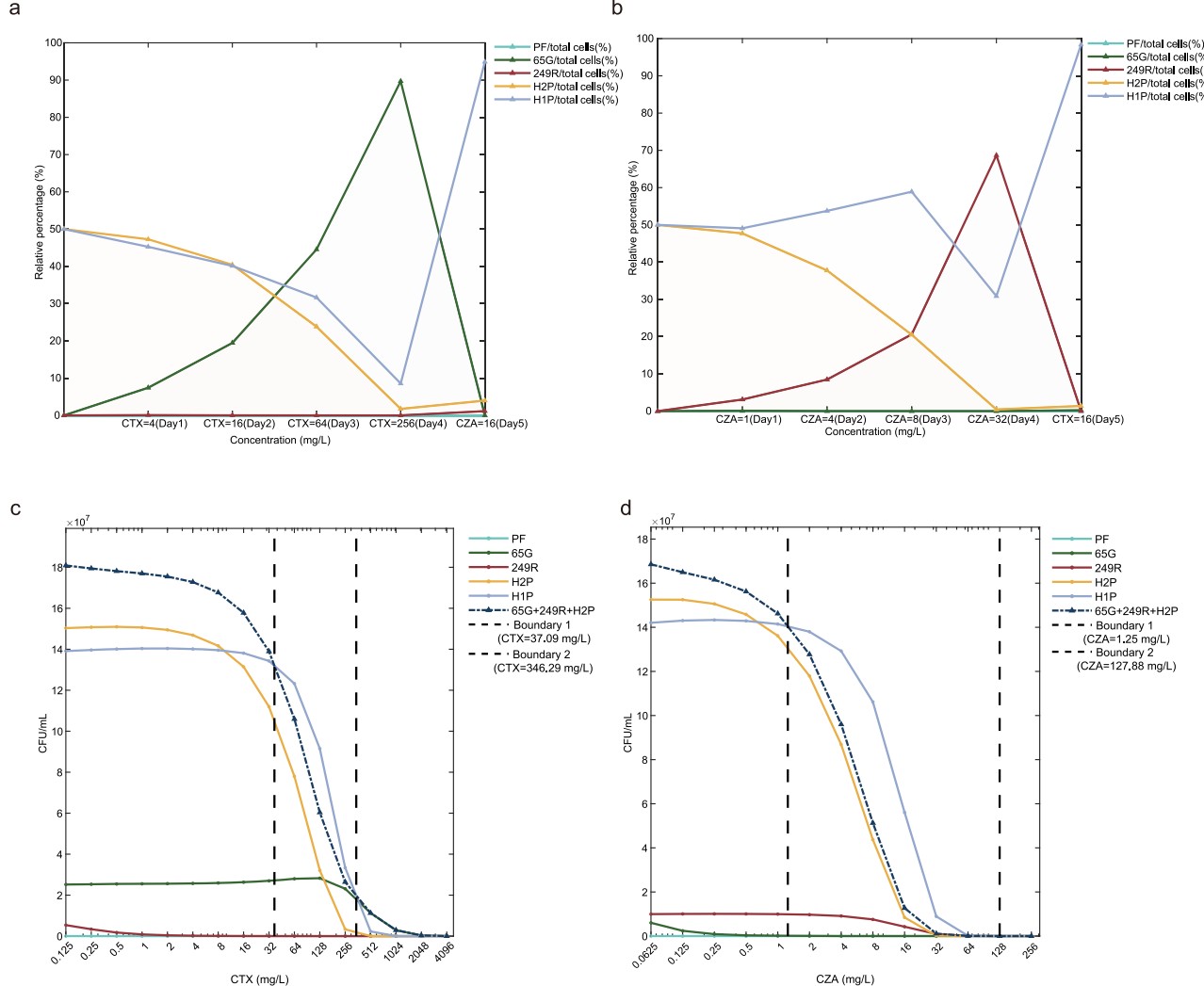

**Fig. 6 | Mathematical model simulations of competitive dynamics and dominance boundaries in H1P and H2P strains. a, b** Simulated competitive dynamics of H1P and H2P subpopulations (initial 1:1 ratio) under stepwise increases in CTX (4, 16, 64, 256 mg/L on days 1–4, then switched to CZA 16 mg/L on day 5; 20 h per cycle, 1:100 dilution, **a** or CZA (1, 4, 8, 32 mg/L on days 1–4, then switched to CTX 16 mg/L on day 5; 20 h per cycle, 1:100 dilution, **b** Subpopulation proportions are shown as filled areas with solid lines: PF (teal), 65G (green), 249R (red), H2P (orange), H1P (purple-blue). Data points indicate proportions at the end of each cycle (ODE-based simulation). **c, d** Dominance boundary analysis of H1P and H2P under gradient CTX

(0.125–4096 mg/L, **c**) or CZA (0.0625–256 mg/L, **d**) over single-cycle 24 h batch culture simulation. Final densities (CFU/mL) are plotted on semilogarithmic axes: H2P (orange circles), H1P (purple-blue circles), 65G (green circles), 249R (red circles), and PF (teal circles). Black dashed lines mark antibiotic concentrations where H1P and H2P (including its single-plasmid derivatives: 65G and 249R) densities are equal, indicating the transition point where H1P's resistance advantage outweighs H2P's growth advantage. CTX cefotaxime, CZA ceftazidime/avibactam, PF plasmid-free, H2P Heterogeneity in two plasmids, H1P Heterogeneity in one plasmid.

genomic data of this bacterium from the NCBI sequence read Archive (SRA) database (6339 isolates), and evaluated the copy number and the presence of mutation heterogeneity in these β-lactamase genes by mapping raw reads. The estimated copy number distributions of the examined genes were highly variable, with a substantial fraction of isolates carrying more than one copy. Multicopy heterogeneity was observed across all β-lactamase genes analyzed (Fig. 7). Notably, as $bla_{SHV}$ is chromosomally carried in almost all *K. pneumoniae* and may additionally occur on plasmids, the proportion of isolates showing multicopy heterogeneity of $bla_{SHV-12-like}$ reached 20.15%. The multicopy heterogeneity rates for other β-lactamase genes, including $bla_{TEM-1-like}$, $bla_{NDM-1-like}$, $bla_{CTX-M-15-like/65-like}$, $bla_{KPC-2-like}$ and $bla_{OXA-48-like}$, were 6.07, 1.20, 3.09, 0.48, and 0.30%, respectively (Table 2). Furthermore, the proportion of multicopy heterogeneity in *K. pneumoniae* derived from humans was higher than that from nonhuman sources. For example, the multicopy heterogeneity of $bla_{SHV-12-like}$ in strains from humans was 20.96%, compared to only 4.01% in strains from

nonhuman sources ($p < 0.0001$). Even no multicopy heterogeneity of $bla_{NDM-1-like}$, $bla_{CTX-M-15-like/65-like}$, $bla_{KPC-2-like}$ and $bla_{OXA-48-like}$ was detected in strains from nonhuman sources, unlike human-derived isolates, which are mostly patient-associated and may have experienced stronger antibiotic selective pressure (Fig. 7 and Table 2).

### Inferred evolutionary trajectory

We use an inferred evolutionary trajectory diagram to illustrate the relationships between several heterogeneity states during the evolutionary process, and a model diagram to describe the evolutionary advantage of resistance gene heterogeneity in the same plasmid (Fig. 8). Based on the evidence mentioned above, we hypothesize that the acquisition of a single $bla_{CTX-M-65}$ by the plasmid represents an early stage in the evolutionary trajectory in this clinical strain. Under drug selective pressure, $bla_{CTX-M-65}$ underwent gene amplification, mutations such as A397G and/or C508T, resulting in various allele forms. Interestingly, these heterogeneous alleles can coexist within a single

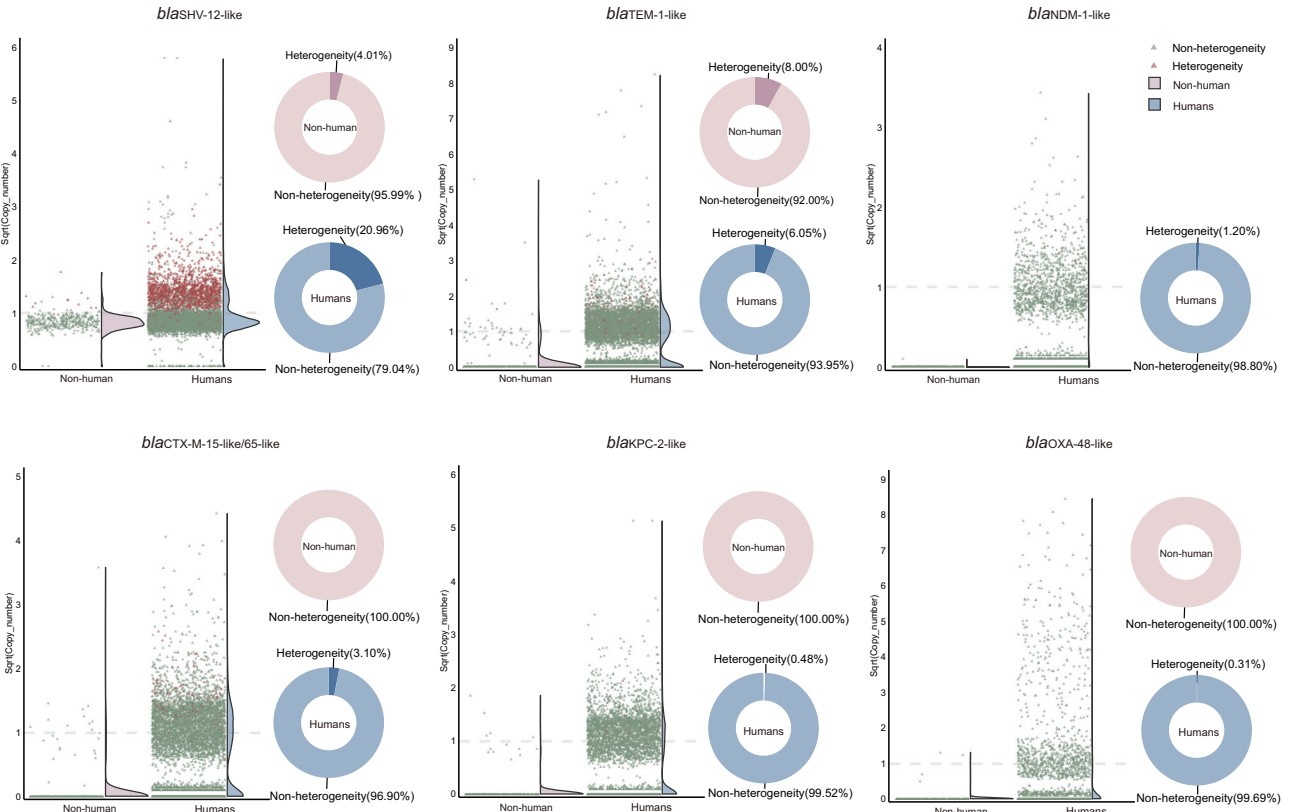

**Fig. 7 | Distribution of multicopy heterogeneity in β-lactamase genes.** Several prevalent β-lactamase genes ($bla_{SHV-12}$, $bla_{TEM-1}$, $bla_{NDM-1}$, $bla_{KPC-2}$, $bla_{CTX-M-15/65}$, and $bla_{OXA-48}$) were evaluated for copy number and mutation heterogeneity by analyzing raw read mapping coverage in *K. pneumoniae*. In the scatter and density plots, the y-axis indicates the square root of the estimated copy number. The circular charts show the proportions of strains carrying resistance genes that exhibit heterogeneous and non-heterogeneous in human versus nonhuman sources.

**Table 2 | The distribution of multicopy heterogeneity for several prevalent β-lactamase genes, including $bla_{SHV-12-like}$, $bla_{TEM-1-like}$, $bla_{NDM-1-like}$, $bla_{KPC-2-like}$, $bla_{CTX-M-15-like/65-like}$, and $bla_{OXA-48-like}$**

| Isolate Source | N | Heterogeneity | | | | | |
|---|---|---|---|---|---|---|---|
| | | $bla_{SHV-12-like}$ | $bla_{TEM-1-like}$ | $bla_{NDM-1-like}$ | $bla_{CTX-M-15-like/65-like}$ | $bla_{KPC-2-like}$ | $bla_{OXA-48-like}$ |
| Humans | 7543 | 20.96%(1565/7467) | 6.05%(264/4367) | 1.20%(14/1168) | 3.10%(130/4191) | 0.48%(15/3100) | 0.31%(3/982) |
| Nonhuman | 376 | 4.01%(15/374) | 8.00%(4/50) | 0.00%(0/0) | 0.00%(0/22) | 0.00%(0/13) | 0.00%(0/4) |
| Total | 7919 | 20.15%(1580/7841) | 6.07%(268/4417) | 1.20%(14/1168) | 3.09%(130/4213) | 0.48%(15/3113) | 0.30%(3/986) |

clonal population, with two resistance genes responding to different drug pressures being carried on the same plasmid, while also acquiring higher resistance level through transient tandem multicopy (Fig. 8a). We speculate that non-heterogeneity and heterogeneity in bacterial individuals exhibit different adaptive behaviors when facing multiple drug (particularly the β-lactams) pressures. Through comparison of the two non-heterogeneity and three heterogeneity models, we found that only single plasmid-level multicopy heterogeneity enables bacteria to maximize adaptability under both constant drug pressures and during drug switching (Fig. 8b).

## Discussion

To some extent, the evolutionary trajectory of antimicrobial resistance in pathogenic bacteria can be explained by understanding their adaptive strategies[30,31]. Bacteria continuously evolve to develop antibiotic resistance, and the mechanisms underlying resistance to CZA have become a major focus of research[6,8,11,32]. Through the analysis of clinical resistance strains, this study has uncovered a strategy in the evolution of bacteria CZA resistance. We characterized a CZA-resistant *K. pneumoniae* strain producing both KPC and CTX-M β-lactamases, in

which CZA resistance was driven by CTX-M-249 rather than in KPC-157. The CZA-resistant CTX-M-249 described here differs from the prevalent ESBL CTX-M-65 by two substitutions, Ser130Gly and Pro167Ser, each of which has been previously observed individually in other CTX-M β-lactamase alleles with defined resistance phenotypes. The Ser130Gly substitution in CTX-M-9 confers high-level resistance to β-lactamase inhibitors while reducing CTX hydrolysis[33]. The Pro167Ser substitution has been repeatedly identified in naturally occurring CTX-M enzymes, including CTX-M-19, CTX-M-62, and CTX-M-219, and significantly enhances CAZ hydrolysis with minimal impact on CTX activity[12,34,35]. As a closely related allele of $bla_{CTX-M-65}$, the phenotypic changes conferred by $bla_{CTX-M-249}$ demonstrated trade-offs[36], since the carrying strain gained CZA resistance while losing resistance to CTX. However, in this study, the bacteria evolved along a strategy to avoid resistance loss caused by trade-offs, by maintaining at least two alleles, $bla_{CTX-M-65}$ and $bla_{CTX-M-249}$, which are both carried on the same plasmid.

Then we discovered that $bla_{CTX-M-65}$ and three closely related alleles carried by this *K. pneumoniae* strain actually presented a complex state of multicopy heterogeneity. Here multicopy is first reflected

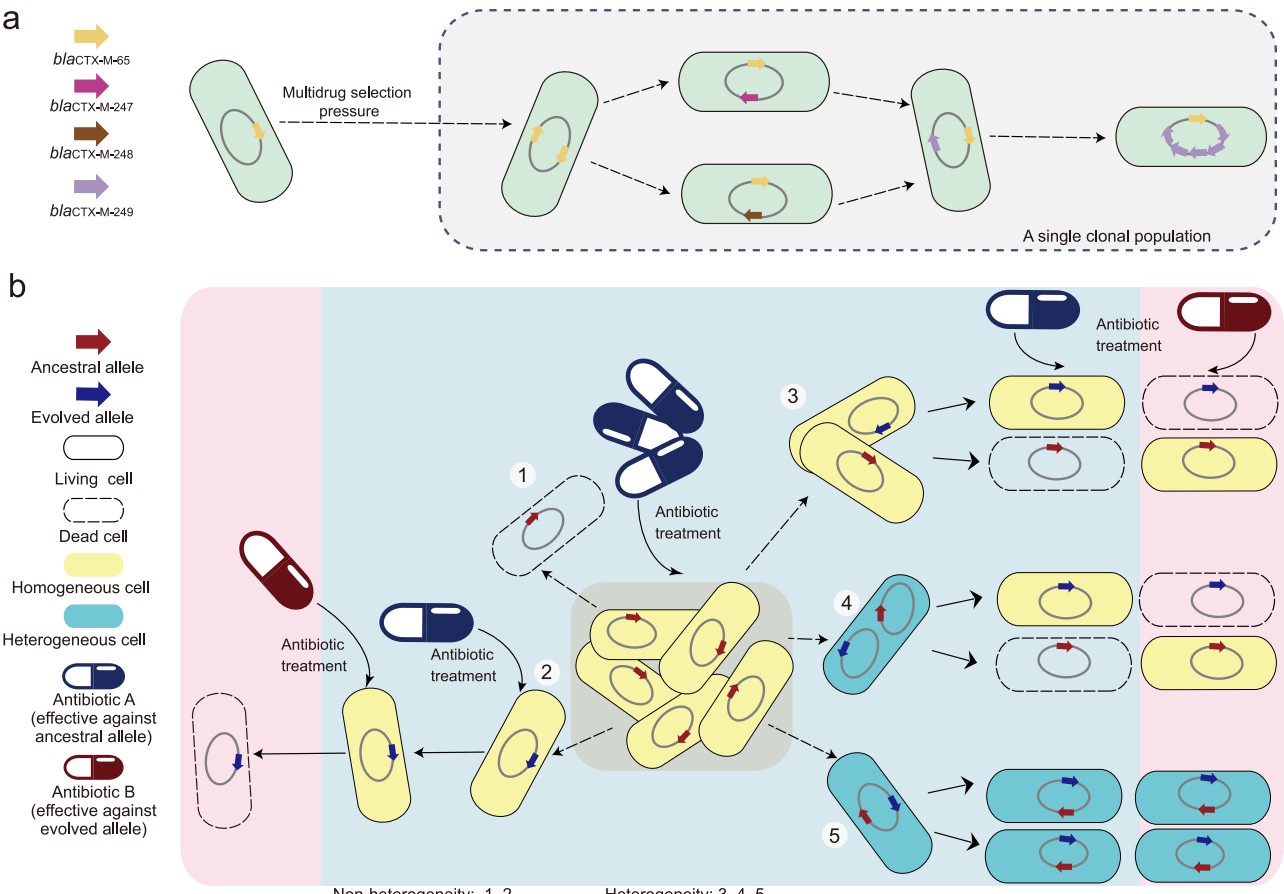

**Fig. 8 | Schematic of the proposed evolutionary trajectory and adaptive advantages of resistance gene heterogeneity. a** A hypothetical evolutionary trajectory schematic illustrating $bla_{CTX-M}$ heterogeneity in the clinical ICU-3. A bacterium carrying a single $bla_{CTX-M-65}$ represents the early stage, followed by the emergence of complex multicopy heterogeneity within a single clonal population. This includes duplication of $bla_{CTX-M-65}$, with one copy acquiring either a single mutation ($bla_{CTX-M-247/248}$) or two mutations ($bla_{CTX-M-249}$). Under antibiotic pressure, transient tandem copies of the mutated gene may be generated. **b** Schematic of five models illustrating the adaptive advantages of multicopy heterogeneity of a

β-lactamase gene on the same plasmid. The models include two non-heterogeneity and three heterogeneity scenarios, tested under hypothetical antibiotic A and B (such as the β-lactams) pressures. In non-heterogeneous models 1 and 2, bacteria are readily killed by either antibiotic. In heterogeneous models 3 and 4, at either the population or single-cell (multi-plasmid) level, plasmid loss during drug switching can still lead to elimination. Only in model 5, with plasmid-level multicopy heterogeneity, can bacteria maximize adaptability under both constant and switching drug pressures.

in the fact that $bla_{CTX-M-65}$ and other alleles are carried on multicopy plasmids. In addition, each plasmid carries two copies of the $bla_{CTX-M}$ locus. Furthermore, we found that under drug pressure, the plasmid can undergo an increase in copy number or transient tandem multicopy events mediated by IS*26* flanking $bla_{CTX-M-249}$[37,38], which could revert once the drug pressure is removed[39]. This complex multicopy state provides a basis for heterogeneity[20]. The heterogeneity refers to the presence of four $bla_{CTX-M}$ variants within a single clone: the predominant $bla_{CTX-M-65}$ and $bla_{CTX-M-249}$, and the less frequent $bla_{CTX-M-247}$ and $bla_{CTX-M-248}$. CTX-M-65 is one of the most widely reported ESBLs in clinical *K. pneumoniae*[40], while CTX-M-249 carries two amino acid substitutions compared to CTX-M-65. CTX-M-247 and CTX-M-248 each have one of these substitutions, suggesting they are intermediates in the evolution from the prevalent $bla_{CTX-M-65}$ to $bla_{CTX-M-249}$. While CTX-M-247-producing strain shows no resistance enhancement to CAZ, CTX, and CZA, and CTX-M-248 only increases CAZ hydrolysis, CTX-M-249-producing strain, which harbors both substitutions, retains CAZ resistance and confers higher resistance to CZA. This exemplifying "positive epistasis" under CZA pressure in the evolutionary process, where combined mutations yield stronger effect than single ones[31,41,42].

The most significant finding of this study is that the clinical strain carried multiple closely related alleles of a resistance gene,

representing paralogous genes that execute a bet-hedging evolutionary strategy. Paralogs are homologous genes within the same genome that arise by gene duplication and often evolve new or specialized functions. Such diversification of resistance profiles allows part of the population to better withstand environmental perturbations, albeit sometimes at the cost of reduced short-term growth[20,22]. In 2018, a study artificially constructed bacteria simultaneously carrying $bla_{TEM-1}$ and $bla_{TEM-12}$, which are closely related alleles but exhibit different resistance profiles. The authors first proposed that such heterogeneity could avoid trade-offs, conferring a bet-hedging effect, which they experimentally confirmed[14]. However, our study observed the bet-hedging effect due to the two paralogous genes, $bla_{CTX-M-65}$ and $bla_{CTX-M-249}$, in a clinical strain, and through in vitro model construction, confirmed the adaptability advantage of heterogeneity under the combined action of two drugs. Furthermore, we found that these two paralogous genes are stably co-localized on the same plasmid, which has significant evolutionary implications because it stabilizes the bet-hedging effect, enhancing the stability and competitiveness of the resistance genes.

Through the comparison of our constructed "Heterogeneity in two plasmids (H2P)", we found that the "Heterogeneity in one plasmid (H1P)" strain shows reduced fitness without drug pressure. Nevertheless, it persists in the population without losing resistance markers

and rapidly gains an advantage when selective pressure shifts between antibiotics. In contrast, the H2P strain is prone to plasmid loss, as cells carrying only one plasmid have relatively higher fitness. During drug switches, these partial-loss derivatives (strain 65G or 249R) are quickly eliminated, placing the H2P strain at a disadvantage. Thus, whether under continuous or changing antibiotic pressures, resistance genes on two plasmids are unstable, whereas co-localization on the same plasmid stably preserves heterogeneity.

In addition to constructing experimental models, we also developed a mathematical model based on a standard Monod term to simulate population growth and competitive dynamics. This model allows us to estimate the continuous changes of different bacterial subpopulations under the combined influence of nutrient limitation, antibiotic pressure, resistance levels, and plasmid burden. In this way, we can identify which resistant subpopulations gain an advantage under specific antibiotic pressures, as their growth is determined by both the benefits of resistance and the costs of fitness. In particular, for competition among multiple subpopulations, the model predicts which subgroup will prevail under given conditions and at specific time points. Moreover, we could calculate the antibiotic pressure thresholds at which different resistant subpopulations gain dominance. Clinically, this provides a means to anticipate whether a resistant clone is likely to become dominant at certain antibiotic concentrations during therapy, thereby enabling earlier and more rational infection control interventions.

The capacity of bacteria to exploit multicopy heterogeneity for multidrug resistance poses a major challenge to antibiotic selection in clinical treatment. However, our results suggest that switching between drug combinations may be an effective counterstrategy. In an additional checkerboard assay, we found that CTX combined with CZA effectively eliminated the bacteria (Fig. S7). With avibactam present, as little as 0.25 mg/L CTX inhibited the growth of strains carrying dual $bla_{CTX-M-65/249}$ copies. This likely reflects avibactam's inhibition of CTX-M-65, while CTX-M-249 exhibits only weak hydrolysis of CTX. MIC testing confirmed that, with avibactam fixed at 4 mg/L, the MIC of CTX against dual-copy heterogeneous strains was only 0.25 mg/L. These findings indicate that resistance arising from the coexistence of multiple genes under multidrug pressure may be overcome by rational drug combinations, providing valuable clinical insight and underscoring the importance of elucidating resistance mechanisms for informed therapy.

The multicopy heterogeneity of resistance genes differs from heteroresistance commonly mentioned in clinical isolates. Heteroresistance refers to a resistant subpopulation within a largely susceptible population, sometimes driven by gene amplification[23,43]. In contrast, multicopy heterogeneity refers to the coexistence of distinct allelic variants of a resistance gene within the same strain or plasmid, resulting in varied resistance profiles and potentially enhancing bacterial adaptation under diverse antibiotic pressures. Enzymes such as β-lactamases, which inactivate antibiotics, can readily undergo point mutations that produce paralogous proteins with altered activity spectra, thereby changing resistance profiles and increasing the likelihood of bet-hedging. Thus, maintaining multiple copies of different β-lactamase variants, such as $bla_{SHV}$, $bla_{TEM}$, $bla_{CTX-M}$, $bla_{NDM}$, $bla_{KPC}$, and $bla_{OXA}$, may represent a common strategy by which clinical strains evolve multidrug resistance. However, the phenomenon of multicopy heterogeneity is often overlooked in bacteria resistance studies. This is because we commonly use genomic sequence analysis to evaluate and track the spread of resistance. Both second-generation sequencing based on the Illumina platform and third-generation sequencing based on Nanopore or PacBio platforms tend to assemble the raw reads into consensus sequences[44–46]. During this assembly process, those heterogeneously paralogous genes with very similar sequences are often concealed. Through re-analysis of the raw data from a large number of *K. pneumoniae* strains, we confirmed that the proportion of multicopy

heterogeneity in clinical isolates may be significant and exceed our expectations. In contrast, the occurrence of multicopy heterogeneity in *K. pneumoniae* strains from human sources is significantly higher, suggesting that clinical drug selective pressure may be a key driver of this phenomenon.

This study also has several limitations. First, it is challenging to select a vector plasmid with a sufficiently low-copy number; therefore, the constructed model may not precisely reflect what might occur in the real world for some large conjugative plasmids, particularly those with extremely low-copy numbers. Second, we did not identify the molecular mechanism underlying the increase in plasmid copy number under CZA pressure. Although we carefully compared the complete genome sequences before and after drug exposure, no related mutations were found. In addition, assessing multicopy heterogeneity using large amounts of raw genomic reads provides only a rough estimation, and detailed characterization of multicopy heterogeneity in each strain will require further experiments.

In summary, we have identified a strategy of resistance evolution in which bacteria use multicopy formation of paralogous genes with different resistance phenotypes to cope with multidrug pressure. Plasmids and some ISs form the basis for multicopy formation, which in turn provides the conditions for heterogeneity among paralogous genes. These heterogeneous multicopy genes bring the advantage of co-selection, and are spontaneously acquired without the need for external horizontal transfer. The heterogeneity of multicopy resistance genes in a single plasmid allows the multidrug resistance state to be indefinitely prolonged, maximizing the bacteria's adaptability to tackle environmental perturbations. This strategy of resistance evolution may be widespread in clinical strains but has long been overlooked due to limitations in detection methods. The recognition of this mechanism will prompt clinicians to re-consider the selection of antimicrobial agents, avoiding the use of same class of antibiotics to prevent bacteria from developing this form of resistance.

## Methods

### Strains and antimicrobial susceptibility testing

Clinical isolates were collected from intensive care unit patients of a tertiary hospital in China (Fig. 1, part I). The study was approved by the Institutional Review Board of the Sir Run Run Shaw Hospital, Zhejiang University School of Medicine (No. 20201217-33), and written informed consent was obtained from all enrolled patients. The species identification of the bacterial strains was performed by Matrix-assisted laser desorption/ionization time-of-flight mass spectrometry (MALDI-TOF MS, bioMérieux, Craponne, France). MICs were determined using the broth microdilution method, the results were interpreted according to the Clinical and Laboratory Standards Institute (CLSI) Guidelines (CLSI, 2022). *Escherichia coli* DH5α (Takara, Tokyo, Japan) and BL21 (DE3) (Weidibio, Shanghai, China) competent cells were used in this study. *E. coli* ATCC 25922 was used as reference strain for quality control.

### Whole-genome sequencing and analysis

Genomic DNA was extracted using a Qiagen minikit (Qiagen, Hilden, Germany). Whole-genome sequencing was performed using both the NovaSeq (Reagent Kit, v1.5, Illumina, San Diego, USA) and the MinION (Flow Cell, R9.4.1, Nanopore, Oxford, UK) platforms (Fig. 1, part I and II). The complete genome was finished by first assembling Illumina short reads, followed by polishing and scaffolding with Nanopore long reads using Unicycler v0.4.8 (https://github.com/rrwick/Unicycler)[47]. Multilocus sequence typing (MLST) was performed by MLST v2.23.0 (https://github.com/tseemann/mlst)[48]. Antimicrobial resistance genes were identified using ABRicate v1.0.1 (https://github.com/tseemann/abricate) with the ResFinder database[49] and insertion sequences were identified using ISFinder[50]. The number of SNPs between *K.*

*pneumoniae* isolates ICU-3 and ICU-18 was identified and assessed with Snippy v4.4.5 (https://github.com/tseemann/snippy) and snp-dists v0.6.3 (https://github.com/tseemann/snp-dists). The multiple alignment of the CTX-M amino acid sequence was performed using ClustalW (https://www.genome.jp/tools-bin/clustalw) and ESPript 3.01[51]. The protein secondary structure was predicted using SWISS-MODEL2[52]. The genetic structures of plasmids and their comparative circular diagram were generated using Proksee (https://proksee.ca/)[53].

### Cloning the β-lactamase genes

PCR products of the complete $bla_{KPC}$ and $bla_{CTX-M}$ genes, amplified from DNA extraction from the clinical strain ICU-3, was ligated into pCR2.1 and transformed into a recipient strain *E. coli* DH5α. Each transformant carrying the recombinant plasmid (also referred to as a colony), therefore, contained only one variant of the PCR product. Colonies were selected on kanamycin-containing plates (50 mg/L), and 100 colonies were further confirmed by Sanger sequencing (Tsingke Biotech, Beijing, China). To determine the specific $bla_{CTX-M}$ variant at two distinct loci, we designed locus-specific primers to differentiate between the two sites, cloned the amplified products, and validated the different variant clones by Sanger sequencing (Fig. 1, part I). Primers (Tsingke Biotech, Beijing, China) used for amplification are listed in Supplementary Table 3.

### Protein expression and purification

To confirm the enzymatic activity of these β-lactamase, CTX-M-65, CTX-M-247, CTX-M-248, and CTX-M-249 with His-tag were expressed and purified. Briefly, the coding sequences of $bla_{CTX-M-65}$, $bla_{CTX-M-247}$, $bla_{CTX-M-248}$, and $bla_{CTX-M-249}$ lacking the N-terminal secretion peptide sequences were cloned into the pET28a(+) by homologous recombination. The resulting expression plasmids were transformed into the BL21(DE3) strain to express the proteins. Each enzyme was purified using the Ni-NTA method described previously[54]. The purified proteins were resolved by sodium dodecyl sulfate-polyacrylamide gel electrophoresis (SDS-PAGE) to quantitate and assess purity.

### Enzyme kinetics

The kinetic parameters were determined using the steady-state kinetic assay. The phosphate-buffered saline (1X, pH 7.2) was used in all enzymatic assay systems. The initial rates of substrate hydrolysis for CAZ ($\Delta\varepsilon 260 = -9,000\,M^{-1}\,cm^{-1}$) and nitrocefin ($\Delta\varepsilon 482 = 15,000\,M^{-1}\,cm^{-1}$) were measured using a D8 UV-visible spectrophotometer (Runqee, Shanghai, China) by monitoring absorbance changes in a 1-cm quartz cuvette[55]. For inhibitors and the low-affinity substrate, the $K_i$ values of enzymes were determined by a direct competitive inhibition assay using 100 μM nitrocefin as a reporter substrate[56,57]. The data were globally fitted to the Michaelis-Menten equation in Graphpad Prism v9.0.0 to determine the values of $K_m$, $k_{cat}$, $K_i$ and their corresponding standard errors.

### Amplicon sequencing of $bla_{CTX-M}$

The clinical strain ICU-3 was cultured in LB broth supplemented with different concentrations of CZA (0, 32, 64 mg/L) overnight at 37 °C and 200 rpm. Then, the genomic DNA was subsequently extracted and standardized to a final concentration of 10 ng/μL following quantification using the Qubit 4™ (Thermo Scientific, Waltham, MA, USA). PCR amplification was performed in a volume of 50 μL, using 25 μL 2×Taq PCR MasterMix (Vazyme, Nanjing, China), 1 μL genomic DNA, 1 μL (10 μM) each primer mixture, and 23 μL double-distilled water. Primers used here are listed in Supplementary Table 3. Sequencing of the PCR amplification products was performed on the NovaSeq (Illumina, San Diego, USA) sequencing platform. Raw reads were processed in CLC Genomics Workbench (QIAGEN, Hilden, Germany) by quality trimming (score limit: 0.01), trimming ambiguous nucleotides (maximum 2), and discarding reads shorter than 100 bp. Filtered reads were aligned to $bla_{CTX-M-65}$, $bla_{CTX-M-247}$, $bla_{CTX-M-248}$, and $bla_{CTX-M-249}$ reference sequences using BLAST. The percentage of each $bla_{CTX-M}$ variant in each sample was calculated as the proportion of reads aligned to each reference sequence across the different CZA concentrations (Fig. 1, part II).

### Digital qPCR

Strain processing and DNA extraction were performed as for amplicon sequencing. Genomic DNA (10 ng/μL) was then ultrasonically sheared on ice. Ultrasonic treatment conditions were set as follows: ultrasonic interval time 10 s, ultrasonic occurrence time 5 s, ultrasonic whole time 2 min, and ultrasonic power 300 W. Following sonication, digital qPCR was performed using the Sniper DQ24 digital qPCR (Sniper, Suzhou, China) system (Fig. 1, part II). Specific primers and probes (Supplementary Table 3) targeted 16S rDNA (approximately eight copies in *K. pneumoniae*)[58], *pgi* (housekeeping gene in *K. pneumoniae*), *psiB_1* (pICU-3 specific single-copy gene), and $bla_{CTX-M}$. Copy numbers based on absolute copy numbers, were calculated as: 16 s rDNA (*vs* chromosome) = 16s rDNA/*pgi*, pICU-3 (*vs* chromosome) = *psiB_1*/*pgi*, $bla_{CTX-M}$ (vs chromosome) = $bla_{CTX-M}$/*pgi*, $bla_{CTX-M}$ (*vs* pICU-3) = $bla_{CTX-M}$/*psiB_1*.

### Construction of plasmids for experimental models

We selected the relatively low-copy-number vector pACYCDuet-1 (Tsingke Biotech, Beijing, China) with p15A replicon (-13 copies), which carries a *cat* gene (conferring chloramphenicol resistance), to construct reporter plasmids harboring either single or dual $bla_{CTX-M}$ variants (Fig. 1, part III). The fluorescent protein genes were fused downstream of the $bla_{CTX-M}$ genes. The following fusion plasmids were constructed: CTX-M-65-eGFP (p65G), CTX-M-249-mCherry (p249R), and CTX-M-65-eGFP/CTX-M-249-mCherry (p65G-249R) (Fig. S2a). In order to successfully screen for the co-transformation of p65G and p249R using CZA and chloramphenicol, the *cat* gene in p249R was disrupted. The H2P strain was generated by co-transforming p65G and p249R (1:1) into *E. coli* DH5α, while the H1P strain carried p65G-249R. Transformants were selected on CZA (4 mg/L) and chloramphenicol (34 mg/L). The coexistence of $bla_{CTX-M-65}$ and $bla_{CTX-M-249}$ in H1P and H2P was confirmed by Sanger sequencing (dual peaks at positions 397: A/G, 508: C/T) and fluorescence microscopy (Olympus, Tokyo, Japan) (Fig. S2b). Images were processed with ImageJ (https://imagej.net/ij/).

### Checkerboard susceptibility assay

The antibiotic susceptibility array was prepared in 96-well flat-bottom plates using the checkerboard method, a standard assay to evaluate synergistic effects of drug combinations[59] (Fig. 1, part III). Strains representing different models (65G, 249R, H2C, H1P, and H2P) were inoculated into wells containing serial dilutions of CTX, CAZ, or their combinations to assess survival. The concentrations of CTX and CAZ were serially diluted two-fold, starting from 4096 and 256 mg/L, respectively. Two 96-well plates were used to perform the checkerboard assay, with the horizontal x-axis representing 12 dilution covering CAZ from 0 to 256 mg/L, and the vertical y-axis representing 16 dilution covering CTX from 0 to 4096 mg/L. Bacteria were inoculated at a final dilution of 1:200 and incubated for 18 h at 37 °C, cultures in each well were measured for optical density ($OD_{600}$) as a measure of growth using Thermo Scientific Multiskan™ GO (Thermo Scientific, Waltham, MA, USA).

### Heterogeneity stability assay

The stability of different heterogeneity models in constructed H1P and H2P strains was evaluated using fluorescence microscopy, flow cytometry, and bacterial counting on agar plates. (Fig. 1, part III). Fluorescence micrographs were collected using an Olympus microscope (Tokyo, Japan) and processed using ImageJ (https://imagej.net/ij/).

The H1P and H2P strains were grown in LB supplemented with either CTX (16 mg/L) or CZA (4 mg/L) in a shaking bath at 37 °C, then the overnight culture was diluted 1:100 was subcultured under the same conditions, and passaged for 6 successive days. The percentages of various populations at day 0, day 1, day 3, and day 6 were enumerated using an LSRFortessa flow cytometer (BD Biosciences, Franklin Lakes, NJ, USA). Data were collected in FITC channel (for eGFP) and PI channel (for mCherry) by 488 and 587 nm excitation. Three biological replicates were performed for each strain. In terms of plate colony counting experiment, H1P and H2P processing methods were the same as those described above, whereas cultures of day 3 and day 6 were serially diluted in PBS and plated on the Mueller-Hinton agar (MHA), and MHA added with either CTX (16 mg/L) and CZA (4 mg/L). Three technical replicates were performed for each sample. After overnight incubation at 37 °C, CFUs were measured by colony counting.

### Fitness determination

The fitness of all constructed strains was assessed by examining growth rates under drug-free and drug-pressure conditions. Three single colonies were chosen and inoculated into 2 mL of Mueller-Hinton Broth (MHB) culture solution containing the corresponding antibiotics for plasmid maintenance, followed by overnight shake culture at 37 °C. The overnight cultures were then diluted 1:1000 (or 1:100 under drug pressure) in fresh MHB medium in 96-well microtiter plates, and three technical replicates were carried out. Optical density at 600 nm ($OD_{600}$) measurements were taken every 5 min over a period of 20 or 24 h at 37 °C to construct a growth curve. Relative growth rates under drug-free conditions, as well as the degree of growth inhibition under CTX or CZA compared with drug-free conditions, were calculated using an R script, and plots were generated using GraphPad Prism v9.0.0.

### Competition experiment

We performed bacterial growth competition experiments under antibiotic pressure (Fig. 1, part III). On the backbone of p65G-249R, the gene encoding blue fluorescent protein (mTagBFP) was subcloned inside the BamHI restriction site of the p65G-249R (named p65G-249R-B). The fluorescence microscope (Olympus, Tokyo, Japan) was used to confirm the successful introduction of the blue fluorescence. Images were processed with ImageJ (https://imagej.net/ij/). H1P-B and H2P strains were grown overnight in LB broth added with CZA and chloramphenicol (4 and 34 mg/L) at 37 °C, and the culture was washed and adjusted to an $OD_{600}$ of 0.5 in PBS. The two individual strains were mixed 1:1 immediately, then the mixture was cultured in LB broth that contained daily increased concentrations of corresponding antibiotics. The mixture was incubated at 37 °C for 20 h under constant shaking, percentage of various populations were enumerated using an LSRFortessa flow cytometer (BD Biosciences, Franklin Lakes, NJ, USA). Data were collected in FITC channel (for eGFP), PI channel (for mCherry) and BV421 channel (for mTgBFP) by 488, 587, and 399 nm excitation.

### Population-level plasmid dynamics model

A mathematical ODE model was constructed to simulate plasmid dynamics and population changes of bacterial communities under antibiotic exposure (Fig. 1, part III). We assumed the bacterial population consists of a plasmid-free (PF) subpopulation (with density at time $t$ denoted by $D_{PF}(t)$) and plasmid-containing subpopulations (including $D_{65G}(t)$, $D_{249R}(t)$, $D_{H2P}(t)$, and $D_{H1P}(t)$). The growth of each subpopulation, limited by environmental resource concentration, was modeled using a standard Monod term[60]: $G_i(S(t)) = \rho_i \mu_i(S(t))$, where $\mu_i(S(t)) = \frac{\mu_{\max,i} S(t)}{K_i + S(t)}$, characterized by the maximum growth rate $\mu_{\max,i}$ and half-saturation constant $K_i$. To simulate the effects of antibiotics (CTX and CZA, denoted as $A^T$ and $A^Z$, respectively), the sensitivity of

subpopulation $D_i$ to CTX and CZA was modeled with a killing rate $\gamma_i^T$ and $\gamma_i^Z$. And the hydrolysis efficiency of enzymes produced by bacteria against CTX and CZA was represented by $\beta_i^T$ and $\beta_i^Z$, respectively. In our model, the plasmid loss rate, denoted by $\lambda$, is based on the binomial distribution of plasmids during bacterial division, where each plasmid has a probability $P = 0.5$ of entering either daughter cell. For the $D_{H2P}$ subpopulation (with initial copy number $n_{p65G}$ of p65G and $n_{p249R}$ of p249R), we evaluated three plasmid loss scenarios: (1) the probability of losing all plasmids (p65G and p249R) is $(0.5)^{n_{p65G} + n_{p249R}}$, resulting in a plasmid-free subpopulation; (2) the probability of losing all p249R to become $D_{65G}$ is $P^{n_{p249R}}(1 - P^{n_{p65G}})$, where the segregational loss rate $\theta_{p249R}$ is defined as this probability; (3) the probability of losing all p65G to become $D_{249R}$ is $P^{n_{p65G}}(1 - P^{n_{p249R}})$, where the segregational loss rate $\theta_{p65G}$ is defined as this probability. To summarize, the system of ordinary differential equations governing the plasmid dynamics at the population level can be expressed as follows:

$$\frac{dS}{dt} = -\left(\mu_{PF}(S)D_{PF} + \mu_{65G}(S)D_{65G} + \mu_{249R}(S)D_{249R} + \mu_{H2P}(S)D_{H2P} + \mu_{H1P}(S)D_{H1P}\right) \tag{1}$$

$$\frac{dA^T}{dt} = -A^T\left(\beta_{PF}^T D_{PF} + \beta_{65G}^T D_{65G} + \beta_{249R}^T D_{249R} + \beta_{H2P}^T D_{H2P} + \beta_{H1P}^T D_{H1P}\right) \tag{2}$$

$$\frac{dA^Z}{dt} = -A^Z\left(\beta_{PF}^Z D_{PF} + \beta_{65G}^Z D_{65G} + \beta_{249R}^Z D_{249R} + \beta_{H2P}^Z D_{H2P} + \beta_{H1P}^Z D_{H1P}\right) \tag{3}$$

$$\frac{dD_{PF}}{dt} = G_{PF}(S)D_{PF} + \lambda\left(G_{65G}(S)D_{65G} + G_{249R}(S)D_{249R} + G_{H2P}(S)D_{H2P} + G_{H1P}(S)D_{H1P}\right) \\ - D_{PF}\left(\gamma_{PF}^T A^T + \gamma_{PF}^Z A^Z\right) \tag{4}$$

$$\frac{dD_{65G}}{dt} = (1 - \lambda)G_{65G}(S)D_{65G} + \theta_{p249R}G_{H2P}(S)D_{H2P} - D_{65G}(\gamma_{65G}^T A^T + \gamma_{65G}^Z A^Z) \tag{5}$$

$$\frac{dD_{249R}}{dt} = (1 - \lambda)G_{249R}(S)D_{249R} + \theta_{p65G}G_{H2P}(S)D_{H2P} - D_{249R}(\gamma_{249R}^T A^T + \gamma_{249R}^Z A^Z) \tag{6}$$

$$\frac{dD_{H2P}}{dt} = (1 - \lambda)G_{H2P}(S)D_{H2P} - D_{H2P}(\gamma_{H2P}^T A^T + \gamma_{H2P}^Z A^Z) \tag{7}$$

$$\frac{dD_{H1P}}{dt} = (1 - \lambda)G_{H1P}(S)D_{H1P} - D_{H1P}(\gamma_{H1P}^T A^T + \gamma_{H1P}^Z A^Z) \tag{8}$$

### Parameter estimation

Following a similar approach reported in the literature[61], we employed a Markov Chain Monte Carlo (MCMC) method with Metropolis-Hastings sampling to estimate growth-related parameters in the Monod model. We estimated the distribution of the maximum growth rate to half-saturation constant ratio $\mu_{\max,i}/K_i$ and the resource conversion rate $\rho_i$ from experimental growth curves of each subpopulation ($D_{PF}$, $D_{65G}$, $D_{249R}$, $D_{H2P}$, and $D_{H1P}$) grown in isolation (Fig. S8 shows the MCMC diagnostic plots). We evaluated the susceptibility of each subpopulation to CTX and CZA through single-drug dose-response experiments, estimating the killing rate parameters $\gamma_i^T$ and $\gamma_i^Z$ (Fig. S9). We measured plasmid copy number using digital qPCR to derive the plasmid loss rate $\lambda$, $\theta_{p249R}$, and $\theta_{p65G}$, with primer details provided in Supplementary Table 3. $OD_{600}$ values were converted to CFU/mL as previously described[62]. Numerical solutions of Eqs. (1–8) were obtained using the standard ODE solver ode45 in MATLAB R2021a (The MathWorks, Inc., Natick, MA, USA). All simulations were performed with time measured in hours. The final parameter values applied in the

numerical simulations of the model are listed in Supplementary Table 4. The corresponding code used is available online (see Code availability).

## Raw data analysis for multicopy heterogeneity

We conducted a comprehensive analysis of β-lactamase gene multicopy heterogeneity by genome-wide raw read alignment of a broad range of *K. pneumoniae* isolates (Fig. 1, part III). The prevalent β-lactamase gene alleles in *K. pneumoniae*, such as $bla_{SHV-12}$, $bla_{TEM-1}$, $bla_{CTX-M-15/65}$, $bla_{NDM-1}$, $bla_{KPC-2}$, and $bla_{OXA-48}$, were used as references for raw reads mapping. Strains carrying genes similar to these alleles were all evaluated for their multicopy heterogeneity status. Considering that the heterogeneity of $bla_{CTX-M}$ may arise from genetically distant ancestral groups, the $bla_{CTX-M-1}$-group and $bla_{CTX-M-9}$-group, we selected $bla_{CTX-M-15}$ and $bla_{CTX-M-65}$ as representative references for the two groups, respectively, and took the union of strains carrying multicopy heterogeneity for these alleles. Gene copy number estimation was performed based on previously established methods[63], while single-nucleotide variations (SNVs) and insertions/deletions (InDels) were analyzed using GATK 4.5.0.0[64], Samtools 1.1, and bcftools 1.20. The corresponding code used is available online (see Code availability). When mapping raw reads to the consensus sequence, sites with less than 30x coverage depth are not considered. The data sources included isolates collected by our laboratory from a large number of hospitals across China between 2017 and 2019[65,66]. Additionally, we downloaded all publicly available *K. pneumoniae* Illumina sequencing reads from the SRA database (https://www.ncbi.nlm.nih.gov/sra), covering the release period from January 1 to December 31, 2023. Some of these strains are derived from humans, with most of the strains coming from patients, while others derived from nonhuman sources such as livestock farming, food, and the natural environment. The detailed BioProject lists are provided in Supplementary Tables 5 and 6.

## Statistical analysis

The *t*-tests were performed to compare colony counting numbers, and fitness cost. One-way ANOVA was used to analyze gene copy numbers. Two-way ANOVA was performed to compare the percentages of dual-fluorescent cells under CTX or CZA pressure, as measured by flow cytometry. Chi-square tests were used for comparison of multicopy heterogeneity in strains from humans or nonhuman sources. *p* values <0.05 were considered significant. The statistical analyses were implemented in GraphPad Prism v9.0.0.

## Reporting summary

Further information on research design is available in the Nature Portfolio Reporting Summary linked to this article.

## Data availability

The novel variants of $bla_{CTX-M}$ we discovered in this study, $bla_{CTX-M-247}$, $bla_{CTX-M-248}$ and $bla_{CTX-M-249}$, have been submitted and designated in GenBank under accession number MZ379780, MZ379781, and MZ379782, respectively. All raw genomic data for strains ICU-3 and ICU-18, under different antibiotic pressures, generated from the Illumina and Nanopore platforms, have been deposited in the SRA under BioProject accession number PRJNA1194354. The pICU-3 and pICU-18 sequences are available under GenBank accession numbers PX405700 and pX405701 [https://www.ncbi.nlm.nih.gov/nuccore/pX405701], respectively. Additionally, the genomic raw sequence data of *K. pneumoniae* from our laboratory were obtained from two surveillance projects (PRJNA1290706 [https://www.ncbi.nlm.nih.gov/sra/?term=PRJNA1290706] and PRJNA1291361). Source data are provided with this paper.

## Code availability

Simulation codes and the codes for GATK-based SNV/InDel calling are available at https://doi.org/10.5281/zenodo.18264007.

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

## Acknowledgements

This work was supported by the National Key Research and Development Program of China (No. 2023YFC2307100 to Y.Y.) and the National Natural Science Foundation of China (No. 82272373 to Y.J. and No. 82402666 to P.Z.). This work was presented as a Top Abstract in an oral report at ESCMID Global 2024, Barcelona. We thank Prof. Hangjin Jiang from the Center for Data Science, Zhejiang University, Hangzhou, China, for his valuable assistance with the mathematical models.

## Author contributions

R.W. and Jingyi Z. contributed equally to this work. Y.J., Y.Y., and J.W. conceived, designed, and supervised the study. R.W. designed and performed the experiments, data curation, formal analysis, visualization, and writing—original draft. Jingyi Z., Y.L., and Junxin Z. designed the experiments. P.Z., D.Z., X.D., and Y.C. performed sample collection and storage. Q.S., Yanfei W., Yinping W., W.H., and J.R. performed data curation and data analysis. Jingyi Z., X.W., H.L., J.R., J.Q., X.H., J.W., and Y.Y. reviewed and edited the manuscript. S.Q. provided technical support (flow cytometry). Y.J., Y.Y., and P.Z. acquired funding. All the authors reviewed the manuscript.

## Competing interests

The authors declare no competing interests.

## Additional information

[1]Department of Infectious Diseases, Sir Run Run Shaw Hospital, Zhejiang University School of Medicine, Hangzhou, Zhejiang, China. [2]Key Laboratory of Microbial Technology and Bioinformatics of Zhejiang Province, Hangzhou, Zhejiang, China. [3]Regional Medical Center for National Institute of Respiratory Diseases, Sir Run Run Shaw Hospital, Zhejiang University School of Medicine, Hangzhou, Zhejiang, China. [4]Centre of Laboratory Medicine, Zhejiang Provincial People's Hospital, People's Hospital of Hangzhou Medical College, Hangzhou, Zhejiang, China. [5]Department of Clinical Laboratory, Sir Run Run Shaw Hospital, Zhejiang University School of Medicine, Hangzhou, Zhejiang, China. [6]Department of Critical Care Medicine, Sir Run Run Shaw Hospital, Zhejiang University School of Medicine, Hangzhou, Zhejiang, China. [7]Zhejiang Key Laboratory of Precise Diagnosis and Treatment of Abdominal Infection, Sir Run Run Shaw Hospital, Zhejiang University School of Medicine, Hangzhou, Zhejiang, China. [8]Department of Life Sciences, University of Bath, Bath, UK. [9]European Molecular Biology Laboratory, European Bioinformatics Institute (EMBL-EBI), Wellcome Genome Campus, Hinxton, Cambridgeshire, UK. [10]College of Systems and Society, Australian National University, Canberra, ACT, Australia. [11]These authors contributed equally: Rui Weng, Jingyi Zhu.
✉e-mail: jw4350@bath.ac.uk; yvys119@zju.edu.cn; jiangy@zju.edu.cn

