## [Transparent Peer Review file · Nature Communications]

Heterogeneous multicopy of *bla*_{CTX-M} variants on the same plasmid enhances evolutionary adaptability in clinical *Klebsiella pneumoniae*

Corresponding Author: Dr Yan Jiang

Version 0:

Reviewer comments:

Reviewer #1

(Remarks to the Author)

Weng et al presents one model case of paralogous gene evolution in one genome by focusing on a plasmid detected in a hospital. By conducting additional survey in publicly available raw reads, they found that the paralogous (beta-lactamase) gene evolution in one plasmid is common. Furthermore, the authors characterized the nature of four CTX-M variants in vitro. These are solid and important data. The statement in the abstract is valid. The manuscript is clearly written.

However, I am not sure about the significance of discussion on the stability of gene heterogeneity, comprising a large part of this study. The fitness advantage of the “dual-copy genes on one plasmid” state (DCG model) over the “dual-copy plasmids” state (DCP model) under drug-switching condition (Fig 5eg) is immediately obvious as far as plasmid loss can occur, unless huge fitness cost arises from the DCG state. The points of discussion should be “when does DCP or mix model win?”. Therefore, to discuss the conditions where DCP/PCG/mix state wins, I think that the authors need to implement mathematical model for the serial passage or competition experiments considering the fitness advantage of plasmid-bearing cells and plasmid loss rate. To address the above problem, ideally, all experiments should be designed using a low copy number and stable plasmid such as mini-F to mimic the condition of the real situation.

Although the authors mentioned the cost of these cell types in the absence of antibiotics (Fig S1), competition experiments and stability test were performed in the presence of antibiotics where plasmid-free cells are immediately killed. Therefore, the cost information is not really linked to experimental data.

The most parts of figure are used to explain the experimental set-up. Figure information should be more condensed. For example, Figure 4 can be condensed to only two line plots: one from flow-cyto (Fig4c), one from CFU (Fig4de). Figure 5 can be condensed to only Fig5eg.

It appears that to select for specific (mix model) cell types, a high concentration of CTX, 256 ug/ml, was used in the experiment. The concentration should be in clinically achievable range (< or = 100 ug/ml).

MINOR POINTS:

FIG1a. Gray area does not precisely connect the circular map and gene map of tandemly amplified loci.

FIG3b. I think that gene map on plasmids is unnecessary as they are redundant with panel a.

FIG4. Plasmid stability data measured by flow cytometer (a-c?) and by CFU count (d, e).

c. Please clarify in legend whether this graph is based on CFU count or summary of flow cytometer counts. d-e: d and e show the same information. I suggest the author should show the data of proportion of the resistance gene-carrying cells by line plot like panel c.

Fig5. d label. Please add ug/ml after antibiotic concentrations because it is not immediately clear whether those are symbol of CTX-M variants.

Reviewer #2

(Remarks to the Author)

Regarding the manuscript "The heterogenetic multicopy of blaCTX-M variants on one plasmid optimizes the evolution adaptability in the clinically isolated *Klebsiella pneumoniae*" by Weng et al.

This manuscript presents a novel and well-supported study on how multicopy heterogeneity of blaCTX-M variants on a single plasmid contributes to the adaptability and stability of antibiotic resistance in *Klebsiella pneumoniae*.

Overall, the manuscript is well-written, and the study is highly relevant, novel, and scientifically rigorous, with important clinical relevance. The combination of whole-genome sequencing, digital PCR, amplicon sequencing, and in vitro modeling provides strong experimental support for the proposed mechanism.

However, the comprehensiveness of the study makes it somewhat difficult to follow, and some sections appear misplaced, affecting the clarity and flow of the manuscript. Below, I have outlined major and minor comments that would benefit the manuscript.

Major Comments

1. Structural Issues & Misplaced Sections:

Several sections appear misplaced within the manuscript.

I recommend restructuring the introduction and results sections to improve clarity and logical flow. Some specific examples:

- Lines 99-111: This section describes the findings of the study, which should not be placed in the introduction. The introduction should focus on background information and knowledge gaps, leading to the study's aim and objectives.
- Lines 120-122: This section describes the aim of the study, which should be included in the introduction, not the results section.
- Lines 331-337: Discussion elements appear within the results section. These should be moved to the discussion section.
- Lines 359-361: Similarly, discussion/conclusion elements are found in the results section and should be relocated.

I suggest a thorough review of the manuscript structure to ensure that each section follows a logical order.

2. Results Section is Long & Difficult to Follow

The results section is long, which is expected with the comprehensiveness of the study, making it a little challenging to follow. Consider referring to the steps in the flow-diagram (Figure 1a) at relevant points in the results. Possibly expand the flow-diagram to further guide the reader through the workflow and experimental design. This small change could significantly improve readability and clarity.

3. Clarification of Nucleotide and Amino Acid Mutations

In Lines 123-124, the authors mention the noteworthy changes, including: "A397G, C508T in the bla...." Then, in Lines 131-133, the authors mention the corresponding amino acid changes: "Ser130Gly and Pro167Ser".

However, based on the alignment in Figure 1b, these mutations appear to correspond to amino acid positions 133 and 170, not 130 and 167.

Clarification needed: Are the mutation positions correctly assigned? If different numbering conventions were used, this should be explicitly stated in the Methods section.

4. Statistical Analysis

Throughout the manuscript, results are stated as "significant", but p-values are missing in multiple places.

Example: Line 192 mentions significance without statistical support.

Include p-values wherever statistical significance is claimed.

5. Figures are Overloaded – Consider Moving Some to Supplementary Material

While only seven figures are presented, each figure contains a high number of panels.

I recommend to move redundant figures to supplementary materials to streamline the main text.

Recommended changes:

- Move Figure 1b and 1c to supplementary material.
- Move Figure 3c and 3d, but expand Figure 1b by integrating the findings for the five different models.
- Move Figure 4a, 4b, and 4d to the supplementary.
- Move entire Figure 5 to the supplementary.

This restructuring would help reduce figure complexity while keeping essential data accessible.

Minor Comments

I have a few minor comments on wording and terminology

- Line 55: "The antibiotics developed by humans..." What about the naturally occurring ones?

- Line 58: "Carbapenems were once considered the last-resort..." Carbapenems are still a last-resort antibiotic, so this should be reworded for accuracy.

- Line 457: "We postulate..." think it should be changed to "We presented" or "We constructed"

- Lines 642-643: "alignment of genome raw reads in a broader range of *K. pneumoniae*." Reword for clarity. Perhaps: "Genome-wide raw read alignment of a broad range of *K. pneumoniae* isolates."

- Figure 2 – Circular Map: What software was used to generate the circular map in Figure 2? This should be mentioned in the figure legend or methods section.

Reviewer #3

(Remarks to the Author)

This manuscript describes a study looking at blaCTM-65 and variants detected at different copy numbers on the same plasmid in a clinical isolate of *Klebsiella pneumoniae*, with different gene combinations and copy numbers influencing resistance to ceftazidime, cefotaxime and ceftazidime+avibactam. Analysis of available sequences suggested that multiple copies of AMR gene variants in a single isolate may be more common than is usually recognized. While the results are potentially interesting and useful, there are some problems. Better sequence analysis, with reference to the available literature, is needed to improve accuracy in some of the descriptions. The detail about the results of experiments is wordy and uses unnecessarily complex terms, making it rather inaccessible and overwhelming. The text would benefit from language editing to make it clearer and easier to follow (and there is some odd/unscientific phrasing in places) and its could be condensed – some examples and suggestions are given below. There are many large complex figures and some thought about how these could be reduced would also be beneficial (e.g., parts showing detailed results of experiments could be made Supplementary, with summaries only in main text) – again I have given some suggestions below.

Major points

1) blaCTX-M-65 and variants, Lines 100-2,129-136,

Lines 100-2 - the information about blaCTX-M-65 could be expanded, explaining that blaCTX-M-65 belongs to the blaCTX-M-9 group but does not have a mutation resulting in Asp240Gly usually associated with CAZ resistance (as found in e.g., in blaCTX-M-27).

Lines 131-46 could be condensed and simplified e.g., it is not really relevant when blaCTX-M-249 or the other variants were named.

Line 131-3 – this is unclear. What was cloned that revealed the variants?

Line 132 – are these amino acid positions numbered according to the Ambler scheme for class A beta-lactamases (PMID: 2039479). If so, this should be clearly stated.

Line 134 – it is antibiotics that have MICs, not isolates.

Line 136 – should be “mutations in blaCTX-M-65”

Line 138 – how were these “clonal isolates” obtained? And it should be “alleles” here.

Lines 145-6 – do not make sense and needs rewording.

2) Plasmids, contexts/amplification of blaCTX-M-65 and variants, Lines 123-5, 179-216, 312-397

The assembled sequence of pICU-3 should be submitted to GenBank, along with the sequence of pICU-18, in addition to raw reads. It is possible to add comments to GenBank entries to explain that e.g. the sequence submitted is a representative of several variants of the same plasmid that can have different numbers of copies of a region that includes blaCTX-M-65 but also the other variants.

The structure of pIUC-3 needs to be explained in the text and compared with pICU-3.

Lines 186-208 - from Fig. 2, but not really explained in the text, it seems that the copy of blaCTX-65 that is not amplified also does not undergo the mutations? If so, this might need to be taken into account when calculating/drawing conclusions from the number of copies e.g. plasmid copy number will increase the copy numbers of both this unchanging copy of blaCTX-M-65 and whichever variant is in the other position, which could be blaCTX-M-65, 247, 248 or 249, while gene amplification will not affect the copy number of the non-amplified blaCTX-M-65 copy. It is not clear if/how this has been taken into account in the calculations.

Lines 182-3 – how was this calculated? What are these numbers?

Line 188-9 – are there any clues in the sequence about what is responsible for this increase in plasmid copy number? This should be looked at.

Line 193 – one of the ONT raw reads provided has 7 copies of the TU, the other 3 plus probably a truncated fourth one. How does this fit with “about three” stated here?

Lines 197–208 – it’s not entirely clear from Methods how these proportions were determined/calculated and a bit more detail might be helpful.

Lines 209-10 – presumably “long reads from isolates grown in 64 mg/L CAZ” is meant? This should be rewritten.

Line 212 – this needs to be looked at further and explained better and more accurately. IS903, or a truncated copy, is commonly found downstream of blaCTX-M-9 group genes in ISEcp1 “transposition units” (TPUs) responsible for moving these genes. Examination of the pICU-3 sequence and raw long reads provided shows that, as expected, IS903 has nothing to do with the amplification. The amplified region instead seems to be a 4844 bp IS26 “translocatable unit” (TU; positions 58367-63210 in pICU-3 sequence provided), something that has been reported previously for different AMR genes (e.g. Refs 33, 34 cited here) forming a chain of “pseudocompound” transposons. The authors need to look at recent reviews and/or papers on IS26 mechanisms of transposition (mainly by RM Hall and colleagues, but also M Chandler and colleagues). I also would suggest looking at whether this TU has been seen before and at papers on the plasmid contexts of blaCTX-M-65 (e.g. PMID: 22984205).

Lines 213-4 – this region cannot not undergo the same type of IS26-mediated tandem amplification as the flanking IS26 copies are in opposite orientations.

3) Lines 218-329 Models of heterogeneity

This part of the text gets hard to follow and could be simplified and condensed.

Line 219 – “plasmid-harboring beta-lactamase genes” does not make sense. Presumably “plasmids harboring” and “blaCTX-M genes” could be specified here, rather than “beta-lactamase genes”.

Lines 221-2 – genes are cloned in fusion with genes, not proteins, and the gene names are blaCTX-M-65 and blaCTX-M-

249. This should be written more accurately (and maybe moved to Methods).

Line 222-5 – “for example” should be deleted if you then list everything that was constructed. Maybe just saying here that plasmids encoding the following fusions were constructed: CTX-M-65-eGFP (p65G), CTX-M-249-mCherry (p249R) and both CTX-M-65-eGFP and CTX-M-249-mCherry (p65G-249R), using R to indicate the red fluorescence rather than “m” for mCherry, and explaining this and also that both colours together give orange.

Lines 226 – the plasmid names could then be used to simplify the text i.e. Model 1 is DH5alpha with p65G (strain 65G), Model 2 is DH5alpha with p249R (strain 249R), Model 3 is a 1:1 mixture 65G and 249R, Model 4 is DH5alpha with both p65G and p249R and Model 5 is DH5alpha with p65G-249R (or maybe consider putting this information a table). Having two names for Model 4/DCPs and Model 5/DCGs is also confusing and seems unnecessary. The names DCP and DCG are confusing as they are too similar (and the “s” at the end is not needed). Any further necessary details (e.g. of transformation) and Lines 234-6 could be left to Methods (or Fig. 2d legend, where they are already) and also consider whether the information in Lines 236-40 would be better mainly moved to Methods.

Lines 240-3 – it might be clearer to say that strain 65G grew up to high concentrations of CTX, but only at low concentrations of CAZ, while 249R grew up to high concentrations of CAZ, but not in CTX, both as expected, if this is what is meant (“tolerable drug” does not make sense here).

Lines 243-7 could then say that Models 3, 4 and 5, which express both CTX-M-65 and CTX-M-249 in different configurations, all showed growth up to relatively high levels of both CTX and CAZ, then explain the differences between Model 3 and Models 4 and 5.

Lines 253 – unclear, needs rewording.

Line 257 – suggest “selection with”.

Line 259-65 – wording does not make sense and is unclear – say that Model 4 has CTX-M-65 (green) and CTX-M-249 (red) encoded by two different plasmids in the same cell. Then that in Model 4 orange fluorescence was replaced by green in the presence of CTX (resistance conferred by CTX-M-65) and by red in the presence of CAZ (resistance conferred by CTX-M-249), then information from Lines 266-9.

Lines 265-71 – could just state that Model 5 has CTX-M-65 and CTX-M-249 encoded on the same plasmid and that orange fluorescence was maintained and reword Lines 269-71 – e.g., I don’t think that using “paralogous” here is helpful.

Lines 272-288, L290-329 – much of the current wording does not make sense or is unclear - the same principles as above can be applied to explain this more simply and clearly.

4) Universality of resistance gene heterogeneity, Lines 331-61

Again, the language could be simplified.

Lines 332 – suggest something like “Having multiple copies of different variants of a resistance gene that confer different phenotypes”.

Lines 335, 350-1, 355-7 – blaSHV is an intrinsic chromosomal gene in *K. pneumoniae*, which is the source of plasmid-borne variants. This needs to be properly explained and I would suggest looking at PMID: 17999968, which discusses blaSHV variants on plasmids and the chromosome. I would also suggest looking at PMID: 28812563 about blaTEM. This could all have implications for the analysis.

Lines 335-57 – the name blaOXA encompasses many families that are not closely related, so it does not make sense to give a single % for multicopy heterogeneity for blaOXA. Was a particular blaOXA looked at? If so, this must be stated. If not, then the analysis may need to be redone.

Line 342 – is this sequence data available and/or associated with a publication? If so, please give the BioProject and/or citation. Suggest “raw sequence data from this species”

Line 351-57 – which blaCTX-M group or did you look at multiple groups? Major sequence differences between members of different blaCTX-M groups (Group 1, Group 9 etc) are not due to mutations, but because the source genes are from different *Kluyvera* species (there are also hybrids between Group 1 and Group 9 genes). Minor changes between blaCTX-M genes within each group are probably explained by mutations that have occurred since capture.

5) Other scientific points

Lines 40, 48, 334-5 – genes encode proteins, they are not encoded. Suggest “carried”/“carry” or “plasmid-borne”, as appropriate.

Lines 75, 382 – it is not clear what is meant by “the CTX-M-14 isoform” here. bla gene variants are assigned numbers by NCBI. A CTX-M-14 sequence with these amino acid changes appear to have been numbered CTX-M-219, described as “inhibitor resistant”, with the accession CP019080 that cites Ref 14. See <https://www.ncbi.nlm.nih.gov/pathogens/refgene/#blaCTX-M-219>.

Line 90-1 – many large conjugative plasmids that typically carry AMR genes are present at around only one copy/cell and “duplication state” needs rewording.

Line 116 – homology is not quantifiable, so what is really meant by “highly homologous” here? “closely-related”? This is from the SNP distance of <10? Also suggest “isolates” here, Line 122 etc.

Lines 123-4 – which blaKPC allele does ICU-18 have? This must be stated.

Line 136 – this should be “mutations” not “mutants”

Lines 139, 403, 405 etc. – I don’t think that it makes sense to use “wildtype” – it can be deleted.

Lines 171-7 – are there other studies reporting on the effects of these mutations?

Line 200 – I would suggest making these % less precise.

Line 210 – specify blaKPC-157 here.

Line 282 – LB originally stood for “lysogeny broth”.

5) Discussion

This is long and wordy and could be condensed. The discussion section should not repeat information in Results (or Introduction) and comments on the rest of the manuscript need to be considered.

Line 365, 482 – is this really completely novel?
Lines 365-76 – this seems more like introductory information and parts seem to repeat the Introduction.
Lines 382-3 – wording does not make sense.
Line 392 – “reflected on several level of meanings” does not make sense.
Line 394-5 – the plasmid copy number and how it was determined and the distance between the blaCTX-M-65-like genes should be stated in Results.
Line 397 - “elements” is not needed – these are IS.
Line 416 - “AS” is not needed here and suggest referring to the paper without naming the authors
Line 417 – the genes are blaTEM-1 and blaTEM-12
Line 428 – it is the constructed plasmids that are lost?
Lines 434-6 – why wasn’t this data included?
Line 436 – “inhibitor” is not needed here.
Line 438 – wording does not make sense.
Line 446 – I think that heteroresistance can also be due to copy number? See papers by D. Anderson.
Line 452 – use “and/or” and delete “dual site mutations”
Lines 457-65 – this information might be in the legend of Fig. 7, which currently very short.
Lines 468-9 – unclear what is meant here.
Line 470 – “studies”
Lines 471-5 – is this really true for long-read if multiple copies of the gene are in different genetic locations, as for copies of blaCTX-M-65 (or variants) both in pICU-3 here?
Line 484 – what is meant by “horizontal transfer elements” here?
Line 492 – “introduction” seems like the wrong word here – maybe “recognition”

6) Methods

These lack detail/clarity in places, but text in other places could be condensed.
Primers - although the text refers to Table 3 in several places, it is not always immediately clear which primer sets are being referred to and it might be helpful to state the primer names in the text or better indicate in Table 3 which section they were used for.
Line 499 – presumably “species identification” is meant here?
Lines 523-7 – the wording does not make sense here and more details are needed. What was cultured for the PCR?
Lines 573-84 – this section can be condensed. The first sentence is not really needed, is something like “Three reporter plasmids based on pACYCDuet1 were constructed” and state which plasmid has which gene. Genes are cloned in fusion with genes, not proteins, and first explain that pACYCDuet1 has a cat gene.

7) Other minor/formatting/English

Title needs some rewording – suggest “Heterogeneity in multiple copies of blaCTX-M-65-like genes in a clinical isolate of *Klebsiella pneumoniae*”.
Throughout – odd/unscientific phrasing needs to be changed, e.g., Line 33 “combat”, Line 42 “lock in, Line 55 – “constant battle”, Line 77 - bacteria do not really “choose their evolutionary paths”, Line 386 - “ingeniously explored”, Line 423 “solidified”, Lines 448, 454 “locked on/onto”.
Throughout – “the blaCTX-M-65” is incorrect. This should either be “the blaCTX-M-65 gene” or preferably just “blaCTX-M-65” e.g. Line 102. (except Line 36, where “the” is needed before “CTX-M-65-type”). Also, for other gene/allele names. “bla” in the name indicates the gene
Throughout – the abbreviation CZA is used for ceftazidime/avibactam. The common abbreviations CAZ and CTX, as used in figures, could also be defined and used in text.
Line 37-8 – I don’t think that “monoclonal” is commonly used to describe bacteria? Maybe “a single strain was found to have multiple copies of blaCTX-M-65 or minor variants that conferred different spectrums of resistance”.
Line 42, 412 etc – I am don’t think that that “paralogous” is the clearest term to use.
Line 47 – suggest “clinical isolates” rather than “patient-isolated strains”.
Lines 47-50, 61-62, 73-74 need rewording.
Line 80-1, 108, 384, 389 suggest just “trade-offs”, rather than the “the trade-offs effect” – or at least “the trade-off effect” and reword “may introduce new roles”.
Line 84-5 – “Compared with... mutations, genetic amplifications” – i.e. delete “the”
Line 93 – either “multicopy resistance plasmids” or “multiple copies of a resistance plasmid”
Lines 95-6, 110 – need rewording.
Lines 99-102 – this long sentence might be better split but certainly needs rewording.
Line 111 – “considerable numbers”
Line 122 – “The results showed” not needed. Suggest “Both isolates belong to ST11 and differ by <10 SNPs”
Line 126 – “clones with blaKPC-157”
Lines 157-60 – suggest “each has one of the two mutations seen in CTX-M-249” and “the two common third generation cephalosporins” is not needed (they have already been discussed).
Line 180 – “long-read sequence assembly”
Line 227 – “respectively” needs to be at the end of the sentence.
Line 228 – “mixing of”
Line 261 - “observation of fluorescent microscope image” is not needed
Line 337 – “mutation” i.e. no “s”.
Lines 339-40, 354-5, 654-5 etc. – I would suggest using “isolates from humans or other sources”.
Line 513 – “was performed”
Line 517 – “between ICU-3 and ICU-18 was calculated”.

Line 530 – don't need both “enzymes” and “enzymatic” here.

Line 534 - “successful different” needs rewording.

Lines 540-1 do not make sense and need rewording. I don't think that “velocities” is the right word here.

Line 557, 569 etc. - “are listed”, not “were listed”

Line 562 – “as for amplicon sequencing”.

Lines 562-3 – the wording here does not make sense.

Lines 568-9 – this is unclear and needs to be explained better.

Lines 578 – “removed”

Line 579, 80 – “DCP strains”, “DCG strains”

Line 581 – “dual antibiotic”

Line 588 – not clear what is meant here and “including” is not needed.

References are normally placed before figure legends. Ref 8 needs page numbers or equivalent. Something seems to be wrong with page numbers in Ref 49. Journal titles need to be correctly abbreviated. Ref 4, 12 etc., remove Title Case.

Species and gene names etc. need to be correctly formatted.

8) Figures

General – some parts could be moved to supplementary, shorter titles would be better and legends could be condensed.

Fig. 1

I think that most parts of this figure are not needed or could be moved to supplementary. I would suggest retaining the grid showing resistance profiles from part a (with amino acid changes for each CTX-M variant listed) plus part d in the main manuscript. Maybe consider adding CTX-M-219 (“CTX-M-14 isoform”) discussed in the text to part b, if retained as Supplementary.

Fig. 2

Part a - on the circular diagram of the plasmid at least the two IS26 flanking blaCTX-M-65/247/248/249 should be shown, plus preferably the other AMR genes, IS and transposons in this region and surrounding the other copy of blaCTX-M-65 (see ISfinder). The diagram to side should properly show the boundaries of IS26 and of all copies of the amplified unit. The blaSHV gene number should be added.

Part b CZA 32 mg/L – colour is too pale, hard to see

Fig. 3

Suggest moving parts a-c to supplementary, simplifying the plasmid diagrams in part b and showing each model above the corresponding plot in part d. The wording of this part of the legend also needs improving – e.g. it is not expression of plasmids, but of fused genes.

Fig 4

Part a – suggest stating clearly what is in DCPs and DCGs on the figure, rather than having to refer to text in legend. Using the names suggested above would help with this.

Part b – this could be supplementary and just show the results in part d in the main manuscript.

Part c – this could be supplementary and just show the results in part e in the main manuscript

Line 749 in legend – respectively appears incorrect here – it is not clear what is being referred to.

Fig. 5

Parts a and b do not both need to be shown.

Parts d and f could be moved to supplementary, just showing the summary of results in parts e and g.

Fig. 6

See comments above on analysis of blaSHV, blaCTX-M and blaOXA genes.

Line 781 – “including” is incorrect here, as all genes analysed are then listed

Fig. 7

What is the evidence for multidrug resistance pressure resulting in two copies of blaCTX-65 on the plasmid? What to the following unlabeled arrows represent?

9) Supplementary

Table S1

How do values compare with previous determinations for CTX-M-65, if available?

Table S3

It should be possible to fit this table on a single page, for ease of reading, e.g. reduce margins, “amplification” is not needed in every line of the last column, remove “the” before “blaCTX-M-65” etc. Final column heading could be “Purpose” instead of “Usage”. PsiB should be pisB in final column, if the gene is meant.

CTX-M-F/R/P – blaCTX-M should be used for the gene and which blaCTX-M genes/alleles would be amplified?

Table S4

This could be split into two tables, one with all isolates from humans, with this stated in the title and the final column removed, and one for other host species giving the details. I don't think that the “Study” column is really needed – readers can get this information from the BioProject, if required. BioProject numbers and host species would be easier to read if aligned to the left.

Version 1:

Reviewer comments:

Reviewer #1

(Remarks to the Author)

The authors responded to my comments appropriately. Simulation results based on real experimental data added strength to their message. I recommend the authors' current results for publication in this journal.

Legibility of this manuscript has been improved. However, I still found several vague descriptions/presentation in the current manuscript. Furthermore, the simulation program was not released to public yet. Currently, readers are difficult to reproduce data analysis.

The following minor points should be improved.

Line 117-121. This sentence is a little too long for me. I recommend the authors rephrase this sentence.

Line 628: The software platform: Python, R, C++, or MATLAB etc is not described. Codes should be deposited to public repository such as Github, figshare.

Line 628: Unit of time used in simulation is hour? Please mention.

Line 699: I think that Code availability section is also needed.

Line 1038 panel e,f: Does this assume "one-round" batch culture competition assay starting with with 1:1 ratio mixture of SPD and DPD at the density of 1/100 of final density, but not serial batch culture transfer like experiments in panel c,d ?. Please clarify.

This is just suggestion: Supplementary figure S9- panel c,d,e,f are important results. I suggest the authors place these panels as a new main figure or add these to Figure 5 (competition assay results figure).

Line 666 and Line1058: right panel Y-axis. The unit in these growth curves appear to be CFU/mL. How OD600nm values (source OD data) were converted to CFU/ml is not clear. Please mention in the main text or legend.

Supplementary Table 5: μ_{max}/K should read μ_{max}/K_i .

Reviewer #2

(Remarks to the Author)

I had the opportunity to review the first version of this manuscript and have now had the pleasure of evaluating the revised version entitled "Heterogeneous multicopy resistance genes on a single plasmid enhance evolutionary adaptability in clinical *Klebsiella pneumoniae*".

The authors have clearly made a substantial effort to address the comments and suggestions provided by all reviewers. The revised manuscript shows significant improvement in structure, clarity, and overall readability.

The reorganization of figures and sections has made the study more coherent and easier to follow.

The updated workflow figure (Fig. 1) effectively guides the reader through the study and provides a clearer overview of the experimental design.

Overall, the authors have responded thoroughly and constructively to the reviewers' feedback, and the manuscript has improved notably as a result. The revisions have strengthened both the presentation and the scientific clarity.

I find that all my major concerns from the first review have been addressed satisfactorily.

At this point, I only have a few minor comments.

Minor comments

The language occasionally shifts between tenses, so a brief consistency check would also be beneficial.

Line 41–46:

"Within this clonal population, blaCTX-M-65 and three of its variants were present..."

Please list all alleles explicitly here (e.g. blaCTX-M-65, blaCTX-M-249, ...) to avoid ambiguity.

Additionally, blaCTX-M-65 is a specific allele within the CTX-M-9 group, not a group itself. Therefore, the other genes are not truly "variants of blaCTX-M-65." They are better described as other CTX-M-9 group blaCTX-M alleles closely related to blaCTX-M-65.

Suggested rephrasing: "Within this clonal population, blaCTX-M-65 and additional CTX-M-9 group blaCTX-M alleles (blaCTX-M-247, blaCTX-M-248 and blaCTX-M-249*) closely related to blaCTX-M-65 were identified."

Please apply consistent terminology throughout the manuscript.

Line 77–79:

"...or blaKPC they carry has undergone..."

Consider replacing the first "or" with "the" and moving this phrase towards the end of the sentence for smoother flow, together with the information in lines 80–81.

Line 108–109:

See comment about line 41-46. As correctly noted in line 112, blaCTX-M-65 belongs to the CTX-M-9 group. Therefore, the other genes mentioned should not be referred to as “variants of blaCTX-M-65,” since they represent distinct alleles within the CTX-M-9 group, not subvariants of blaCTX-M-65. Please rephrase to “other CTX-M-9 group blaCTX-M alleles (closely related to blaCTX-M-65).” blaCTX-M-65 was the first to be described in the literature; however, due to heterogeneity among these alleles, it is not possible to determine which appeared first in the isolate, and thus which may be the derivative of the other.

Line 132–133:

Consider clarifying this sentence to read: “...included the nucleotide substitution A392G in...”

Line 415, 422, and 426:

Figure 7 is first introduced in the Discussion. New material should not be presented at this stage. Consider referring to Figure 7 earlier in the Results section..

Figure 1:

Consider referring explicitly to the different parts of Figure 1 within the Methods section to improve readability and guide the reader through the workflow.

Reviewer #3

(Remarks to the Author)

This revised manuscript describes detection of blaCTM-65 and minor variants on the same plasmid in a clinical isolate of *Klebsiella pneumoniae*, with different gene combinations and copy numbers influencing resistance to ceftazidime, cefotaxime and ceftazidime+avibactam. Different models with different combinations of plasmids carrying blaCTX-M-65 and/or variants were used to explore the dynamics. Available sequences were also examined to look for multiple variants of selected bla genes in a single isolate, which suggests that this may be more common than is usually recognized. While the data are substantial, interesting and useful, and the authors have addressed many points raised by all three reviewers, improving the manuscript, there are still a few problems with scientific accuracy etc, as detailed below. While I think that the manuscript structure is better, further condensing/reorganising and editing/simplifying the text could still improve accuracy, ease of reading and flow – some examples are given below, including figure legends.

Scientific points/accuracy

1) ORIGINAL COMMENT: Line 90-1 – many large conjugative plasmids that typically carry AMR genes are present at around only one copy/cell

RESPONSE: Yes, there are some large plasmids that exist only as one copy. However, it has been reported that plasmid copy number ranges from only slightly higher than chromosomal copy number in low copy plasmids to tens of copies per cell in high copy plasmids, which means that most plasmids have more than one copy per bacterial cell (Nat Rev Microbiol, 2021; 19(6):347–359). Our wording was not sufficiently rigorous, and we have revised “which exist” to “most of which exist”.

NEW COMMENT: The paper listed here (cited as Ref 22) also states that “Low copy number plasmids (LCPs) are typically large (from tens to hundreds of kilobases), have a low copy number and are frequently conjugative”. Many plasmids carrying AMR genes fit this category – probably including pICU-3/pICU-8 studied here - so while “most of which exist” could be numerically accurate, I would still argue that the most important plasmids for spreading AMR are mainly low copy.

This also relates to Reviewer 1’s first comment, about using a low-copy number plasmid for models. While I appreciate the problems with this explained in the response to this comment, I think that it is important to at least state the copy number of pACYCDuet-1 somewhere (apologies if this is already there and I missed it) and maybe also explain that there were technical problems with using a lower copy number plasmid. I think it’s also important to acknowledge (in the Discussion) that the models here may not exactly reflect what might happen “in the real world” for large conjugative plasmids with e.g., addiction systems

ORIGINAL COMMENT: “duplication state” needs rewording.

NEW COMMENT: Now Line 99 – I still think that “a duplication state” is an odd and unclear description for plasmids. If “as multiple copies” is meant, this should be stated, but see comment above.

2) Lines 65-71

NEW COMMENT: Lines 65-71 - I think that beta-lactams would still be considered the most important class of antibiotics, and use of beta-lactamase inhibitors is not new (clavulanic acid etc.), although there are new types such as avibactam, so I would suggest some rephrasing here.

Also consider whether refs 4, 5, 6, and 7 are all really necessary here.

3) blaCTX-M-9 group variants, mutations and phenotypes, Lines 109-115.

ORIGINAL COMMENT: blaCTX-M-65 and variants, Lines 100-2,129-136

Lines 100-2 - the information about blaCTX-M-65 could be expanded, explaining that blaCTX-M-65 belongs to the blaCTX-M-9 group but does not have a mutation resulting in Asp240Gly usually associated with CAZ resistance (as found in e.g., in blaCTX-M-27).

RESPONSE: ...“blaCTX-M-65 belongs to the blaCTX-M-9 group, but unlike typical members of this group (such as blaCTX-M-27), it does not carry the Asp240Gly substitution that enhances resistance to CAZ. Instead, blaCTX-M-65 exhibits strong hydrolytic activity against CTX, rather than CAZ”

NEW COMMENT: Now Lines 100-2,129-136.

The ancestral CTX-M-9 group members (CTX-M-14, CTX-M-9) do not have the Asp240Gly mutation (nor do CTX-M-1 or blaCTX-M-3, equivalent ancestors in the CTX-M-1 group). The “CTX” in CTX-M was used to indicate resistance to CTX, with mutant versions that that confer CAZ resistance arising later, so variants with the Asp240Gly mutation aren’t really “typical”. Also note that CTX-M-65 is a variant of CTX-M-14.

Lines 109-10 - “originally” can be deleted. From the title Refs 26 and 27 cited here don’t seem to be specifically about CTX-M-65?

ORIGINAL COMMENT: Lines 171-7 – are there other studies reporting on the effects of these mutations?

RESPONSE: The dual mutations at positions 130 and 167 in blaCTX-M-65 has not yet been reported. However, the effects of single mutation at each of these sites on CTX-M enzymatic hydroactivity have been documented in several studies. For example, in *Antimicrobial Agents and Chemotherapy* (2003; 47(9):2958–2961), the Ser130Gly mutation was shown to reduce the hydrolytic activity of CTX-M enzymes against cefotaxime while increasing resistance to inhibitors including sulbactam, clavulanic acid, and tazobactam. In *PLoS Pathogens* (2010; 6(1):e1000735), the Pro167Ser mutation was reported to decrease the hydrolytic activity of group 1 CTX-M enzymes against cefotaxime. By contrast, in *Antimicrobial Agents and Chemotherapy* (2006; 50(2):731–738), the Pro167Ser mutation was shown to enhance the hydrolytic activity of group 9 CTX-M enzymes against ceftazidime.

NEW COMMENT: Now Lines 100-2, 129-136. This information should be included in the manuscript and the relevant papers cited, to make it clear that these are not novel mutations and their effects have been described previously, even if they were first found in a new combination here. Ref 14 describing CTX-M-14 also discusses the effects of the P167S mutation.

4) Lines 125-163 - example of suggested reorganisation/simplification

Sections could be combined to reduce repetition and condense. Maybe separate section on enzyme kinetics?

Line 139 - this wording doesn’t make sense – the gene that you are attempting to clone is not blaCTX-M-65, but the mutant from ICU-3. Maybe something like “We attempted to clone blaCTX-M-249 from ICU-3 into pCR2.1, but this resulted in colonies with different blaCTX-M genes. Most had blaCTX-M-65 or blaCTX-M-249, but two other variants, blaCTX-M-247 and blaCTX-M-248 were also present in a few.

Lines 159-163 – it is the clones encoding the CTX-M enzyme that are susceptible/resistant.

5) Lines 133-6 blaKPC-157

PMID: 40643750 (published July 2025) describes KPC-157 having the same phenotype as KPC-2. Referring to this paper would allow the text here to be simplified and would back up the result obtained here. Suggest something like “By testing the resistance conferred by blaKPC-57 cloned into pCR2.1 we found that this change did not cause an increase in CAZ resistance, as recently demonstrated (Ref)” (the carbapenem resistance is not really relevant to this manuscript).

6) Plasmid sequence differences, Lines 176-204

Line 182 - My alignment of the sequences of pICU-3 and pICU-18 provided/in GenBank shows that the two changes in addition to the ones in blaKPC and the nearby blaCTX-M gene are in adjacent bases in one copy of IS26. These changes should be carefully checked by mapping Nanopore reads, and if correct, properly described. See also PMID: 34015086. Some rewording/reorganising might also help to make things clearer. For example, maybe first say that long read sequencing showed that ICU-18 carries blaKPC-2 and two copies of blaCTX-M-65, one near blaKPC-2 and one about 54 kb away, on a single plasmid named pICU-18. Then say that ICU-3 has an almost identical plasmid, named pICU-3, with blaKPC-157, one of the four blaCTX-M-65 variants close by and blaCTX-M-65 at the other location, Note that “Backbone” in reference to a plasmid is usually taken to mean the region encoding plasmid functions, into which regions containing resistance genes are inserted.

7) Line 187 – plasmid copy number

ORIGINAL COMMENT: Lines 188-9 – are there ANY clues in the sequence about what is responsible for this increase in plasmid copy number? This should be looked at.

RESPONSE: ... the molecular mechanism underlying the increase in plasmid copy number remains unclear. We performed long-read sequencing and assembly under CZA pressure and compared the results with drug-free conditions, focusing on the plasmid replicon and replication-associated regulatory genes, but detected no genetic changes that could account for the elevated plasmid copy number.

NEW COMMENT: Now Lines 187 - would this analysis have identified any heterogeneity in these regions of the plasmids? If not, maybe it would be worth looking at this? I also think it would be worth mentioning in the manuscript itself that there appear to be no changes in the plasmid that would be expected to increase copy number, if this is the case.

8) ORIGINAL COMMENT: Line 212 ... I also would suggest looking at whether this TU has been seen before and at papers on the plasmid contexts of blaCTX-M-65 (e.g. PMID: 22984205).

RESPONSE: reports specifically involving blaCTX-M-65 TUs remain limited. In our previous work, we identified an IS26-mediated tandem multicopy structure of blaCTX-M-65 in an *E. coli* isolate (*J Glob Antimicrob Resist*, 2023; 35:202–209), though its structure differed from that observed in the present study. In addition, another study from China (*J Antimicrob Chemother*, 2013; 68(1):46–50) also reported a similar IS26-mediated tandem duplication unit of blaCTX-M-65 in *E. coli*.

NEW COMMENT: Please consider referring to the JAC paper.

9) ORIGINAL COMMENT: Lines 218-329 Models of heterogeneity

This part of the text gets hard to follow and could be simplified and condensed.

RESPONSE: ... we revised the model names and abbreviations: Model 4 is now referred to as Dual-Plasmid Duplicates (DPD), meaning heterogeneous blaCTX-M copies are duplicated on separate plasmids; Model 5 is now referred to as Single-Plasmid Duplicates (SPD), meaning heterogeneous blaCTX-M copies are duplicated on a single plasmid. The

difference between the initials “D” and “S” helps readers quickly distinguish dual-plasmid from single-plasmid.

NEW COMMENT: While I think that the revised model names are better, and agree that it is easier to tell the acronyms apart, to me “Dual plasmid duplicates” and “Single plasmid duplicates” are confusing descriptions. The important thing is that the two genes are different and where they are located. I would suggest considering these names further – maybe 1P for two different variants in the same plasmid and 2P for the two different variants on different plasmids, and perhaps 2C for two different variants in two cells?

10) ORIGINAL COMMENT: Line 351-57 – which blaCTX-M group or did you look at multiple groups? Major sequence differences between members of different blaCTX-M groups (Group 1, Group 9 etc) are not due to mutations, but because the source genes are from different *Kluyvera* species (there are also hybrids between Group 1 and Group 9 genes). Minor changes between blaCTX-M genes within each group are probably explained by mutations that have occurred since capture.

RESPONSE: ...this part of our analysis was performed using raw genomic data to assess the heterogeneity status of a specific β -lactamase gene in each strain across a large dataset. In practice, we first identify the target gene through homology search (BLAST), and then determine heterogeneity by analyzing sequence differences among all raw reads contributing to the assembly. Therefore, whether the reference used for BLAST belongs to a Group 1 or Group 9 blaCTX-M gene does not affect our ability to first locate a specific blaCTX-M genotype within the genome. For our analysis of blaCTX-M, we used blaCTX-M-65 as the reference sequence. As long as a blaCTX-M gene is present in the genome (even one from a divergent group such as Group 1), its presence can still be identified. We then analyze whether heterogeneity exists among all reads assembled into that gene.

NEW COMMENT: I still think that giving a combined % here for all blaCTX-M genes is not ideal. It might be interesting to focus on e.g. blaCTX-1 group and 9 groups (the most common) and see whether the level of heterogeneity differs (although hybrids might interfere with this).

11) General comments to streamline text.

“p” indicates a plasmid, so using “plasmid pXXX” or “the pXXX plasmid” is not necessary.

Cloning something in implies a vector. e.g. Lines 134-9 - “into pCR2.1 to verify” is sufficient,

“The CTX-M-65 beta lactamase” etc can just be “CTX-M-65”. Also remove “the” before CTX-M-65 on Lines 168, 169 etc - not needed.

Delete things like “Results showed that” – they just add unnecessary words.

Remove unnecessary descriptions of Methods in Results

12) Examples where wording would improve clarity/accuracy etc

Title – while I appreciate the reason for wanting to include reference to a single plasmid, I think that the title is now too general – only a single gene type has been looked at and only in a single isolate (and all experiments models were in *E. coli*).

Line 45 – “elevated” in relation to what? This may not be the best word here.

Lines 49 – “heterogeneity duplicated within a single plasmid” is confusing.

Lines 52-3 – “heterogeneity is widespread among various beta-lactamase genes in clinical isolates” is not clear – maybe something like “many clinical isolates have multiple variants of a beta-lactamase gene” if this is what is meant.

Line 55- “multidrug resistance” may not be the best phrase here – this is commonly used to mean resistance to multiple classes of antibiotics, while only beta-lactams are being described here.

Lines 61-3 – “Humans have” might be better.

Line 74 – suggest “particularly those caused by”.

Lines 77-9 – sentence gets hard to follow. More strains than what?

Line 80 – I don’t think that “Beyond KPC carbapenemases” or “occurring” are needed here and “within” could be “in”.

Lines 81-3 – if CMY-172 and CTX-M-219 give CZA resistance than this needs to be clearly stated.

Lines 85- - wording could be improved.

Line 90 – this could just say “However, gene duplications can alleviate trade-offs because”.

Line 116 – suggest “plasmid-borne”.

Line 118 – “location”, not “localization”.

Lines 130-131 – this could just be “(Table 1). Both isolates belong...”

Line 155 - as noted in previous comments, the variants do not have MICs, the antibiotics do.

Line 168 - “For enzyme inhibition” is not needed here.

Lines 170 – “Thus” could be used instead of “These Results demonstrate” “point mutations” doesn’t need a dash, “enzyme’s” can be deleted

Line 193 – which “drug”?

Line 198 – “the blaCTX-M gene near blaKPC-157”

Lines 199-200 – the IS26, not the TU, that “bracket” the blaCTX-M gene, which is part of the TU and it is a single TU that is amplified.

Line 207 – “In this study” is not needed and “of plasmid-harboring blaCTX-M” does not make sense – Suggest “of plasmids carrying blaCTX-M” or “plasmid-borne blaCTX-M”.

Lines 208-9 - Delete “To facilitate observation” and “corresponding” has no meaning in this context.

Line 231 – maybe “The existence of multiple variants of the same gene that confer distinct resistance profiles enables bacteria to withstand the combined effects of multiple drugs”

Line 239 – “are encoded”

Lines 253 etc – “anchored” is not the right word here. “When both copies of blaCTX are on the same plasmid” would be simpler and more accurate, also Line 415.

Line 259 – change “supplemented” to “modified”

Line 270 - "surged" is not scientific.
Line 281 - "of" here does not make sense.
Line 308 - "gained superiority" is unscientific.
Lines 314, 317, 318 - "raw genomic data"
Line 316 - "heterogeneity of" and "such as" is not clear - just list all gene types examined.
Line 320-1 - "by mapping raw reads" if this is what is meant?
Lines 321-2 - all this needs to say is "The estimated copy number distributions of the genes examined" - it has already been made clear that they are bla genes and from K. pneumoniae.
Lines 324-5 - blaSHV is expected to be in the chromosome of all K. pneumoniae and it may or may not additionally be on plasmids - this needs to be properly explained.
Line 325, 331 - sentences shouldn't start with "And"
Line 488-90 - which flow cells/chemistry was used. The assembly is not combined, the reads are combined for the assembly?
Lines 494-5 - needs rewording.
Line 507 - "Colonies were selected", "Positive" is not needed.
Line 510 - "...sites, cloned.."
Line 528 - what is meant by "the low-affinity substrate"?
Line 557 - a psiB gene is found on many plasmids, it is not specific to pICU-3 and there seem to be two genes annotated as psiB (psiB1 and psiB2) in the pICU-3 sequence in PX405700, but only ~57% identical). The psiB primers and probe in Table S3 match psiB, but this needs to be explained more accurately.
Lines 580-581 - "12 dilutions covering", "16 dilutions covering"?
Line 589 - "grown" not "growth".
Line 602 - it's the plasmids that have the fitness costs, not the strains?
Lines 704-5 - suggest "The pICU-3 and pICU-18 sequences are available under GenBank accession numbers PX405700 and pX405701, respectively".

13) Discussion

The Discussion is still long, and some parts need rewording to improve accuracy. Also, I think that it is important not to generalise results beyond at least bla genes, maybe not beyond blaCTX-M. See comments on Fig. 7 below.
Lines 337-45 - this information still seems like it repeats and/or should be in the Introduction.
Line 350 - what is meant by "ESBL mutations" here? In which gene(s)? CMY-172 and CTX-M-219 are already mentioned in the Introduction.
Line 351 - needs rewording - a clonal strain is not a variant, either "trade-offs" or "a trade-off phenomenon" - see previous comments. AST is done on isolates/strains, enzyme kinetics uses proteins, neither is done with genes.
Line 354 - change "explored" - see previous comments.
Line 357, 360, 421 - genes are not encoded on plasmids, they are e.g. carried by plasmids, found on plasmids - see previous comments.
Line 358 - "In the subsequent study" does not make sense here.
Line 362 - the "locus encoding CTX-M-249" is the blaCTX-M-249 gene - this term can be used. A gene can't undergo an "increase in plasmid copy number".
Line 367 - it is really the blaCTX-M-65 gene that is widespread.
Line 370 - "the novel" not needed.
Lines 371-32 - the enzymes hydrolyse, but it is the strains encoding them that shows resistance.
Line 377 - it is multicopy of essentially a single gene (based on blaCTX-M-65) not multiple genes
Lines 377-8 - what "functioned as a bet hedging strategy"?
Lines 382-4 - "simultaneously carrying blaTEM-1 and blaTEM-12" and state differences in sequences and phenotypes, "trade-off losses", not "trade-offs losses".
Line 385 - change "brought by" to "due to"
Line 386 - "in a clinical strain" - only one strain studied here showed this effect.
Line 393 - no dash needed in "single antibiotic"
Line 398 - "genes split between" doesn't really make sense here - "genes on" would be fine.
Line 415 etc - "anchored in" is not the best way to say this. "on".
Lines 416-7 - unclear - do you mean that the first step was for the plasmid to acquire a single copy of blaCTX-M-65?
Lines 441-2 - the wording here could still be made clearer.

14) ORIGINAL COMMENT: Lines 471-5 - is this really true for long-read if multiple copies of the gene are in different genetic locations, as for copies of blaCTX-M-65 (or variants) both in pICU-3 here?

RESPONSE: ...when multiple copies of a gene are located in different genetic contexts—as with blaCTX-M-65 (or its variants) present in both regions of pICU-3—the relatively lower base accuracy of long-read sequencing makes it difficult to resolve which specific variant resides in which repeat region.

NEW COMMENT: I don't really agree with this. From my own use of read mapping with ONT reads (R10), it can be pretty easy to distinguish real nt differences (occur in almost all ONT reads) from errors (each occurs in only one or two sequences).

15) References

ORIGINAL COMMENT: Ref 8 needs page numbers or equivalent. Something seems to be wrong with page numbers in Ref 49. Journal titles need to be correctly abbreviated. Ref 4, 12 etc., remove Title Case. Species and gene names etc. need to be correctly formatted.

RESPONSE: and carefully checked all reference entries, correcting errors and inconsistencies in formatting. Species and

gene names have also been italicized accordingly.

NEW COMMENT: There are still a few problems.

Ref 4 has no page numbers or equivalent

Refs 4,11 – check if in vitro/vivo are in italics in these papers.

Refs 8, 23, 28, 55– check if beta symbols are used in these papers.

Refs 13,15, 53, 54, 55 – page number equivalents look odd.

Ref 19 – probably should be “Typhimurium”, not in italics and *rrn* in italics.

Ref 37 – “26” of IS26 should probably be in italics.

Ref 61 – “a tool”

16) Figures

The figures and their organisation/distribution between the main manuscript and Supplementary is improved, but more consistency in colours etc across figures/parts of figures would improve clarity. For example, in the main manuscript, the bar chart at the bottom of Fig. 1 Part II uses different colours for blaCTX-M-65 and the three variants from the diagrams in Part I. What appear to be the same graphs in Fig. 1 Part I and Fig. 2 b and c use different colours again, as do Fig. 3 and Fig. 7. In Fig. 7 (a) blue is used for blaCTX-M-65 and a dark red colour is used for blaCTX-M-249, but in (b) the ancestral allele (blaCTX-M-65 in part a) is in a dark red and the evolved allele (blaCTX-M-249 in part a) is in blue. Also, in Fig. 1 Part III the colours of the blaCTX-M genes in the models are hard to see.

Maybe start with green for GFP, red for mCherry, orange for both and blue for mTagBFP, as these are fixed, then find different, easily distinguishable colours to show blaCTX-M-65 and the three variants and use these consistently in diagrams, graphs etc consistently throughout, where possible.

Figure legends often need some rewriting to improve accuracy/calidity

Fig. 1 legend

Lines 881-2 – is ICU-18 really a “clone”?

Line 883 – cloning also identified the different variants

Line 884 – “Plasmid location”

Lines 886-7 - “different multicopy heterogeneity states”, “heterogeneity stability” can be described more clearly with less jargon.

Lines 889-90 – “multiple minor variants of various bla genes in a single isolate from SRA data”

Fig. 2 legend

Line 892 – “diagram” not needed”.

Line 893 – pICU-18 is not shown in the diagram.

Line 894 – “elements” not needed – they are IS. The IS26 form the TUs, and all of the tandem TUs could be shown on the diagram – see previous comment.

Line 895 – “ONT read”? Relative to what?

I think that it would be helpful in part c to make it clear that these copy numbers are due to a combination of things i.e., every plasmid would have a copy of blaCTX-M-65 (non-amplified copy flanked by inverted IS26) and the % of blaCTX-M-249 is a combination of both mutation and amplification, at least under some conditions – see previous comment relating to this.

Fig. 3

p15A_ori could be removed from the diagrams at the top to simplify them (it's in all of the plasmids) and note that pACYCDuet1 has this replicon in Methods (Lines 563-71).

Legend

Lines 900-1 – title could be improved.

Text can be simplified and condensed – see comments above and avoid explaining things that are obvious from the figure. e.g. Lines 901-2 “Model 1 (strain 65G) has p65G (blaCTX-M-65-GFP) only, Model 2 (strain 249R) has p249R (blaCTX-M-249-mCherry), Model 3 is a 1:1 mixture of 65G and 249R (population level heterogeneity, PLH). Model 4 has p65G and p249R in the same cell (2P) and Model 5 has p65G-249R, (blaCTX-M-65-GFP+blaCTX-M-249-mCherry) on a single plasmid (1P), using alternative model names suggested above.

Line 970 – “results for the five models.

Line 980 – “co-selection by”

Lines 909-10 “represent”, not “represented”

Line 911-2 – “in each cell represent the” and “values of bacterial... concentration” are not needed.

Fig. 4

Part (a) - suggest stating % of what on graph, rather than in legend.

Part (B) – not clear why a dotted line is used for “all” but a continuous line for CTX resistant. Might be better if all lines are dotted.

Legend

Title, Lines 914-5 – probably doesn't need “during serial passages”.

Lines 916-7 – probably not needed - see above (but should be “is plotted”)

Lines 917-9, 922-3 – legends should describe what is in the figure, not repeat Results.

Line 920 – suggest “by plating serial dilutions on CTX or CAZ and counting CFU”, Lines 920-22 are then not needed.

Fig. 5

Suggest stating relative % of what on graph, rather than in legend. Maybe colours used could match the fluorescence colours? Also, 65G is used elsewhere as a strain name, but here “65G” is really the 2 plasmid strain that has lost p249R, and conversely “249R” is really the 2 plasmid strain that has lost p65G. Also “SPD” is used here, not SPD-B, as in the legend. Some simple explanatory diagrams might be useful here – showing original DPD, SPD-B and loss of one plasmid or another to give “65G” and “249R”.

Legend

Line 926 – suggest “(additional blue fluorescence)”.

Fig. 6

See comments above on blaCTX-M genes.

Line 934 – delete “such as”, “-like” should not be subscript.

Line 935 – is this really mapping to a reference not “alignment”?

Lines 937-8 – it’s not really the strains that are “(non)heterogeneous”, they have multiple variants of the gene of interest.

Fig. 7

This scheme may be generalisable to e.g. other blaCTX-M gene families (e.g. group 1) or other bla genes and beta-lactam antibiotics (where one mutation can have a drastic effect on phenotype) but might not apply to other types of antibiotic resistance genes.

See comments above on colours used for blaCTX-M genes. What do pink and blue parts signify?

Suggest flipping the direction of the curved arrows on the two left hand diagrams, to make it clearer that these parts are “moving away from” the central diagram.

Legend

Line 942 -suggest “blaCTX-M gene heterogeneity in ICU-3” if that is what is meant?

Line 945 – suggest “single mutation or two mutations”

Line 946 - suggest “...pressure transient tandem copies of the mutated gene may be generated”

Line 947 – suggest “of heterogeneity in blaCTX-M genes carried on the same plasmid”

Line 948-53 – I would suggest trying to simplify the text, e.g. “non-heterogeneity” = single gene variant, “plasmid-level multi-copy heterogeneity” = both gene variants on the same plasmid.

17) Supplementary

Table S1 - column heading says “inhabitants”, should be “inhibitors”.

Table S2, 3rd column heading use “allele(s)”?

Table S3 – “probes, not “Probes” in heading, suggest making it clear where CTX-M primers are only relevant for blaCTX-M group 9 genes.

Table S4 – shorten and reword titles – suggest “BioProjects containing Illumina sequence reads for *K. pneumoniae* isolated from humans” etc.

Supplementary figures – check for accuracy and condense if possible - see comments on legends for main figures.

The images or other third party material in this Peer Review File are included in the article’s Creative Commons license, unless indicated otherwise in a credit line to the material. If material is not included in the article’s Creative Commons license and your intended use is not permitted by statutory regulation or exceeds the permitted use, you will need to obtain permission directly from the copyright holder.

Response to Reviewer #1 (Remarks to the Author):

Weng et al presents one model case of paralogous gene evolution in one genome by focusing on a plasmid detected in a hospital. By conducting additional survey in publicly available raw reads, they found that the paralogous (beta-lactamase) gene evolution in one plasmid is common. Furthermore, the authors characterized the nature of four CTX-M variants in vitro. These are solid and important data. The statement in the abstract is valid. The manuscript is clearly written.

However, I am not sure about the significance of discussion on the stability of gene heterogeneity, comprising a large part of this study. The fitness advantage of the “dual-copy genes on one plasmid” state (DCG model) over the “dual-copy plasmids” state (DCP model) under drug-switching condition (Fig 5eg) is immediately obvious as far as plasmid loss can occur, unless huge fitness cost arises from the DCG state. The points of discussion should be “when does DCP or mix model win?”. Therefore, to discuss the conditions where DCP/PCG/mix state wins, I think that the authors need to implement mathematical model for the serial passage or competition experiments considering the fitness advantage of plasmid-bearing cells and plasmid loss rate. To address the above problem, ideally, all experiments should be designed using a low copy number and stable plasmid such as mini-F to mimic the condition of the real situation.

Response: We sincerely thank the reviewers for their valuable suggestions, which have provided us with important insights to enhance the scientific depth of our study.

In response to the reviewers' recommendation to redesign the experiments using low-copy-number and stable plasmids such as mini-F, we tested the low-copy plasmid pCC1FOS (Genome Res, 2002;12(9):1434-44), whose copy number and stability are comparable to mini-F. However, our experiments revealed that pCC1FOS has several limitations: (1) its extremely low copy number (approximately 1–2) likely led to insufficient fluorescent protein expression when enhanced green fluorescent protein (eGFP) or red fluorescent protein (mCherry) gene was fused downstream of the resistance genes *bla*_{CTX-M-65} or *bla*_{CTX-M-249}, resulting in fluorescence signals below the detection threshold (Chem Rev, 2002;102(3):759-81) and

rendering subsequent dynamic tracking experiments infeasible. (2) it could not support the construction of the DPD model (the DCP model described in our previous manuscript, which carries both p65G and p249R plasmids), as DPD strains require the simultaneous maintenance of at least two plasmids, while pCC1FOS's strict copy-number control prevented dual-plasmid coexistence; (3) its copy number was too low to realistically mimic clinical strains, which we have confirmed to possess plasmid copy numbers greater than two, often increasing to over a dozen under antibiotic pressure. Based on these factors, we concluded that pCC1FOS is not suitable for achieving accurate simulation and stability, whereas the plasmid pACYCDuet-1 used in our study was confirmed by digital qPCR to have a copy number of approximately 13, placing it in the medium-to-low range, which better reflects clinical conditions and facilitates fluorescent labeling for visualization and validation.

Following the reviewers' suggestion, we developed an ordinary differential equation (ODE) model using estimated parameters from the pACYCDuet-1 system to simulate the dynamics of plasmid-free (PF), 65G, 249R, DPD, and SPD (the DCG model described in our previous manuscript, which carries a single plasmid p65G-249R) strain populations under gradients of the antibiotics CTX and CZA. This model, based on Monod growth kinetics and incorporating resource limitation, antibiotic killing effects, enzymatic hydrolysis efficiency, and plasmid loss rates, produced simulation results for serial passage (Fig. S6 a-b) and competition experiments (Fig. S6 c-d) that were consistent with our earlier experimental findings, supporting its validity. In addition, we applied this model to explore the dominance boundaries between DPD and SPD, which helps address the question of "when does DPD or SPD model win?". The results showed that when CTX concentrations were below 39.31 mg/L, the growth fitness advantage of DPD exceeded that of SPD. When CTX was between 39.31 and 367.01 mg/L, the resistance advantage of SPD outweighed the growth advantage of DPD, and SPD dominated. At CTX concentrations above 367.01 mg/L, one of the plasmid-loss derivatives of DPD, 65G, became dominant. Similarly, for CZA, when concentrations were below 1.34 mg/L, DPD had a growth advantage over SPD. When CZA was between 1.34 and 127.91 mg/L, SPD's resistance advantage surpassed the growth advantage of DPD, allowing SPD to dominate. At

concentrations above 127.91 mg/L, another plasmid-loss derivative of DPD, 249R, prevailed.

Although the authors mentioned the cost of these cell types in the absence of antibiotics (Fig S1), competition experiments and stability test were performed in the presence of antibiotics where plasmid-free cells are immediately killed. Therefore, the cost information is not really linked to experimental data.

Response: We thank the reviewer and have followed this recommendation. Indeed, there was an insufficient correlation between the data on fitness costs under antibiotic-free conditions and the results from stability and competition experiments. To address this, we performed additional growth curve experiments to systematically evaluate the growth fitness of PF (plasmid-free, i.e., DH5 α), 65G, 249R, DPD, and SPD strains under antibiotic-free conditions (0 mg/L) as well as across a range of antibiotic concentrations. Specifically, we tested CZA at 0, 1, 4, 8, and 32 mg/L, and CTX at 0, 4, 16, 64, and 256 mg/L, which together covered the antibiotic concentration ranges used in our stability and competition assays. We measured OD₆₀₀ over time, calculated the area under the OD₆₀₀-time curve (AUC), and quantified growth performance using the relative growth inhibition rate [$100 \times (1 - \text{AUC}/\text{AUC}_0)$, where AUC₀ represents the AUC for each strain at 0 mg/L], thereby providing an integrated measure of growth fitness across different levels of antibiotic pressure (Fig. S5).

The most parts of figure are used to explain the experimental set-up. Figure information should be more condensed. For example, Figure 4 can be condensed to only two line plots: one from flow-cyto (Fig4c), one from CFU (Fig4de). Figure 5 can be condensed to only Fig5eg.

Response: We sincerely appreciate your suggestion to make the figures more concise, and we agree that adjusting Figures 4 and 5 can improve the focus of data presentation. Since several other reviewers also recommended modifications or adjustments to the figures, we carefully considered all feedback and followed the principle of highlighting only the most important and essential information. The specific adjustments are as follows: We reconstructed Figure 1 as a workflow diagram and moved the original panels 1b - d to Supplementary Figure 1. From the

original Figure 3, we retained panels b and d, while moving panels a and c to Supplementary Figure 2. In Figure 4, panels a, b, and d were moved to Supplementary Figure 3, panel c was retained, and panel e was changed from a bar chart to a line chart to improve clarity. From the original Figure 5, panels a, b, c, d, and f were moved to Supplementary Figure 4, leaving only panels e and g, which were revised into the current Figure 5.

It appears that to select for specific (mix model) cell types, a high concentration of CTX, 256 ug/ml, was used in the experiment. The concentration should be in clinically achievable range (< or = 100 ug/ml).

Response: We sincerely thank you for pointing out that the CTX concentration used in our experiments (256 $\mu\text{g}/\text{mL}$) exceeds the clinically achievable range ($\leq 100 \mu\text{g}/\text{mL}$), and we agree that adjusting to clinically relevant concentrations would strengthen the applicability of our study. The original in vitro experiments employed 256 $\mu\text{g}/\text{mL}$ primarily to rapidly screen for dominant patterns and to investigate bacterial competition dynamics under extreme antibiotic pressure. In response to your suggestion, we applied an ordinary differential equation (ODE) model to simulate population dynamics across a broader range of CTX concentrations (0.125–4096 $\mu\text{g}/\text{mL}$) and confirmed that the competitive dynamics observed at clinically relevant concentrations ($\leq 100 \mu\text{g}/\text{mL}$) were consistent, thereby providing a more detailed analysis (Fig. S6).

MINOR POINTS:

FIG1a. Gray area does not precisely connect the circular map and gene map of tandemly amplified loci.

Response: We sincerely thank you for pointing out the issue regarding the connection of the gray-shaded regions in the figure. We think you may be referring to Fig. 2a (rather than Fig. 1a), as Fig. 2a presents the circular plasmid map and the gene diagram of its tandem amplification sites, with gray regions used to indicate the corresponding relationships. In response to your comment, we have revised Fig. 2a and optimized the rendering of the gray-

shaded areas to ensure precise connections between the circular map and the gene diagram of the tandem amplification sites.

FIG3b. I think that gene map on plasmids is unnecessary as they are redundant with panel a.

Response: Thank you for your suggestion. Figure 3b, which illustrates bacterial strains carrying different plasmids, corresponds directly to the five models we constructed, as reflected in the checkerboard susceptibility results shown in Figure 3d. In line with your advice and other reviewers' comments on simplifying figures, we have revised Figure 3 by retaining panels 3b and 3d into the new Figure 3. Panels 3a and 3c have been moved to Supplementary Figure S2a and S2b to remove redundancy and improve clarity.

FIG4. Plasmid stability data measured by flow cytometer (a-c?) and by CFU count (d, e).

c. Please clarify in legend whether this graph is based on CFU count or summary of flow cytometer counts. d-e: d and e show the same information. I suggest the author should show the data of proportion of the resistance gene-carrying cells by line plot like panel c.

Response: We greatly appreciate your suggestion, which is highly valuable for improving the clarity of the figures and the effectiveness of data presentation. We have revised Figure 4 as follows: Figure 4c, which is based on flow cytometry data and shows the proportion of cells carrying resistance genes, has been updated with a legend explicitly stating that the data were obtained from flow cytometry counts, and the figure legend has been revised accordingly to ensure accuracy. Since Figures 4d and 4e conveyed overlapping information, we have moved Figure 4d to Supplementary Figure 3c, and converted Figure 4e from a bar chart to a line graph (now Figure 4b) to enhance clarity.

Fig5. d label. Please add ug/ml after antibiotic concentrations because it is not immediately clear whether those are symbol of CTX-M variants.

Response: Many thanks. We have added the unit "mg/L" to all antibiotic concentrations (e.g., CTX and CZA) in the labels of Figure 5d and 5f to clearly indicate that these values represent

antibiotic concentrations rather than CTX-M variants. According to the reviewer's suggestions, we have moved these two panels to the current Supplementary Figure 4.

Reviewer #2 (Remarks to the Author):

Regarding the manuscript "The heterogenetic multicopy of blaCTX-M variants on one plasmid optimizes the evolution adaptability in the clinically isolated *Klebsiella pneumoniae*" by Weng et al.

This manuscript presents a novel and well-supported study on how multicopy heterogeneity of blaCTX-M variants on a single plasmid contributes to the adaptability and stability of antibiotic resistance in *Klebsiella pneumoniae*.

Overall, the manuscript is well-written, and the study is highly relevant, novel, and scientifically rigorous, with important clinical relevance. The combination of whole-genome sequencing, digital PCR, amplicon sequencing, and in vitro modeling provides strong experimental support for the proposed mechanism.

However, the comprehensiveness of the study makes it somewhat difficult to follow, and some sections appear misplaced, affecting the clarity and flow of the manuscript. Below, I have outlined major and minor comments that would benefit the manuscript.

Response: We sincerely appreciate the reviewers' valuable suggestions. We have thoroughly revised the manuscript, reconstructed the workflow diagram (Fig. 1), and reorganized the presentation of the results accordingly. Substantial language improvements were also made in the Introduction, Discussion, and Methods sections. In addition, the composition and order of the figures were adjusted to make them clearer and more concise. We hope these revisions have further strengthened and improved the study.

Major Comments

1. Structural Issues & Misplaced Sections:

Several sections appear misplaced within the manuscript.

I recommend restructuring the introduction and results sections to improve clarity and logical

flow. Some specific examples:

- Lines 99-111: This section describes the findings of the study, which should not be placed in the introduction. The introduction should focus on background information and knowledge gaps, leading to the study's aim and objectives.

Response: Thank you for this valuable suggestion. We have restructured the last paragraph of the Introduction to place more emphasis on the background information and then clearly present the study's aim and objectives.

- Lines 120-122: This section describes the aim of the study, which should be included in the introduction, not the results section.

Response: We moved these sentences, similar to the aim of the study, to the last paragraph of the Introduction. Here, we condensed and revised the following sentence to make the meaning complete and concise.

- Lines 331-337: Discussion elements appear within the results section. These should be moved to the discussion section.

Response: Thank you for your suggestion. We have moved these sentences to the appropriate section of the Discussion. In fact, we reorganized the description of the results regarding the universality of resistance gene heterogeneity, presenting only the results and keeping the sentences concise.

- Lines 359-361: Similarly, discussion/conclusion elements are found in the results section and should be relocated.

Response: Similar to the above point, we reorganized this part of the results, and all discussion-like statements, including the one you mentioned, have been moved to the Discussion section.

I suggest a thorough review of the manuscript structure to ensure that each section follows a logical order.

Response: Thank you for your suggestion. We have made substantial revisions to the manuscript, making the description of the results more concise and clear, and adjusting the logical structure of different sections to improve coherence.

2. Results Section is Long & Difficult to Follow

The results section is long, which is expected with the comprehensiveness of the study, making it a little challenging to follow. Consider referring to the steps in the flow-diagram (Figure 1a) at relevant points in the results. Possibly expand the flow-diagram to further guide the reader through the workflow and experimental design. This small change could significantly improve readability and clarity.

Response: We sincerely thank you for pointing out the complexity of the Results section and for your valuable suggestion to improve the workflow diagram. To enhance readability, we expanded and refined Fig. 1 (originally Fig. 1a) by dividing it into three sections, thereby optimizing the workflow to more clearly guide readers through the research process. Specifically, the revised Fig. 1 now consists of three parts corresponding to the core components of our study. Part I: discovery of clinical CZA resistant strain and identification of the novel variant which mediated the resistance; Part II: confirmation of the profile of heterogeneous multicopy *bla*_{CTX-M} variants on plasmids; Part III: exploring the advantages of single-plasmid duplicates in experimental and mathematical models. Each part incorporates detailed research content and uses arrows and annotations to visually present the logical flow. In the Results section, we now refer to the relevant parts of Fig. 1 in the corresponding paragraphs.

3. Clarification of Nucleotide and Amino Acid Mutations

In Lines 123-124, the authors mention the noteworthy changes, including: “A397G, C508T in the bla....” Then, in Lines 131-133, the authors mention the corresponding amino acid changes: “Ser130Gly and Pro167Ser”.

However, based on the alignment in Figure 1b, these mutations appear to correspond to amino acid positions 133 and 170, not 130 and 167.

Clarification needed: Are the mutation positions correctly assigned? If different numbering conventions were used, this should be explicitly stated in the Methods section.

Response: Thank you for drawing attention to the description in lines 123–124 (nucleotide substitutions A397G and C508T in *bla*_{CTX-M-249}) and lines 131–133 (amino acid substitutions Ser130Gly and Pro167Ser). To clarify the correspondence between nomenclature and numbering, we confirm that A397G and C508T are based on nucleotide numbering of the *bla*_{CTX-M-249} gene, whereas the amino acid numbering Ser130Gly and Pro167Ser follows the ABL standard numbering scheme for Class A β -lactamases (Biochem J, 1991; 276:269-70). To improve clarity, we have added an explicit note about ABL numbering and reference 30 in the main text, and indicated the amino acid numbering in Supplementary Fig. 1a.

4. Statistical Analysis

Throughout the manuscript, results are stated as “significant”, but p-values are missing in multiple places.

Example: Line 192 mentions significance without statistical support.

Include p-values wherever statistical significance is claimed.

Response: Thank you for raising this question. We have thoroughly reviewed the entire manuscript and have systematically added the precise *p*-values, wherever statistical significance is claimed. This includes all data concerning: gene copy numbers, derived from digital qPCR; colony counting numbers, obtained through plate counting assays; fitness cost, calculated from growth curve analysis; percentage of dual-fluorescent cells under CTX or CZA pressure analyzed by flow cytometry. For the specific example cited by the reviewer (Line 192 in the original manuscript), we have revised the sentence to include the statistical support as follows: “Digital qPCR results showed that, compared with drug-free culture, exposure to 32 mg/L CZA increased *bla*_{CTX-M} copy number to >10 per chromosome equivalent ($p = 0.0030$), primarily driven by a rise in plasmid copy number (from ~2 to >7, $p = 0.0184$). At a higher 64 mg/L CZA, *bla*_{CTX-M} copy number also approached 10 ($p = 0.0081$), but without a marked plasmid copy number increase ($p = 0.8735$); instead, the number of *bla*_{CTX-M} copies per plasmid

rose to ~ 3 ($p = 0.0069$).” These comprehensive statistical details have been added throughout the manuscript (marked in the revised file).

5. Figures are Overloaded – Consider Moving Some to Supplementary Material

While only seven figures are presented, each figure contains a high number of panels.

I recommend to move redundant figures to supplementary materials to streamline the main text.

Recommended changes:

- Move Figure 1b and 1c to supplementary material.
- Move Figure 3c and 3d, but expand Figure 1b by integrating the findings for the five different models.
- Move Figure 4a, 4b, and 4d to the supplementary.
- Move entire Figure 5 to the supplementary.

This restructuring would help reduce figure complexity while keeping essential data accessible.

Response: We greatly appreciate your suggestion and agree that moving redundant subfigures to the supplementary materials can improve the focus of the main text. Taking into account both your recommendations and comments from other reviewers, we have revised the figures as follows: Figure 1: we expanded the original Figure 1a (now Figure 1), and the content of the original Figure 1d has been incorporated into Figure 1 (part I) as a grid map; the original Figures 1b, 1c, and 1d have been moved to the supplementary materials (now Figures S1a, S1b, and S1c). Figure 3: we retained Figures 3b and 3d and combined them into the new Figure 3, while Figures 3a and 3c were moved to Supplementary Figures S2a and S2b to eliminate redundancy and enhance clarity. Figure 4: we moved Figures 4a, 4b, and 4d to Supplementary Figures S3a, S3b, and S3c; the original Figure 4e was converted from a bar chart to a line graph (now Figure 4b) to improve readability; the current Figure 4 retains the original Figure 4c (now Figure 4a) and the revised Figure 4b. Figure 5: we moved Figures 5a–d and 5f to Supplementary Figures S4a–e, while retaining only the original Figure 5e and 5g (now Figure 5a and 5b), which represents the core data, in order to maintain conciseness in the main text.

Minor Comments

I have a few minor comments on wording and terminology

- Line 55: “The antibiotics developed by humans...” What about the naturally occurring ones?

Response: Thank you for pointing that out. We have revised the first sentence of the Introduction for clarity.

- Line 58: “Carbapenems were once considered the last-resort...” Carbapenems are still a last-resort antibiotic, so this should be reworded for accuracy.

Response: We agree with the point that carbapenems are still a last-resort antibiotic, and we have reworded this sentence accordingly.

- Line 457: “We postulate...” think it should be changed to “We presented” or “We constructed”

Response: We have rephrased the sentence in question and the subsequent paragraph describing Figure 7b to eliminate ambiguity. In addition, to improve the conciseness of the main text, we have moved this text to the legend of Figure 7.

- Lines 642-643: “alignment of genome raw reads in a broader range of *K. pneumoniae*.”
Reword for clarity. Perhaps: “Genome-wide raw read alignment of a broad range of *K. pneumoniae* isolates.”

Response: We have revised this sentence as suggested.

- Figure 2 – Circular Map: What software was used to generate the circular map in Figure 2?
This should be mentioned in the figure legend or methods section.

Response: The genetic structures of plasmids and their comparative circular maps were generated using Proksee. We have described it in the methods section with reference 52.

Reviewer #3 (Remarks to the Author):

This manuscript describes a study looking at blaCTM-65 and variants detected at different copy

numbers on the same plasmid in a clinical isolate of *Klebsiella pneumoniae*, with different gene combinations and copy numbers influencing resistance to ceftazidime, cefotaxime and ceftazidime+avibactam. Analysis of available sequences suggested that multiple copies of AMR gene variants in a single isolate may be more common than is usually recognized. While the results are potentially interesting and useful, there are some problems. Better sequence analysis, with reference to the available literature, is needed to improve accuracy in some of the descriptions. The detail about the results of experiments is wordy and uses unnecessarily complex terms, making it rather inaccessible and overwhelming. The text would benefit from language editing to make it clearer and easier to follow (and there is some odd/unscientific phrasing in places) and it could be condensed – some examples and suggestions are given below. There are many large complex figures and some thought about how these could be reduced would also be beneficial (e.g., parts showing detailed results of experiments could be made Supplementary, with summaries only in main text) – again I have given some suggestions below.

Response: We sincerely appreciate the reviewer's valuable suggestions. Following your advice, we have thoroughly revised the manuscript, reorganized its structure, ensured more concise and clear wording, adjusted the composition and order of the figures, and moved redundant figures to the supplementary materials. Below, we provide point-by-point responses to your comments, and we hope that our revisions have substantially improved the overall quality of the study.

Major points

1) blaCTX-M-65 and variants, Lines 100-2,129-136,

Lines 100-2 - the information about blaCTX-M-65 could be expanded, explaining that blaCTX-M-65 belongs to the blaCTX-M-9 group but does not have a mutation resulting in Asp240Gly usually associated with CAZ resistance (as found in e.g., in blaCTX-M-27).

Response: We have added some information about *bla*_{CTX-M-65} in the Introduction section. CTX-M-type β -lactamase genes comprise numerous variants that are classified into several groups based on sequence differences, with the two major groups being the *bla*_{CTX-M-1} group and the

*bla*_{CTX-M-9} group. *bla*_{CTX-M-65} belongs to the *bla*_{CTX-M-9} group, but unlike typical members of this group (such as *bla*_{CTX-M-27}), it does not carry the Asp240Gly substitution that enhances resistance to CAZ. Instead, *bla*_{CTX-M-65} exhibits strong hydrolytic activity against CTX, rather than CAZ.

Lines 131-46 could be condensed and simplified e.g., it is not really relevant when *bla*_{CTX-M-249} or the other variants were named.

Response: We substantially reorganized this paragraph to make the expression more concise and clearer. Instead of separately introducing the naming of *bla*_{CTX-M-65} and the other variants, we more concisely highlighted their heterogeneous genotypes (See the section “CTX-M-65 variant mediating CZA resistance”).

Line 131-3 – this is unclear. What was cloned that revealed the variants?

Response: Thank you for the suggestion. We realize that we did not explain the cloning experiment clearly. Here, cloning refers to the process in which the PCR product of the target gene from a clinical strain was ligated into the vector plasmid pCR2.1 and then transformed into a recipient strain, so that each transformant successfully obtaining the recombinant plasmid (also referred to as a colony) would carry only one variant of the PCR product. We used this method to confirm the presence of multiple variants. In cases where the target gene in the strain exhibits multicopy heterogeneity, the PCR product will consist of a mixture of different variants, and sequencing the gene fragment on the recombinant plasmid allows clear identification of which variant has been successfully cloned, since typically a single plasmid is introduced into only one clone. The proportions of different variants recovered from the clones can thus provide a rough estimate of the variant composition in the original PCR product. We have substantially reorganized this paragraph to express it more concisely and clearly.

Line 132 – are these amino acid positions numbered according to the Ambler scheme for class A beta-lactamases (PMID: 2039479). If so, this should be clearly stated.

Response: Yes, we confirm that the amino acid numbering Ser130Gly and Pro167Ser follows the ABL standard numbering scheme for Class A β -lactamases (Biochem J, 1991; 276:269-70). To improve clarity, we have added an explicit note about ABL numbering and reference 30 in the main text, and indicated the amino acid numbering in Supplementary Fig. 1a.

Line 134 – it is antibiotics that have MICs, not isolates.

Response: Thank you for the reminder. We have corrected this statement in the main text when described the MICs.

Line 136 – should be “mutations in blaCTX-M-65”

Response: We agree with your suggested revision; however, since this section has already been reorganized, we adopted alternative wording.

Line 138 – how were these “clonal isolates” obtained? And it should be “alleles” here.

Response: We have now provided a detailed description in the Methods section of how these clones were obtained. In this part, we reorganized the section and prefer to use variants to represent the different mutants of *bla*_{CTX-M-65}.

Lines 145-6 – do not make sense and needs rewording.

Response: Our intention here was to clarify that *bla*_{KPC-157} does not exhibit multicopy heterogeneity similar to *bla*_{CTX-M-65}. However, since the previous paragraph had already explained that *bla*_{KPC-157} clones cannot mediate CZA resistance, this statement was redundant and has been removed.

2) Plasmids, contexts/amplification of blaCTX-M-65 and variants, Lines 123-5, 179-216, 312-397

The assembled sequence of pICU-3 should be submitted to GenBank, along with the sequence of pICU-18, in addition to raw reads. It is possible to add comments to GenBank entries to

explain that e.g. the sequence submitted is a representative of several variants of the same plasmid that can have different numbers of copies of a region that includes blaCTX-M-65 but also the other variants.

Response: Thank you for your suggestion. All raw genomic data for strains ICU-3 and ICU-18, under different antibiotic pressures, generated from both Illumina and Nanopore platforms, have been deposited in the SRA under BioProject accession number PRJNA1194354. The plasmid assembly files (pICU-3 and pICU-18) are also available in GenBank under accession numbers PX405700 and PX405701. In addition, we have added comments to the GenBank entries to clarify that at one locus of *bla*_{CTX-M} carried by plasmid pICU-3, either *bla*_{CTX-M-65} or any of its three variants may be present.

The structure of pIUC-3 needs to be explained in the text and compared with pICU-3.

Response: We speculate that the reviewer intended to compare the structures of pIUC-3 and pIUC-18. We have added an explanation in the main text. Plasmid pICU-18 shared an identical backbone with pICU-3, differing only in that both *bla*_{CTX-M} copies corresponded to *bla*_{CTX-M-65} without mutations at A397G or C508T, and the *bla*_{KPC-2} genotype was still retained.

Lines 186-208 - from Fig. 2, but not really explained in the text, it seems that the copy of blaCTX-65 that is not amplified also does not undergo the mutations? If so, this might need to be taken into account when calculating/drawing conclusions from the number of copies e.g. plasmid copy number will increase the copy numbers of both this unchanging copy of blaCTX-M-65 and whichever variant is in the other position, which could be blaCTX-M-65, 247, 248 or 249, while gene amplification will not affect the copy number of the non-amplified blaCTX-M-65 copy. It is not clear if/how this has been taken into account in the calculations.

Response: We reorganized the description of this section to provide a clearer explanation. On plasmid pICU-3, *bla*_{CTX-M} was present in two copies, separated by ~54 kb. The copy corresponding to *bla*_{CTX-M-65} that is not amplified also remains unmutated, as confirmed by PCR amplification of flanking specific regions followed by cloning. The absence of tandem

duplication at this locus is explained by the opposite orientations of the flanking IS26. In contrast, the other copy located near *bla*_{KPC-157} could represent *bla*_{CTX-M-65} or one of its variants, with *bla*_{CTX-M-249} being most frequently recovered. Under moderate CZA pressure (32 mg/L), the primary effect was an increase in plasmid copy number, leading to elevated *bla*_{CTX-M} copies at both loci. The expansion mainly involved plasmids carrying the *bla*_{CTX-M-65} + *bla*_{CTX-M-249} combination rather than *bla*_{CTX-M-65} with the other three variants. Consequently, *bla*_{CTX-M} copy number relative to the plasmid remained stable but increased relative to the chromosome, with a slight higher proportion of *bla*_{CTX-M-249}, possibly accompanied by limited tandem amplification as later observed under higher CZA pressure. At 64 mg/L CZA, plasmid copy number reverted to the drug-free level, whereas the *bla*_{CTX-M-249} copy near *bla*_{KPC-157} underwent more IS26 “translocatable unit” (TU) - mediated tandem amplifications, resulting in increased *bla*_{CTX-M} copy number per plasmid and a further rise in the proportion of *bla*_{CTX-M-249}. Digital qPCR and amplicon sequencing results support this interpretation.

Lines 182-3 – how was this calculated? What are these numbers?

Response: We realized that our description of the digital qPCR method and the calculation of copy numbers was not sufficiently clear. We have now reorganized this section to make the presentation of the digital qPCR results more explicit. Specifically, we used digital qPCR to quantify the absolute copy numbers of the chromosomal housekeeping gene *pgi*, 16S rDNA (known to have eight copies per chromosome), and the plasmid-encoded single-copy genes *psiB* and *bla*_{CTX-M} (variants not distinguished) in each sample. These values were then normalized to *pgi* or *psiB* to calculate the relative copy numbers with respect to the chromosome or plasmid, respectively. Correspondingly, we also expanded the description of the digital qPCR methodology in the Methods section.

Line 188-9 – are there any clues in the sequence about what is responsible for this increase in plasmid copy number? This should be looked at.

Response: In this study, we found that the tandem amplification of one *bla*_{CTX-M} copy is likely mediated by the IS26 TUs bracketing the resistance gene. However, the molecular mechanism underlying the increase in plasmid copy number remains unclear. We performed long-read sequencing and assembly under CZA pressure and compared the results with drug-free conditions, focusing on the plasmid replicon and replication-associated regulatory genes, but detected no genetic changes that could account for the elevated plasmid copy number.

Line 193 – one of the ONT raw reads provided has 7 copies of the TU, the other 3 plus probably a truncated fourth one. How does this fit with "about three" stated here?

Response: Here “about three” was estimated based on the digital qPCR results (Fig. 2b). The value of three copies can be regarded as an average representing the overall increase in copy number. In reality, the number of expanded plasmid fragments likely varies among individual bacterial cells; for example, our raw data show cases with seven copies, incomplete expansions with four copies, as well as other cases with fewer than three copies (including some cells without amplification, remaining as a single copy). Therefore, in Fig. 2a, the schematic of copy number increase is illustrated as three copies, serving only as a representation of this average situation.

Lines 197–208 – it’s not entirely clear from Methods how these proportions were determined/calculated and a bit more detail might be helpful.

Response: The proportion of each variant within the clonal population was quantified by amplicon sequencing. Accordingly, we have reorganized the description of the amplicon sequencing methodology in the Methods section to provide a more details.

Lines 209-10 – presumably “long reads from isolates grown in 64 mg/L CAZ” is meant? This should be rewritten.

Response: We apologize for not expressing this clearly. Under different concentrations of CZA pressure, the copy number of genes on bacterial plasmids may change. Therefore, when bacteria

cultured under antibiotic pressure were subjected to long-read sequencing (Nanopore platform), we analyzed the raw reads before assembly to determine whether the target gene exhibited multicopy status. In the manuscript, the intended meaning of this sentence is that the raw long reads we analyzed were derived from bacterial genomic sequences obtained under 64 mg/L CZA pressure. We have now reorganized this sentence for greater clarity.

Line 212 – this needs to be looked at further and explained better and more accurately. IS903, or a truncated copy, is commonly found downstream of *bla*_{CTX-M-9} group genes in *ISEcp1* “transposition units” (TPUs) responsible for moving these genes. Examination of the pICU-3 sequence and raw long reads provided shows that, as expected, IS903 has nothing to do with the amplification. The amplified region instead seems to be a 4844 bp IS26 “translocatable unit” (TU; positions 58367-63210 in pICU-3 sequence provided), something that has been reported previously for different AMR genes (e.g. Refs 33, 34 cited here) forming a chain of “pseudocompound” transposons. The authors need to look at recent reviews and/or papers on IS26 mechanisms of transposition (mainly by RM Hall and colleagues, but also M Chandler and colleagues). I also would suggest looking at whether this TU has been seen before and at papers on the plasmid contexts of *bla*_{CTX-M-65} (e.g. PMID: 22984205).

Response: Thank you for your suggestion, with which we fully agree. We also consider that the tandem duplication of one *bla*_{CTX-M} copy on the plasmid is mediated by the IS26 TUs rather than by *ISEcp1*. We have reviewed recent papers by R.M. Hall and colleagues, which elucidate the mechanism by which IS26 TUs mediate the transfer of various AMR genes through the formation of chains of “pseudocompound” transposons. In recent years, numerous reports have described the formation of IS26-mediated TUs in clinical isolates, including cases where TUs carry *bla*_{CTX-M}. However, reports specifically involving *bla*_{CTX-M-65} TUs remain limited. In our previous work, we identified an IS26-mediated tandem multicopy structure of *bla*_{CTX-M-65} in an *E. coli* isolate (J Glob Antimicrob Resist, 2023; 35:202 – 209), though its structure differed from that observed in the present study. In addition, another study from China (J Antimicrob

Chemother, 2013; 68(1):46 – 50) also reported a similar IS26-mediated tandem duplication unit of *bla*_{CTX-M-65} in *E. coli*.

Lines 213-4 – this region cannot not undergo the same type of IS26-mediated tandem amplification as the flanking IS26 copies are in opposite orientations.

Response: We sincerely thank you for pointing this out, as it helps us explain why the other copy of *bla*_{CTX-M-65} cannot undergo IS26-mediated tandem amplification. Upon doublechecking the sequence, we confirmed that the IS26 elements flanking the other *bla*_{CTX-M-65} copy are in opposite orientations, which prevents the formation of a tandem amplification structure. We have also added this explanation to the main text.

3) Lines 218-329 Models of heterogeneity

This part of the text gets hard to follow and could be simplified and condensed.

Line 219 – “plasmid-harboring beta-lactamase genes” does not make sense. Presumably “plasmids harboring” and “*bla*_{CTX-M} genes” could be specified here, rather than “beta-lactamase genes”.

Response: we have replaced beta-lactamase with *bla*_{CTX-M} based the suggestion.

Lines 221-2 – genes are cloned in fusion with genes, not proteins, and the gene names are *bla*_{CTX-M-65} and *bla*_{CTX-M-249}. This should be written more accurately (and maybe moved to Methods).

Response: We have noted that the fusion was with genes, not proteins. Therefore, the sentence has been revised to: “To facilitate observation, we fused fluorescent protein genes downstream of the corresponding resistance genes.” Here we just briefly mention the use of fluorescent proteins to aid reader understanding, while the methodological details are provided in the Methods section.

Line 222-5 – “for example” should be deleted if you then list everything that was constructed. Maybe just saying here that plasmids encoding the following fusions were constructed: CTX-M-65-eGFP (p65G), CTX-M-249-mCherry (p249R) and both CTX-M-65-eGFP and CTX-M-249-mCherry (p65G-249R), using R to indicate the red fluorescence rather than “m” for mCherry, and explaining this and also that both colours together give orange.

Response: Thank you very much for this excellent suggestion. First, we removed “for example” from the main text, and then we revised and simplified the description of the model construction as recommended here. We also used “R” to indicate red fluorescence instead of “m” for mCherry, which makes it clearer for readers to associate directly with the fluorescence color. Finally, in the main text we now describe the resultant color of red and green as “orange” rather than “orange-yellow.”

Lines 226 – the plasmid names could then be used to simplify the text i.e. Model 1 is DH5alpha with p65G (strain 65G), Model 2 is DH5alpha with p249R (strain 249R), Model 3 is a 1:1 mixture 65G and 249R, Model 4 is DH5alpha with both p65G and p249R and Model 5 is DH5alpha with p65G-249R (or maybe consider putting this information a table). Having two names for Model 4/DCPs and Model 5/DCGs is also confusing and seems unnecessary. The names DCP and DCG are confusing as they are too similar (and the “s” at the end is not needed). Any further necessary details (e.g. of transformation) and Lines 234-6 could be left to Methods (or Fig. 2d legend, where they are already) and also consider whether the information in Lines 236-40 would be better mainly moved to Methods.

Response: Thank you for the helpful suggestion. We have revised the main text according to this way of describing the five models. The five models are already labeled in Figure 3, so we no longer list them in Table separately. Models 3, 4, and 5 represent different patterns of multicopy heterogeneity that we constructed, while the original terms DCP and DCG refer to the key factors distinguishing the latter two patterns. We gave them abbreviations to make it easier to describe the results later (since our subsequent research will focus on these two models) and also to make the labeling in the figure more concise. After carefully reconsidering the key

factors that distinguish these two important models, we concluded that whether the resistant gene duplicates are located on a single plasmid or on two separate plasmids is the most critical difference. Therefore, we revised the model names and abbreviations: Model 4 is now referred to as Dual-Plasmid Duplicates (DPD), meaning heterogeneous *bla*_{CTX-M} copies are duplicated on separate plasmids; Model 5 is now referred to as Single-Plasmid Duplicates (SPD), meaning heterogeneous *bla*_{CTX-M} copies are duplicated on a single plasmid. The difference between the initials “D” and “S” helps readers quickly distinguish dual-plasmid from single-plasmid. We have accordingly updated the main text and figures.

Other further necessary details, such as transformation and confirmation of fluorescence patterns, have been moved to the Methods section. The description of the x-axis and y-axis in the checkerboard susceptibility assay has also been moved to Methods, and we have reorganized the corresponding result descriptions.

Lines 240-3 – it might be clearer to say that strain 65G grew up to high concentrations of CTX, but only at low concentrations of CAZ, while 249R grew up to high concentrations of CAZ, but not in CTX, both as expected, if this is what is meant (“tolerable drug” does not make sense here).

Response: Many thanks. We have revised this description as suggested.

Lines 243-7 could then say that Models 3, 4 and 5, which express both CTX-M-65 and CTX-M-249 in different configurations, all showed growth up to relatively high levels of both CTX and CAZ, then explain the differences between Model 3 and Models 4 and 5.

Response: We have revised this description as suggested.

Lines 253 – unclear, needs rewording.

Response: We reorganized this sentence as follows: “The duplication of resistance genes into two variants with distinct resistance profiles enables bacteria to withstand the combined effects of multiple drugs simultaneously.”

Line 257 – suggest “selection with”.

Response: This section has been rephrased. To align with the restructured text and for conciseness, the phrase “selective pressures of CTX or CZA” has been changed to “under CTX or CZA pressure”.

Line 259-65 – wording does not make sense and is unclear – say that Model 4 has CTX-M-65 (green) and CTX-M-249 (red) encoded by two different plasmids in the same cell. Then that in Model 4 orange fluorescence was replaced by green in the presence of CTX (resistance conferred by CTX-M-65) and by red in the presence of CAZ (resistance conferred by CTX-M-249), then information from Lines 266-9.

Response: We have reorganized this paragraph as suggested.

Lines 265-71 – could just state that Model 5 has CTX-M-65 and CTX-M-249 encoded on the same plasmid and that orange fluorescence was maintained and reword Lines 269-71 – e.g., I don’t think that using “paralogous” here is helpful.

Response: We have reorganized this paragraph as suggested and removed the final sentence. However, we believe that the concept of “paralogous” remains important in this study. Paralogs are homologous genes within the same genome that originate from gene duplication, and they can evolve new or specialized functions, thereby diversifying gene functionality. This is essentially analogous to the process by which *bla*_{CTX-M-65} and its variants arise and develop differentiated resistance profiles. Therefore, we have retained the use of “paralogous” in other parts of the manuscript and further clarified this concept in the Discussion.

Lines 272-288, L290-329 – much of the current wording does not make sense or is unclear - the same principles as above can be applied to explain this more simply and clearly.

Response: These two sections in the Results have been rephrased to make the descriptions more concise and clear. Some method-related details were simplified or moved to the Methods

section. The last paragraph of the “competition assay” section was more in line with a discussion, so it was removed here and the relevant statements were adjusted and incorporated into the Discussion section.

4) Universality of resistance gene heterogeneity, Lines 331-61

Again, the language could be simplified.

Lines 332 – suggest something like “Having multiple copies of different variants of a resistance gene that confer different phenotypes”.

Response: Following a suggestion from another reviewer, we have moved this sentence to the Discussion section and refined it to improve the flow and conciseness of the text.

Lines 335, 350-1, 355-7 – *bla*SHV is an intrinsic chromosomal gene in *K. pneumoniae*, which is the source of plasmid-borne variants. This needs to be properly explained and I would suggest looking at PMID: 17999968, which discusses *bla*SHV variants on plasmids and the chromosome. I would also suggest looking at PMID: 28812563 about *bla*TEM. This could all have implications for the analysis.

Response: In this section of the analysis, we only consider several types of heterogeneity in resistance genes, including those on a single plasmid, among multiple plasmids, and between the chromosome and plasmids. For *bla*SHV, we have also taken this into account: the chromosome usually carries one copy, and plasmids may also acquire another copy. This situation is actually a form of heterogeneity, which we included in our statistics. The same idea has been applied to *bla*TEM. We have carefully read the two articles concerning *bla*SHV (Antimicrob Agents Chemother, 2008;52(2):441-5) and *bla*TEM (Nat Ecol Evol, 2016;1(1):10), and in fact, the latter had a very significant influence on the way we developed our research approach.

Lines 335-57 – the name *bla*OXA encompasses many families that are not closely related, so it does not make sense to give a single % for multicopy heterogeneity for *bla*OXA. Was a

particular blaOXA looked at? If so, this must be stated. If not, then the analysis may need to be redone.

Response: Thank you for this helpful suggestion. You are correct that *bla*_{OXA} encompasses many families that are not closely related. Therefore, we reanalyzed the data focusing on the *bla*_{OXA-48} family, which is widely prevalent in *K. pneumoniae*. We used the *bla*_{OXA-48} type as the reference for homology searches, enabling the identification of *bla*_{OXA-48-like} genes in bacterial genomes and thereby allowing estimation of their copy number and heterogeneity. We also reorganized the description of these results and updated the corresponding labels and numbers in the figures and tables.

Line 342 – is this sequence data available and/or associated with a publication? If so, please give the BioProject and/or citation. Suggest “raw sequence data from this species”

Response: The genomic raw sequence data of *K. pneumoniae* from our laboratory were obtained from two surveillance projects. One is a nationwide epidemiological investigation involving 28 hospitals from China, the results of which have already been published (J Antimicrob Chemother 2024; 79: 2292 – 2297); the other is a regional epidemiological investigation involving 36 hospitals in southeastern China, which has not yet been published. In the main text, we cited the published article and provided the BioProject accession number (PRJNA1290706) for the raw sequence data of the unpublished study.

Line 351-57 – which blaCTX-M group or did you look at multiple groups? Major sequence differences between members of different blaCTX-M groups (Group 1, Group 9 etc) are not due to mutations, but because the source genes are from different Kluyvera species (there are also hybrids between Group 1 and Group 9 genes). Minor changes between blaCTX-M genes within each group are probably explained by mutations that have occurred since capture.

Response: Thank you for your suggestion, and we fully agree with your point. The major sequence differences between members of different *bla*_{CTX-M} groups (Group 1, Group 9, etc.)

are not due to mutations. Therefore, if a strain carries both a *bla*_{CTX-M} gene from Group 1 and another from Group 9, strictly speaking, such heterogeneity does not correspond to the copy number – derived mutational heterogeneity that we describe in our study. However, this part of our analysis was performed using raw genomic data to assess the heterogeneity status of a specific β -lactamase gene in each strain across a large dataset. In practice, we first identify the target gene through homology search (BLAST), and then determine heterogeneity by analyzing sequence differences among all raw reads contributing to the assembly. Therefore, whether the reference used for BLAST belongs to a Group 1 or Group 9 *bla*_{CTX-M} gene does not affect our ability to first locate a specific *bla*_{CTX-M} genotype within the genome. For our analysis of *bla*_{CTX-M}, we used *bla*_{CTX-M-65} as the reference sequence. As long as a *bla*_{CTX-M} gene is present in the genome (even one from a divergent group such as Group 1), its presence can still be identified. We then analyze whether heterogeneity exists among all reads assembled into that gene.

5) Other scientific points

Lines 40, 48, 334-5 – genes encode proteins, they are not encoded. Suggest “carried”/“carry” or “plasmid-borne”, as appropriate.

Response: Thank you for the suggestions. Depending on the meaning of each sentence, we used different words to replace “encode”. For instance, in the abstract, “located” and “carried” are used instead. Please refer to the main text for details; please note that some instances may have been removed due to whole structure revisions.

Lines 75, 382 – it is not clear what is meant by “the CTX-M-14 isoform” here. *bla* gene variants are assigned numbers by NCBI. A CTX-M-14 sequence with these amino acid changes appear to have been numbered CTX-M-219, described as “inhibitor resistant”, with the accession CP019080 that cites Ref 14. See <https://www.ncbi.nlm.nih.gov/pathogens/refgene/#blaCTX-M-219>.

Response: Here, “*bla*_{CTX-M-14} isoform” refers to a variant of *bla*_{CTX-M-14} conferring resistance to

CZA, as reported in Ref. 14, which has been numbered *bla*_{CTX-M-219} by NCBI. We have already revised the original sentence in the main text as follows: “Although such reports are currently rare, there have been reports of CMY-172 and CTX-M-219, the latter representing a variant of the widely disseminated CTX-M-14.”

Line 90-1 – many large conjugative plasmids that typically carry AMR genes are present at around only one copy/cell and “duplication state” needs rewording.

Response: Yes, there are some large plasmids that exist only as one copy. However, it has been reported that plasmid copy number ranges from only slightly higher than chromosomal copy number in low copy plasmids to tens of copies per cell in high copy plasmids, which means that most plasmids have more than one copy per bacterial cell (Nat Rev Microbiol, 2021; 19(6):347 – 359). Our wording was not sufficiently rigorous, and we have revised “which exist” to “most of which exist”.

Line 116 – homology is not quantifiable, so what is really meant by “highly homologous” here? “closely-related”? This is from the SNP distance of <10? Also suggest “isolates” here, Line 122 etc.

Response: Yes, here we intended to express that the two isolates were highly similar in their genomic sequences, based on the result of an SNP distance of <10. We have revised the sentence to: “we discovered two bloodstream infection isolates that were closely-related.”. At another site, we also replaced “strains” with “isolates”.

Lines 123-4 – which *bla*_{KPC} allele does ICU-18 have? This must be stated.

Response: In ICU-18, the *bla*_{KPC} allele is *bla*_{KPC-2}, whereas in ICU-3, after the A392G mutation, it is *bla*_{KPC-157}. We have added this description in the main text.

Line 136 – this should be “mutations” not “mutants”

Response: We agree with your suggested revision; however, since this section has already been reorganized, we adopted alternative wording.

Lines 139, 403, 405 etc. – I don't think that it makes sense to use "wildtype" – it can be deleted.

Response: We reviewed the instances where "wild-type" was used in the main text and deleted most of them. Only at a sentence in Discussion ("CTX-M-247 and CTX-M-248 each have one of these substitutions, suggesting they are intermediates in the evolution from the prevalent *bla*_{CTX-M-65} to the novel *bla*_{CTX-M-249}."), we changed "wild-type *bla*_{CTX-M-6}" to "prevalent *bla*_{CTX-M-65}", to correspond with "novel *bla*_{CTX-M-249}".

Lines 171-7 – are there other studies reporting on the effects of these mutations?

Response: The dual mutations at positions 130 and 167 in *bla*_{CTX-M-65} has not yet been reported. However, the effects of single mutation at each of these sites on CTX-M enzymatic hydroactivity have been documented in several studies. For example, in *Antimicrobial Agents and Chemotherapy* (2003; 47(9):2958 – 2961), the Ser130Gly mutation was shown to reduce the hydrolytic activity of CTX-M enzymes against cefotaxime while increasing resistance to inhibitors including sulbactam, clavulanic acid, and tazobactam. In *PLoS Pathogens* (2010; 6(1):e1000735), the Pro167Ser mutation was reported to decrease the hydrolytic activity of group 1 CTX-M enzymes against cefotaxime. By contrast, in *Antimicrobial Agents and Chemotherapy* (2006; 50(2):731 – 738), the Pro167Ser mutation was shown to enhance the hydrolytic activity of group 9 CTX-M enzymes against ceftazidime.

Line 200 – I would suggest making these % less precise.

Response: We appreciate this helpful suggestion. In fact, the proportion of each variant under different drug conditions is unstable and difficult to estimate with high precision. We have made revisions and retained only the single-digit place of the percentage values.

Line 210 – specify blaKPC-157 here.

Response: We have revised “*bla*_{KPC}” with “*bla*_{KPC-157}” as suggested.

Line 282 – LB originally stood for “lysogeny broth”.

Response: We have replaced “Luria-Bertani (LB) broth” with “Lysogeny Broth (LB)” as suggested.

5) Discussion

This is long and wordy and could be condensed. The discussion section should not repeat information in Results (or Introduction) and comments on the rest of the manuscript need to be considered.

Line 365, 482 – is this really completely novel?

Response: What we refer to as a “novel mechanism” in our study actually describes a bet-hedging evolutionary phenomenon or strategy that we observed and proposed. Specifically, multiple copies of paralogous resistance genes allow bacteria to maintain their ability to withstand multidrug pressure, while the fact that these copies are located on the same plasmid ensures they are less likely to be lost. This is particularly significant because their placement on a single plasmid allows for more stable functionality—an aspect not previously highlighted in earlier studies. We revised corresponding sentences to describe this bacterial evolutionary phenomenon as observing a novel strategy (emphasizing the observed phenomenon) rather than identifying a novel mechanism (emphasizing the discovered principle). We have rephrased the sentence accordingly.

Lines 365-76 – this seems more like introductory information and parts seem to repeat the Introduction.

Response: We reorganized the first paragraph of the Discussion, removing Introduction-like statements such as “With the widespread use of antibiotics, the prevalence of resistant bacteria

has become a significant clinical issue” and “has been listed as a priority pathogen due to its severe multidrug resistance”. Other sentences were streamlined to serve as a transition to the subsequent content.

Lines 382-3 – wording does not make sense.

Response: We have reorganized this sentence as follows: “As a variant of *bla*_{CTX-M-65}, the clonal strain carrying *bla*_{CTX-M-249} demonstrated a trade-offs phenomenon, as both antimicrobial susceptibility testing and enzyme kinetics assays showed that it gained CZA resistance while losing resistance to CTX (decreasing from 128 mg/L to 0.5 mg/L).”.

Line 392 – “reflected on several level of meanings” does not make sense.

Response: We have removed this sentence.

Line 394-5 – the plasmid copy number and how it was determined and the distance between the *bla*_{CTX-M-65}-like genes should be stated in Results.

Response: The contents described in this sentence have been moved to the Results section.

Line 397 - “elements” is not needed – these are IS.

Response: The word “elements” has been removed as suggested.

Line 416 - “AS” is not needed here and suggest referring to the paper without naming the authors

Response: Following your suggestion, this sentence has been reorganized.

Line 417 – the genes are *bla*_{TEM-1} and *bla*_{TEM-12}

Response: They have been revised as suggested in the main text.

Line 428 – it is the constructed plasmids that are lost?

Response: Yes. If the heterogeneity of the resistance genes in two copies is constructed on two separate plasmids within the same bacterial cell, one of the plasmids can be easily lost in the absence of selective pressure. Correspondingly, when both resistance genes are located on the same plasmid, this heterogeneity advantage can be anchored in even under changing selective pressures.

Lines 434-6 – why wasn't this data included?

Response: Because of this, we can theoretically deduce that a combination of low concentrations of cefotaxime and CZA can effectively eliminate the bacteria. We have included the results of this checkerboard susceptibility assay in the supplementary figure 7.

Line 436 – “inhibitor” is not needed here.

Response: The word “inhibitor” has been removed as suggested.

Line 438 – wording does not make sense.

Response: This sentence has been revised as: “This likely reflects avibactam’s inhibition of CTX-M-65, while CTX-M-249 exhibits only weak hydrolysis of CTX.”.

Line 446 – I think that heteroresistance can also be due to copy number? See papers by D. Anderson.

Response: Yes, heteroresistance can also be due to copy number (Nat Rev Microbiol. 2019; 17(8):479–496), because instability in the copy number of specific resistance genes within a population may lead to differences in gene expression levels, thereby resulting in heteroresistance. However, in our study, we would like to emphasize that gene amplifications generate multiple copies of this gene, and different point-mutations can occur among these copies, leading to heterogeneity in resistance phenotypes. This is also what we aimed to present in Fig. 7a. The coexistence of multiple variants of resistance genes together with copy number variation can further complicate heteroresistance within the population.

Line 452 – use “and/or” and delete “dual site mutations”

Response: We have revised this sentence as suggested.

Lines 457-65 – this information might be in the legend of Fig. 7, which currently very short.

Response: This paragraph has been moved to the legend of Fig. 7.

Lines 468-9 – unclear what is meant here.

Response: We revised the wording of this sentence to make it clearer. The revised version is:

“In contrast, multicopy heterogeneity denotes diverse resistance gene alleles that generate varied resistance profiles, potentially enabling adaptation to multidrugs.”

Line 470 – “studies”

Response: The word “study” has been replaced by “studies”.

Lines 471-5 – is this really true for long-read if multiple copies of the gene are in different genetic locations, as for copies of *bla*_{CTX-M-65} (or variants) both in pICU-3 here?

Response: What we intend to convey here is that even with third-generation sequencing based on Nanopore or PacBio platforms, the copy number of tandem repeats may still be misrepresented after assembly. This is because assemblies tend to merge raw reads into consensus sequences, whereas the actual copy number of tandemly repeated genes can vary even within a single clonal population. Moreover, when multiple copies of a gene are located in different genetic contexts—as with *bla*_{CTX-M-65} (or its variants) present in both regions of pICU-3—the relatively lower base accuracy of long-read sequencing makes it difficult to resolve which specific variant resides in which repeat region.

Line 484 – what is meant by “horizontal transfer elements” here?

Response: Here, “horizontal transfer elements” referred to IS elements, and we have now modified it to “some ISs.”

Line 492 – “introduction” seems like the wrong word here – maybe “recognition”

Response: The word “introduction” has been replaced by “recognition”.

6) Methods

These lack detail/clarity in places, but text in other places could be condensed.

Primers - although the text refers to Table 3 in several places, it is not always immediately clear which primer sets are being referred to and it might be helpful to state the primer names in the text or better indicate in Table 3 which section they were used for.

Response: Thank you for your suggestion. In Table S3, we had already indicated the purpose of each reference in the “Usage” column. Following your suggestion, we have now added another column specifying the section of the main text where each reference was cited.

Line 499 – presumably “species identification” is meant here?

Response: We have revised this description as suggested.

Lines 523-7 – the wording does not make sense here and more details are needed. What was cultured for the PCR?

Response: Regarding the “Cloning the β -lactamase genes” methodology, we have thoroughly reorganized this section and added more details. Cloning in this study refers to the process in which the PCR product of the target gene from a clinical strain was ligated into the vector plasmid pCR2.1 and then transformed into a recipient strain, so that each transformant successfully obtaining the recombinant plasmid (also referred to as a colony) would carry only one variant of the PCR product. PCR products amplified from DNA extraction from the clinical strain ICU-3.

Lines 573-84 – this section can be condensed. The first sentence is not really needed, is something like “Three reporter plasmids based on pACYCDuet1 were constructed” and state which plasmid has which gene. Genes are cloned in fusion with genes, not proteins, and first explain that pACYCDuet1 has a cat gene.

Response: The corresponding paragraph has been rewritten and condensed, and we revised the methodological descriptions in line with your recommendations.

7) Other minor/formatting/English

Title needs some rewording – suggest “Heterogeneity in multiple copies of blaCTX-M-65-like genes in a clinical isolate of *Klebsiella pneumoniae*”.

Response: We have reconsidered the title, striving for conciseness, focus, and a clear reflection of the significance of the study. Your suggestions were very helpful, and we referred to them during the revision. However, the location of multiple-copy genes is very important, because one of our novel findings is that when two paralogous genes with different resistance profiles are located on same plasmid, this bet-hedging feature can be stably maintained. In addition, another important significance of our study is that the heterogeneous multicopy enhances evolutionary adaptability, so this also needs to be reflected in the title. After considering all opinions, we have revised the title to: “Heterogeneous multicopy resistance genes on a single plasmid enhance evolutionary adaptability in clinical *Klebsiella pneumoniae*”.

Throughout – odd/unscientific phrasing needs to be changed, e.g., Line 33 “combat”, Line 42 “lock in, Line 55 – “constant battle”, Line 77 - bacteria do not really “choose their evolutionary paths”, Line 386 - “ingeniously explored”, Line 423 “solidified”, Lines 448, 454 “locked on/onto”.

Response: We greatly appreciate these very detailed suggestions and have revised the text accordingly, word by word. The terms “combat”, “lock in” in the abstract, and “constant battle” have been removed in the reorganized paragraph. “how these bacteria choose their evolutionary paths” has been changed to “the trajectory of bacterial resistance evolution”; “ingeniously

explored” has been simplified to “explored”; “solidified” has been revised to “stably co-localized”; and expressions such as “locked on/onto” have all been replaced with “anchored in”

Throughout – “the blaCTX-M-65” is incorrect. This should either be “the blaCTX-M-65 gene” or preferably just “blaCTX-M-65” e.g. Line 102. (except Line 36, where “the” is needed before “CTX-M-65-type”). Also, for other gene/allele names. “bla” in the name indicates the gene

Response: We carefully checked all occurrences of *bla*_{CTX-M-65} and other gene names throughout the manuscript and corrected any inappropriate expressions.

Throughout – the abbreviation CZA is used for ceftazidime/avibactam. The common abbreviations CAZ and CTX, as used in figures, could also be defined and used in text.

Response: We revised the abbreviations for antibiotics in both the main text, figures and tables. Specifically, CZA is used for ceftazidime/avibactam, while CAZ and CTX represent ceftazidime and cefotaxime, respectively.

Line 37-8 – I don’t think that “monoclonal” is commonly used to describe bacteria? Maybe “a single strain was found to have multiple copies of blaCTX-M-65 or minor variants that conferred different spectrums of resistance”.

Response: We agree that “monoclonal” was not the most accurate wording and have replaced it with “clonal” or “a single clonal,” as appropriate.

Line 42 ,412 etc – I am don’t think that that “paralogous” is the clearest term to use.

Response: Thank you for your suggestion. We have revised the use of “paralogous” in certain paragraphs. However, we still believe that the concept of “paralogous” remains important in this study. Paralogs are homologous genes within the same genome that originate from gene duplication, and they can evolve new or specialized functions, thereby diversifying gene functionality. This is essentially analogous to the process by which *bla*_{CTX-M-65} and its variants arise and develop differentiated resistance profiles. Therefore, we have retained the use of

“paralogous” in some parts of the manuscript and further clarified this concept in the Discussion.

Line 47 – suggest “clinical isolates” rather than “patient-isolated strains”.

Response: We have revised this sentence as suggested.

Lines 47-50, 61-62, 73-74 need rewording.

Response: All of the sentences you mentioned have been reorganized.

Line 80-1, 108, 384, 389 suggest just “trade-offs”, rather than the “the trade-offs effect” – or at least “the trade-off effect” and reword “may introduce new roles”.

Response: In the sections you noted, we have replaced the wording with “trade-offs.” The sentence “may introduce new roles” has also been rewritten.

Line 84-5 – “Compared with... mutations, genetic amplifications” – i.e. delete “the”

Response: We have revised this sentence as suggested.

Line 93 – either “multicopy resistance plasmids” or “multiple copies of a resistance plasmid”

Response: We have revised this phrase with “multiple copies of a resistance plasmid”.

Lines 95-6, 110 – need rewording.

Response: All of the sentences you mentioned have been reworded.

Lines 99-102 – this long sentence might be better split but certainly needs rewording.

Response: This sentence has been reworded.

Line 111 – “considerable numbers”

Response: Since the paragraph was reorganized, this sentence has been deleted.

Line 122 – “The results showed” not needed. Suggest “Both isolates belong to ST11 and differ by <10 SNPs”

Response: We have revised this sentence as suggested.

Line 126 – “clones with blaKPC-157”

Response: We have replaced the “clones of *bla*_{KPC-157} gene” with “clones with *bla*_{KPC-157} gene”.

Lines 157-60 – suggest “each has one of the two mutations seen in CTX-M-249” and “the two common third generation cephalosporins” is not needed (they have already been discussed).

Response: This paragraph has been reorganized and the sentence you mentioned has been removed.

Line 180 – “long-read sequence assembly”

Response: We have revised this phrase as suggested.

Line 227 – “respectively” needs to be at the end of the sentence.

Response: This paragraph has been condensed and the sentence has been thoroughly revised.

Line 228 – “mixing of”

Response: This paragraph has been thoroughly revised.

Line 261 - “observation of fluorescent microscope image” is not needed

Response: We have reorganized this sentence according to your suggestion.

Line 337 – “mutation” i.e. no “s”.

Response: This paragraph has been reorganized and the sentence you mentioned has been removed.

Lines 339-40, 354-5, 654-5 etc. – I would suggest using “isolates from humans or other sources”.

Response: All of the description about “*Homo sapiens*-derived” or “non-*H. sapiens*-derived” in the main text, figures and tables have been revised by “Humans” or “non-human sources”.

Line 513 – “was performed”

Response: We have revised “were performed” with “was performed” as suggested.

Line 517 – “between ICU-3 and ICU-18 was calculated”.

Response: We have revised this sentence as suggested.

Line 530 – don’t need both “enzymes” and “enzymatic” here.

Response: We revised this sentence as “To confirm the enzymatic activity of these β -lactamase,”.

Line 534 - “successful different” needs rewording.

Response: The “successful different” has been revised as “The resulting”.

Lines 540-1 do not make sense and need rewording. I don’t think that “velocities” is the right word here.

Response: This sentence has been revised and the “initial velocities” has been modified with “initial rates”.

Line 557, 569 etc. - “are listed”, not “were listed”

Response: We have replaced the “were listed” with “are listed”.

Line 562 – “as for amplicon sequencing”.

Response: We have revised this sentence as suggested.

Lines 562-3 – the wording here does not make sense.

Response: This sentence has been revised as follows: Strain processing and DNA extraction were performed as for amplicon sequencing. Genomic DNA (10 ng/μL) was then ultrasonically sheared on ice.

Lines 568-9 – this is unclear and needs to be explained better.

Response: This section has been reorganized with additional methodological details.

Lines 578 – “removed”

Response: We have revised this word as suggested.

Line 579, 80 – “DCP strains”, “DCG strains”

Response: This section has been thoroughly reorganized, and the terms DCP and DCG have been replaced with DPD and SPD throughout the main text to improve clarity.

Line 581 – “dual antibiotic”

Response: This sentence has been revised and the “dual antibiotic” has been removed.

Line 588 – not clear what is meant here and “including” is not needed.

Response: This sentence has been revised as follows: Strains representing different models (65G, 249R, a 65G/249R mix, DPD, and SPD) were inoculated into wells containing serial dilutions of CTX, CAZ, or their combinations to assess survival.

References are normally placed before figure legends. Ref 8 needs page numbers or equivalent. Something seems to be wrong with page numbers in Ref 49. Journal titles need to be correctly abbreviated. Ref 4, 12 etc., remove Title Case. Species and gene names etc. need to be correctly formatted.

Response: We have placed the references before the figure legends and carefully checked all reference entries, correcting errors and inconsistencies in formatting. Species and gene names have also been italicized accordingly.

8) Figures

General – some parts could be moved to supplementary, shorter titles would be better and legends could be condensed.

Response: We sincerely appreciate your thoughtful suggestions. Taking into account comments from multiple reviewers, we have made substantial adjustments to the composition and order of the figures. Several panels have been moved to the Supplementary Materials, concise titles have been added to each figure, and the legends have been condensed.

Fig. 1

I think that most parts of this figure are not needed or could be moved to supplementary. I would suggest retaining the grid showing resistance profiles from part a (with amino acid changes for each CTX-M variant listed) plus part d in the main manuscript. Maybe consider adding CTX-M-219 (“CTX-M-14 isoform”) discussed in the text to part b, if retained as Supplementary.

Response: In line with other reviewers’ feedback, we expanded the workflow originally shown in Fig. 1a to help readers more easily follow the overall logic of the study. Panels b, c, and d of the original Fig. 1 have been moved to Supplementary Fig. 1, with panel b now including an amino acid sequence alignment of CTX-M-14 and CTX-M-219. The resistance profiles previously shown in panel d are now represented in the new Fig. 1 (part I) using a grid map, which also lists the amino acid changes for each CTX-M variant.

Fig. 2

Part a - on the circular diagram of the plasmid at least the two IS26 flanking blaCTX-M-65/247/248/249 should be shown, plus preferably the other AMR genes, IS and transposons in this region and surrounding the other copy of blaCTX-M-65 (see ISfinder). The diagram to side

should properly show the boundaries of IS26 and of all copies of the amplified unit. The *bla*SHV gene number should be added.

Part b CZA 32 mg/L – colour is too pale, hard to see

Response: We updated the content and color scheme of Fig. 2 to improve clarity and visual contrast. In the circular diagram of plasmid pICU-3, both loci of *bla*_{CTX-M-65} and its variants are highlighted in red, and their flanking IS26 elements are annotated along with other AMR genes, ISs, or transposons. In the enlarged gray-background inset, the IS26-TU structure is delineated, with an example of three tandem TU copies shown as a multicopy state. The *bla*_{SHV-12-like} gene is also annotated. In panel b, colors were deepened for better visibility.

Fig. 3

Suggest moving parts a-c to supplementary, simplifying the plasmid diagrams in part b and showing each model above the corresponding plot in part d. The wording of this part of the legend also needs improving – e.g. it is not expression of plasmids, but of fused genes.

Response: Following your advice, we moved panels a and c of Fig. 3 to the Supplementary Materials, while panels b and d were reorganized to place the plasmid structures of each model directly above their corresponding checkerboard assay results. The legend has been rewritten accordingly.

Fig 4

Part a – suggest stating clearly what is in DCPs and DCGs on the figure, rather than having to refer to text in legend. Using the names suggested above would help with this.

Part b – this could be supplementary and just show the results in part d in the main manuscript.

Part c – this could be supplementary and just show the results in part e in the main manuscript

Line 749 in legend – respectively appears incorrect here – it is not clear what is being referred to.

Response: Considering feedback from several reviewers, we moved panels a, b, and d of Fig. 4 to the Supplementary Materials. Panel c, which shows flow cytometry counts, was revised

and retained. Panel e, which reflects colony counts from dilution plating, has been converted from a bar chart to a line chart for consistency with panel c, and both are now presented together in the new Fig. 4. Legends were reorganized and simplified.

Fig. 5

Parts a and b do not both need to be shown.

Parts d and f could be moved to supplementary, just showing the summary of results in parts e and g.

Response: For Fig. 5, only panels e and g were retained in the revised version. Panels a, b, c, d, and f were moved to the Supplementary Materials, with the legend correspondingly reorganized.

Fig. 6

See comments above on analysis of *bla*SHV, *bla*CTX-M and *bla*OXA genes.

Line 781 – “including” is incorrect here, as all genes analysed are then listed

Response: As described in our response to earlier comments, Fig. 6 was also revised. In the *bla*OXA analysis, the reference sequence was updated to *bla*OXA-48, and annotations for human versus non-human sources were updated. The related results section in the main text was substantially revised to improve accuracy and clarity.

Fig. 7

What is the evidence for multidrug resistance pressure resulting in two copies of *bla*CTX-65 on the plasmid? What to the following unlabeled arrows represent?

Response: We also modified Fig. 7a to make the schematic more intuitive. Fig. 7 is a conceptual model figure, where the proposed evolutionary trajectories are hypotheses based on current evidence, intended to stimulate further thought. In Fig. 7a, the dashed box represents several possible states of multicopy heterogeneity of resistance genes within a single clonal population. The unlabeled arrows indicate inferred evolutionary trajectories: at one locus, *bla*CTX-M-65 may first acquire either of two point-mutations (yielding *bla*CTX-M-247 or *bla*CTX-M-248). If both

mutations occur together, *bla*_{CTX-M-249} may emerge, conferring CZA resistance. Furthermore, multicopy tandem duplication of *bla*_{CTX-M-249} at this locus could occur, further enhancing resistance levels.

9) Supplementary

Table S1

How do values compare with previous determinations for CTX-M-65, if available?

Response: Thank you for your valuable suggestions. In this study, we measured the kinetic parameters (e.g., K_m , k_{cat}) of CTX-M-65 and its variants (CTX-M-247, CTX-M-248, and CTX-M-249) to evaluate their catalytic properties toward antibiotics and inhibitors. In response to your suggestion regarding comparison with previous studies, we searched the literature and found only one relevant report on CTX-M-65 kinetics (J Med Microbiol, 2009; 58(Pt 6):811 – 815). However, this publication did not provide sufficient experimental details (e.g., buffer pH, temperature, substrate concentrations, or enzyme purification methods), which limited the feasibility of a direct comparison. To ensure comparability in our own work, we purified both the CTX-M-65 and its three variants and performed kinetic assays under identical experimental conditions, thereby minimizing variability.

Table S3

It should be possible to fit this table on a single page, for ease of reading, e.g. reduce margins, “amplification” is not needed in every line of the last column, remove “the” before “*bla*CTX-M-65” etc. Final column heading could be “Purpose” instead of “Usage”. *PsiB* should be *pisB* in final column, if the gene is meant.

Response: We also appreciate your suggestions on optimizing Table S3. Following your advice, we adjusted margins and fonts so that the table now fits on a single page. “Usage” was changed to “Purpose,” “Name” to “Primer/Probe,” and the unnecessary “the” preceding genes such as *bla*_{CTX-M-65} was removed. In addition, “*PsiB*” was corrected to “*pisB*”.

CTX-M-F/R/P – blaCTX-M should be used for the gene and which blaCTX-M genes/alleles would be amplified?

Response: In the “Purpose” column of Table S3, we replaced “CTX-M” with *bla*_{CTX-M}. The CTX-M-F/R/P primers were used for digital qPCR to amplify all *bla*_{CTX-M-65} and its variants (*bla*_{CTX-M-247}, *bla*_{CTX-M-248}, *bla*_{CTX-M-249}) in order to quantify total copy numbers, while the relative proportions of each variant in a single clonal population were determined by amplicon sequencing.

Table S4

This could be split into two tables, one with all isolates from humans, with this stated in the title and the final column removed, and one for other host species giving the details. I don’t think that the “Study” column is really needed – readers can get this information from the BioProject, if required. BioProject numbers and host species would be easier to read if aligned to the left.

Response: As you suggested, we split Table S4 into two separate tables and revised the column titles. The final column was removed, and BioProject numbers as well as host species entries are now left-aligned.

REVIEWER COMMENTS

Reviewer #1 (Remarks to the Author):

The authors responded to my comments appropriately. Simulation results based on real experimental data added strength to their message. I recommend the authors' current results for publication in this journal.

Legibility of this manuscript has been improved. However, I still found several vague descriptions/presentation in the current manuscript. Furthermore, the simulation program was not released to public yet. Currently, readers are difficult to reproduce data analysis.

The following minor points should be improved.

Line 117-121. This sentence is a little too long for me. I recommend the authors rephrase this sentence.

Response: Thank you for your suggestion. This sentence here states the aim of this study. We have condensed it as follows: The aim of this study was to explore how variants coexistence, plasmid location and multicopy heterogeneity support bacteria resistance and promote an evolutionary bet-hedging strategy under multidrug pressure.

Line 628: The software platform: Python, R, C++, or MATLAB etc is not described. Codes should be deposited to public repository such as Github, figshare.

Response: We thank the reviewer for pointing this out. All simulations presented in this study were performed using MATLAB R2021a (The MathWorks, Inc., Natick, MA, USA); this information has now been clearly stated in the Methods section. To ensure transparency and reproducibility, the complete simulation codes have now been deposited in a public GitHub repository: <https://github.com/Rui-Weng/Heterogeneous-multicopy>. The corresponding links have been added to the "Code availability" section of the revised manuscript.

Line 628: Unit of time used in simulation is hour? Please mention.

Response: Yes, the unit of time in all simulations is hour. This has now been explicitly stated in the revised manuscript in the Methods subsection “Parameter estimation” as: “All simulations were performed with time measured in hours.”

Line 699: I think that Code availability section is also needed.

Response: We thank the reviewer for the suggestion. Simulation codes and the codes for GATK-based SNV/InDel calling are available at <https://github.com/Rui-Weng/Heterogeneous-multicopy>.

Line 1038 panel e,f: Does this assume “one-round” batch culture competition assay starting with with 1:1 ratio mixture of SPD and DPD at the density of 1/100 of final density, but not serial batch culture transfer like experiments in panel c,d ?. Please clarify.

Response: Yes, previous Supplementary Fig. 6 e-f represents a one-round batch culture competition simulation: DPD and SPD were initially mixed at a 1:1 ratio and cultured for 24 h without any dilution or serial transfer. The results show the final population densities after 24 h under different antibiotic concentrations. The black dashed line indicates the antibiotic concentration at which the total amount of SPD equals the total population derived from DPD (DPD + 65G + 249R), defining the boundary. This setup differs from the serial (1:100 dilution) transfer simulations shown in panels a - b. To avoid confusion, we have clearly annotated this in the revised figure legend and the corresponding MATLAB code as “single-cycle 24-h batch culture simulation (no serial transfer).” The previously referenced Supplementary Fig. 6 c-f have been moved to new Fig. 6 in the revised version. The models formerly referred to as SPD and DPD have been renamed in the updated manuscript as H1P (Heterogeneity in One Plasmid) and H2P (Heterogeneity in Two Plasmids), respectively.

This is just suggestion: Supplementary figure S9- panel c,d,e,f are important results. I suggest the authors place these panels as a new main figure or add these to Figure 5 (competition assay results figure).

Response: We believe the reviewer is kindly referring to Supplementary Fig. S6. We have promoted panels c-f of Supplementary Fig. S6 to a new main Fig. 6. Panels a and b of the original Supplementary Fig. S6 are retained in the Supplementary Materials as the revised Supplementary Fig. S6. Consequently, the original main Figures 6 and 7 have been renumbered as Figures 7 and 8, respectively, and all figure citations in the manuscript have been updated accordingly.

Line 666 and Line1058: right panel Y-axis. The unit in these growth curves appear to be CFU/mL. How OD600nm values (source OD data) were converted to CFU/ml is not clear. Please mention in the main text or legend.

Response: We thank the reviewer for pointing out this ambiguity. In our experiments, we used the Monod growth model (ref. 60 and 61) to simulate strain growth, where population size is expressed in CFU/mL. To convert the experimental OD₆₀₀ measurements to CFU/mL, we followed the method and formulation described in *Mol Biol Evol.* 2014;31(1):232-8, establishing a standard calibration curve to determine a linear conversion function. Therefore, all simulation results are presented in CFU/mL. This has now been explicitly stated in the revised manuscript in the Methods subsection “Parameter estimation” as: “OD₆₀₀ values were converted to CFU/mL as previously described”.

Supplementary Table 5: μ_{\max}/K should read μ_{\max}/K_i .

Response: We thank the reviewer for this comment. All parameters ($\mu_{\max,i}/K_i$, ρ_i , γ_i^T , γ_i^Z , β_i^T , β_i^Z) in Supplementary Table 5 (now Supplementary Table 4) have been corrected to include proper subscripts. The revised table is included in the resubmitted supplementary materials.

Reviewer #2 (Remarks to the Author):

I had the opportunity to review the first version of this manuscript and have now had the pleasure of evaluating the revised version entitled “Heterogeneous multicopy resistance genes on a single plasmid enhance evolutionary adaptability in clinical *Klebsiella pneumoniae*”.

The authors have clearly made a substantial effort to address the comments and suggestions provided by all reviewers. The revised manuscript shows significant improvement in structure, clarity, and overall readability.

The reorganization of figures and sections has made the study more coherent and easier to follow.

The updated workflow figure (Fig. 1) effectively guides the reader through the study and provides a clearer overview of the experimental design.

Overall, the authors have responded thoroughly and constructively to the reviewers’ feedback, and the manuscript has improved notably as a result. The revisions have strengthened both the presentation and the scientific clarity.

I find that all my major concerns from the first review have been addressed satisfactorily.

At this point, I only have a few minor comments.

Minor comments

The language occasionally shifts between tenses, so a brief consistency check would also be beneficial.

Response: Thank you for your valuable suggestion. We have carefully revised the manuscript, especially the language, throughout the manuscript. During the revision process, we paid particular attention to maintaining tense consistency and ensuring linguistic accuracy and fluency.

Line 41–46:

“Within this clonal population, blaCTX-M-65 and three of its variants were present...”

Please list all alleles explicitly here (e.g. blaCTX-M-65, blaCTX-M-249, ...) to avoid ambiguity.

Additionally, blaCTX-M-65 is a specific allele within the CTX-M-9 group, not a group itself. Therefore, the other genes are not truly “variants of blaCTX-M-65.” They are better described as other CTX-M-9 group blaCTX-M alleles closely related to blaCTX-M-65.

Suggested rephrasing: “Within this clonal population, blaCTX-M-65 and additional CTX-M-9 group blaCTX-M alleles (blaCTX-M-247, blaCTX-M-248 and blaCTX-M-249*) closely related to blaCTX-M-65 were identified.”

Please apply consistent terminology throughout the manuscript.

Response: This is an excellent suggestion. We agree that it is more accurate to describe *bla*_{CTX-M-247}, *bla*_{CTX-M-248} and *bla*_{CTX-M-249} as “alleles closely related to *bla*_{CTX-M-65}” Accordingly, we have revised the sentence in the Abstract following your advice and slightly adjusted the surrounding sentences to improve accuracy and conciseness. Following the same principle, we also modified similar expressions throughout the manuscript, replacing phrases such as “*bla*_{CTX-M-65} and three of its variants” with “*bla*_{CTX-M-65} and three closely related alleles”. In some contexts, we used the more general term “*bla*_{CTX-M} variant” when appropriate.

Line 77–79:

“...or blaKPC they carry has undergone...”

Consider replacing the first “or” with “the” and moving this phrase towards the end of the sentence for smoother flow, together with the information in lines 80–81.

Response: We have revised this one and the following related sentence as: “For instance, *K. pneumoniae* strains have acquired metallo-β-lactamases (MBLs) or exhibit increased expression mediated by multicopy of *bla*_{KPC}, both of which contribute to resistance. The *bla*_{KPC} they carry has also undergone point-mutations, mostly within the omega loop, similar to those observed in extended-spectrum β-lactamases (ESBLs) that mediate resistance to CZA. However, such omega loop mutations in ESBLs remain rare, with only a few CZA-non-

susceptible cases reported, including CMY-172 and CTX-M-219, the latter being a closely related allele of the widely disseminated CTX-M-14.”.

Line 108–109:

See comment about line 41-46. As correctly noted in line 112, blaCTX-M-65 belongs to the CTX-M-9 group. Therefore, the other genes mentioned should not be referred to as “variants of blaCTX-M-65,” since they represent distinct alleles within the CTX-M-9 group, not subvariants of blaCTX-M-65. Please rephrase to “other CTX-M-9 group blaCTX-M alleles (closely related to blaCTX-M-65).” blaCTX-M-65 was the first to be described in the literature; however, due to heterogeneity among these alleles, it is not possible to determine which appeared first in the isolate, and thus which may be the derivative of the other.

Response: We fully agree with your comment. We have revised this sentence according to your suggestion and made minor syntactic adjustments to improve accuracy and conciseness. We also rechecked the entire manuscript for similar phrases such as “*bla*_{CTX-M-65} and three of its variants” and revised them accordingly. Wherever applicable, we now use expressions such as “*bla*_{CTX-M-65} and three closely related alleles” or, in some cases, the general term “*bla*_{CTX-M} variant”.

Line 132–133:

Consider clarifying this sentence to read: “...included the nucleotide substitution A392G in...”

Response: We have revised this description as suggested.

Line 415, 422, and 426:

Figure 7 is first introduced in the Discussion. New material should not be presented at this stage. Consider referring to Figure 7 earlier in the Results section..

Response: Thank you for this valuable suggestion. Considering that Figure 7 in the original manuscript presents a diagram of an inferred evolutionary trajectory, we also agree that this content fits better in the Results section. Therefore, we have added a new subsection at the end

of the Results section and moved this part from the Discussion to there. The title of this new subsection is “Inferred evolutionary trajectory”.

Figure 1:

Consider referring explicitly to the different parts of Figure 1 within the Methods section to improve readability and guide the reader through the workflow.

Response: We have now specified in the Methods section which experimental methods correspond to each part of Figure 1, so that readers can clearly understand which techniques were used in each part of the study design.

Reviewer #3 (Remarks to the Author):

This revised manuscript describes detection of blaCTM-65 and minor variants on the same plasmid in a clinical isolate of *Klebsiella pneumoniae*, with different gene combinations and copy numbers influencing resistance to ceftazidime, cefotaxime and ceftazidime+avibactam. Different models with different combinations of plasmids carrying blaCTX-M-65 and/or variants were used to explore the dynamics. Available sequences were also examined to look for multiple variants of selected bla genes in a single isolate, which suggests that this may be more common than is usually recognized.

While the data are substantial, interesting and useful, and the authors have addressed many points raised by all three reviewers, improving the manuscript, there are still a few problems with scientific accuracy etc, as detailed below. While I think that the manuscript structure is better, further condensing/reorganising and editing/simplifying the text could still improve accuracy, ease of reading and flow – some examples are given below, including figure legends.

Scientific points/accuracy

1) ORIGINAL COMMENT: Line 90-1 – many large conjugative plasmids that typically carry AMR genes are present at around only one copy/cell

RESPONSE: Yes, there are some large plasmids that exist only as one copy. However, it has been reported that plasmid copy number ranges from only slightly higher than chromosomal copy number in low copy plasmids to tens of copies per cell in high copy plasmids, which means that most plasmids have more than one copy per bacterial cell (Nat Rev Microbiol, 2021; 19(6):347–359). Our wording was not sufficiently rigorous, and we have revised “which exist” to “most of which exist”.

NEW COMMENT: The paper listed here (cited as Ref 22) also states that “Low copy number plasmids (LCPs) are typically large (from tens to hundreds of kilobases), have a low copy number and are frequently conjugative”. Many plasmids carrying AMR genes fit this category – probably including pICU-3/pICU-8 studied here - so while “most of which exist” could be numerically accurate, I would still argue that the most important plasmids for spreading AMR are mainly low copy.

This also relates to Reviewer 1’s first comment, about using a low-copy number plasmid for models. While I appreciate the problems with this explained in the response to this comment, I think that it is important to at least state the copy number of pACYCDuet-1 somewhere (apologies if this is already there and I missed it) and maybe also explain that there were technical problems with using a lower copy number plasmid. I think it’s also important to acknowledge (in the Discussion) that the models here may not exactly reflect what might happen “in the real world” for large conjugative plasmids with e.g., addiction systems

Response: Thank you for your constructive comments. We acknowledge that “Low-copy-number plasmids” are typically large, have a low copy number and are frequently conjugative, which likely includes pICU-3/pICU-18. We have incorporated this concept that typical large conjugative plasmids usually exist at very low copy numbers into the revised sentence in the Introduction, which now reads: “A significant number of pathogenic bacteria, such as *K. pneumoniae*, disseminate resistance through the transfer of multicopy plasmids, although typical large conjugative plasmids usually exist at very low copy numbers.”.

We have also revised the Methods section describing construction of plasmids for experimental models, clarifying that we used the relatively low-copy-number vector pACYCDuet-1

(approximately 13 copies). Furthermore, we acknowledge that this technical problem represents one of the limitations of this study. We have added this point to the Discussion section, noting that: “It is challenging to select a vector plasmid with a sufficiently low copy number; therefore, the constructed model may not precisely reflect what might occur in the real world for some large conjugative plasmids, particularly those with extremely low copy numbers.”.

ORIGINAL COMMENT: “duplication state” needs rewording.

NEW COMMENT: Now Line 99 – I still think that “a duplication state” is an odd and unclear description for plasmids. If “as multiple copies” is meant, this should be stated, but see comment above.

Response: Referring to the previous response, we have revised this sentence to: “A significant number of pathogenic bacteria, such as *K. pneumoniae*, disseminate resistance through the transfer of multicopy plasmids, although typical large conjugative plasmids usually exist at very low copy numbers.”.

2) Lines 65-71

NEW COMMENT: Lines 65-71 - I think that beta-lactams would still be considered the most important class of antibiotics, and use of beta-lactamase inhibitors is not new (clavulanic acid etc.), although there are new types such as avibactam, so I would suggest some rephrasing here. Also consider whether refs 4, 5, 6, and 7 are all really necessary here.

Response: We also agree with your point. This background description was somewhat lengthy, so we have streamlined it. We combined the two sentences and focused on highlighting CZA as an alternative therapeutic approach. The previous refs 6 and 7 have also been removed. The revised sentence is as follows: However, the widespread prevalence of β -lactamase- or carbapenemase-producing strains in recent decades has driven development of alternative therapeutic approaches, notably β -lactam/ β -lactamase inhibitor combinations such as ceftazidime/avibactam (CZA), approved by the FDA in 2015.

3) blaCTX-M-9 group variants, mutations and phenotypes, Lines 109-115.

ORIGINAL COMMENT: blaCTX-M-65 and variants, Lines 100-2,129-136

Lines 100-2 - the information about blaCTX-M-65 could be expanded, explaining that blaCTX-M-65 belongs to the blaCTX-M-9 group but does not have a mutation resulting in Asp240Gly usually associated with CAZ resistance (as found in e.g., in blaCTX-M-27).

RESPONSE: ...“blaCTX-M-65 belongs to the blaCTX-M-9 group, but unlike typical members of this group (such as blaCTX-M-27), it does not carry the Asp240Gly substitution that enhances resistance to CAZ. Instead, blaCTX-M-65 exhibits strong hydrolytic activity against CTX, rather than CAZ”

NEW COMMENT: Now Lines 100-2,129-136.

The ancestral CTX-M-9 group members (CTX-M-14, CTX-M-9) do not have the Asp240Gly mutation (nor do CTX-M-1 or blaCTX-M-3, equivalent ancestors in the CTX-M-1 group). The “CTX” in CTX-M was used to indicate resistance to CTX, with mutant versions that confer CAZ resistance arising later, so variants with the Asp240Gly mutation aren't really “typical”. Also note that CTX-M-65 is a variant of CTX-M-14.

Lines 109-10 - “originally” can be deleted. From the title Refs 26 and 27 cited here don't seem to be specifically about CTX-M-65?

Response: Thank you for pointing out the issue in our description of the CTX-M group background. We have reorganized this section and added information about CTX-M-9 group members. The revised sentence now reads as follows: “*bla*_{CTX-M-65} belongs to the *bla*_{CTX-M-9} group and represents a variant derived from the ancestral member *bla*_{CTX-M-14}. Similar to CTX-M-14, CTX-M-65 exhibits strong hydrolytic activity against cefotaxime (CTX). Other members of the CTX-M-9 group, such as CTX-M-27, possess an Asp240Gly substitution, which confers enhanced resistance to ceftazidime (CAZ) compared with CTX.” We have also revised the phrase “which is originally a widely prevalent ESBL” to “a widely prevalent ESBL gene in recent years” for greater accuracy. Refs 26 and 27 have been replaced with new refs: *Comp Immunol Microbiol Infect Dis*, 2022; 86:101815 (ref. 24); *J Antimicrob Chemother*, 2013; 68(1):46-50 (ref. 25).

ORIGINAL COMMENT: Lines 171-7 – are there other studies reporting on the effects of these mutations?

RESPONSE: The dual mutations at positions 130 and 167 in blaCTX-M-65 has not yet been reported. However, the effects of single mutation at each of these sites on CTX-M enzymatic hydroactivity have been documented in several studies. For example, in *Antimicrobial Agents and Chemotherapy* (2003; 47(9):2958–2961), the Ser130Gly mutation was shown to reduce the hydrolytic activity of CTX-M enzymes against cefotaxime while increasing resistance to inhibitors including sulbactam, clavulanic acid, and tazobactam. In *PLoS Pathogens* (2010; 6(1):e1000735), the Pro167Ser mutation was reported to decrease the hydrolytic activity of group 1 CTX-M enzymes against cefotaxime. By contrast, in *Antimicrobial Agents and Chemotherapy* (2006; 50(2):731–738), the Pro167Ser mutation was shown to enhance the hydrolytic activity of group 9 CTX-M enzymes against ceftazidime.

NEW COMMENT: Now Lines 100-2, 129-136. This information should be included in the manuscript and the relevant papers cited, to make it clear that these are not novel mutations and their effects have been described previously, even if they were first found in a new combination here. Ref 14 describing CTX-M-14 also discusses the effects of the P167S mutation.

Response: We fully agree that the individual Ser130Gly and Pro167Ser substitutions are not novel, as both have occurred in multiple CTX-M β -lactamase subtypes and their resistance phenotypes have been previously characterized. The Ser130Gly substitution in CTX-M-9 confers high-level resistance to β -lactamase inhibitors while reducing cefotaxime hydrolysis (*Antimicrob Agents Chemother*, 2003; 47:2958-2961, ref. 33). The Pro167Ser substitution has been repeatedly identified in naturally occurring CTX-M enzymes, including CTX-M-19 (*Antimicrob Agents Chemother*, 2004; 48:1454-1460, ref. 34), CTX-M-62 (*Antimicrob Agents Chemother*, 2010; 54:3039-3042, ref. 35), and CTX-M-219 (*J Antimicrob Chemother*, 2017; 72:2483-2488, ref. 12), and significantly enhances ceftazidime hydrolysis with minimal impact on cefotaxime activity. However, no previous reports have described the occurrence of either

or both of these substitutions in comparison with CTX-M-65. We have now added discussion of these two mutations to the revised manuscript.

4) Lines 125-163 - example of suggested reorganisation/simplification

Sections could be combined to reduce repetition and condense. Maybe separate section on enzyme kinetics?

Response: This is a useful suggestion. We have merged the first and second subsections of the Results section into one, with the new title “Clinically isolated CZA-resistant *K. pneumoniae* mediated by a CTX-M variant”. In addition, we have condensed the combined section to make the language more concise.

Regarding the resistance profiles and enzyme kinetics of the four *bla*_{CTX-M} variants, we still have kept them together, as the enzymatic kinetic assays were performed to confirm the phenotypic resistance levels of these variants. Presenting them in a single section allows easier comparison for readers. However, we have also streamlined this part to improve clarity and brevity.

Line 139 - this wording doesn't make sense – the gene that you are attempting to clone is not *bla*_{CTX-M-65}, but the mutant from ICU-3. Maybe something like “We attempted to clone *bla*_{CTX-M-249} from ICU-3 into pCR2.1, but this resulted in colonies with different *bla*_{CTX-M} genes. Most had *bla*_{CTX-M-65} or *bla*_{CTX-M-249}, but two other variants, *bla*_{CTX-M-247} and *bla*_{CTX-M-248} were also present in a few.

Response: Thank you for pointing out that our previous wording was inaccurate. We have revised the sentence as follows: “We then cloned the *bla*_{CTX-M-65} mutant from ICU-3 into pCR2.1, but this resulted in colonies exhibiting heterogeneous genotypes.”.

Lines 159-163 – it is the clones encoding the CTX-M enzyme that are susceptible/resistant.

Response: Thank you for your professional and accurate correction, which helped us avoid a conceptual mistake. We have revised these sentences to emphasize that it is the clone, rather

than the enzyme, that exhibits susceptibility or resistance. The revised sentences are as follows:
“The clone expressing CTX-M-65 β -lactamase exhibited high-level resistance to CTX (>256 mg/L) and low-level resistance to CAZ (16 mg/L), but was susceptible to CZA (0.25 mg/L). In contrast, the CTX-M-249-producing clone was resistant to both CAZ (64 mg/L) and CZA (64 mg/L), but highly susceptible to CTX (0.5 mg/L). The CTX-M-247-producing clone was susceptible to both cephalosporins (CAZ: 1 mg/L, CTX: 0.125 mg/L) and to CZA (0.25 mg/L), whereas the CTX-M-248-producing clone exhibited resistance to CAZ (256 mg/L) and CTX (8 mg/L) but remained susceptible to CZA (8 mg/L).”.

5) Lines 133-6 blaKPC-157

PMID: 40643750IF: 2.8 Q3 (published July 2025) describes KPC-157 having the same phenotype as KPC-2. Referring to this paper would allow the text here to be simplified and would back up the result obtained here. Suggest something like “By testing the resistance conferred by blaKPC-57 cloned into pCR2.1 we found that this change did not cause an increase in CAZ resistance, as recently demonstrated (Ref)” (the carbapenem resistance is not really relevant to this manuscript).

Response: Thank you for your useful suggestion. We have replaced the previous description with the sentence you recommended and added the corresponding references (ref. 28).

6) Plasmid sequence differences, Lines 176-204

Line 182 - My alignment of the sequences of pICU-3 and pICU-18 provided/in GenBank shows that the two changes in addition to the ones in blaKPC and the nearby blaCTX-M gene are in adjacent bases in one copy of IS26. These changes should be carefully checked by mapping Nanopore reads, and if correct, properly described. See also PMID: 34015086 IF: 3.6 Q2 . Some rewording/reorganising might also help to make things clearer. For example, maybe first say that long read sequencing showed that ICU-18 carries blaKPC-2 and two copies of blaCTX-M-65, one near blaKPC-2 and one about 54 kb away, on a single plasmid named pICU-18. Then

say that ICU-3 has an almost identical plasmid, named pICU-3, with *bla*_{KPC-157}, one of the four *bla*_{CTX-M-65} variants close by and *bla*_{CTX-M-65} at the other location,

Note that “Backbone” in reference to a plasmid is usually taken to mean the region encoding plasmid functions, into which regions containing resistance genes are inserted.

Response: Thank you very much. We double-checked the alignment of the sequences of pICU-3 and pICU-18. You are correct that the two base differences are located within the IS26 element, and this variation results in a single amino acid substitution. However, since this ORF lacks 33 amino acids and represents a truncated IS26, the one–amino acid difference is unlikely to cause any functional change in this IS element. Accordingly, we have revised the annotation files and the legend of the plasmid map in Fig. 2.

We attempted to reorganize the description of the plasmid comparison section as suggested: “Long-read sequencing revealed that the CZA-susceptible isolate ICU-18 carries *bla*_{KPC-2} and two copies of *bla*_{CTX-M-65}, one near *bla*_{KPC-2} and one about 54 kb away, on a single plasmid named pICU-18. The CZA-resistant isolate ICU-3 has an almost identical plasmid, pICU-3, containing *bla*_{KPC-157}, one of the four *bla*_{CTX-M} variants near it, and *bla*_{CTX-M} at the other location. The two plasmids differed by only five SNPs, including two within *bla*_{CTX-M}, one within *bla*_{KPC}, and two within a truncated IS26 element. The specific alleles at each locus were confirmed by PCR amplification of the flanking regions followed by cloning.”

However, this phrasing does not clearly and directly describe the two *bla*_{CTX-M} copies located on pICU-3, which is the main focus we wish to emphasize. Therefore, considering our intent to highlight the features of pICU-3, we have revised this paragraph as follows: “In the clinical *K. pneumoniae* isolate ICU-3, long-read sequencing revealed that *bla*_{KPC} and two copies of *bla*_{CTX-M} are co-localized on the same plasmid, pICU-3. Notably, the two copies of *bla*_{CTX-M} are positioned at two loci approximately 54 kb apart. One locus consistently carries *bla*_{CTX-M-65}, whereas the other locus, located near *bla*_{KPC-157}, can harbor any of four *bla*_{CTX-M} variants, with *bla*_{CTX-M-249} being the most frequently recovered. Plasmid pICU-18 shares an identical backbone

with pICU-3, differing only by five SNPs, including two within *bla*_{CTX-M}, one within *bla*_{KPC}, and two within a truncated IS26 element.”.

7) Line 187 – plasmid copy number

ORIGINAL COMMENT: Lines 188-9 – are there any clues in the sequence about what is responsible for this increase in plasmid copy number? This should be looked at.

RESPONSE: ... the molecular mechanism underlying the increase in plasmid copy number remains unclear. We performed long-read sequencing and assembly under CZA pressure and compared the results with drug-free conditions, focusing on the plasmid replicon and replication-associated regulatory genes, but detected no genetic changes that could account for the elevated plasmid copy number.

NEW COMMENT: Now Lines 187 - would this analysis have identified any heterogeneity in these regions of the plasmids? If not, maybe it would be worth looking at this? I also think it would be worth mentioning in the manuscript itself that there appear to be no changes in the plasmid that would be expected to increase copy number, if this is the case.

Response: We thank the reviewer for this valuable suggestion. In response, we applied a more rigorous analytical approach. We used breseq 0.34.1 (a variant detection tool based on raw reads; *BMC Genomics*. 2014;15(1):1039), using the no-drug group as the reference, to perform whole-genome analysis and variant searching on the raw reads from the CZA-exposed group. However, we still did not identify any mutations that could explain the increase in plasmid copy number. We have explicitly included this limitation in the revised Discussion section as follows: “Second, we did not identify the molecular mechanism underlying the increase in plasmid copy number under CZA pressure. Although we carefully compared the complete genome sequences before and after drug exposure, no related mutations were found.”.

8) ORIGINAL COMMENT: Line 212 ... I also would suggest looking at whether this TU has been seen before and at papers on the plasmid contexts of *bla*_{CTX-M-65} (e.g. PMID: 22984205IF: 3.6 Q2). RESPONSE: reports specifically involving *bla*_{CTX-M-65} TUs remain

limited. In our previous work, we identified an IS26-mediated tandem multicopy structure of blaCTX-M-65 in an E. coli isolate (J Glob Antimicrob Resist, 2023; 35:202–209), though its structure differed from that observed in the present study. In addition, another study from China (J Antimicrob Chemother, 2013; 68(1):46–50) also reported a similar IS26-mediated tandem duplication unit of blaCTX-M-65 in E. coli.

NEW COMMENT: Please consider referring to the JAC paper.

Response: We thank the reviewer for this helpful suggestion. We have carefully compared the blaCTX-M-65 translocatable unit (TU) identified in our study with the multidrug resistance region (MRR) of plasmid pHN7A8 (GenBank: JN232517.1) reported in the literature (J Antimicrob Chemother, 2013; 68(1):46-50). The results show that both share an identical IS26-mediated tandem repeat module flanking the blaCTX-M-65 gene. Accordingly, we have added a citation to this JAC paper in the revised manuscript (now listed as ref. 25).

9) ORIGINAL COMMENT: Lines 218-329 Models of heterogeneity

This part of the text gets hard to follow and could be simplified and condensed.

RESPONSE: ... we revised the model names and abbreviations: Model 4 is now referred to as Dual-Plasmid Duplicates (DPD), meaning heterogeneous blaCTX-M copies are duplicated on separate plasmids; Model 5 is now referred to as Single-Plasmid Duplicates (SPD), meaning heterogeneous blaCTX-M copies are duplicated on a single plasmid. The difference between the initials “D” and “S” helps readers quickly distinguish dual-plasmid from single-plasmid.

NEW COMMENT: While I think that the revised model names are better, and agree that it is easier to tell the acronyms apart, to me “Dual plasmid duplicates” and “Single plasmid duplicates” are confusing descriptions. The important thing is that the two genes are different and where they are located. I would suggest considering these names further – maybe 1P for two different variants in the same plasmid and 2P for the two different variants on different plasmids, and perhaps 2C for two different variants in two cells?

Response: Thank you for your practical suggestions regarding the models of heterogeneity section and we have been continuously refining this part. After further consideration, we

carefully revisited the naming of the models. Indeed, the key information that needs to be conveyed by the model names is that the two genes are different and where they are located. We believe that names such as “1P”, “2P”, and “2C” are simple and effective. However, adding a leading “H” to indicate heterogeneity could make the meaning clearer. In addition, we considered the fact that in our mathematical modeling code, variable names cannot start with a number, as this would interfere with program recognition. Taking these points together, we decided to use “H1P” to represent “Heterogeneity in One Plasmid”, “H2P” for “Heterogeneity in Two Plasmids”, and “H2C” for “Heterogeneity in Two Cells”. This naming approach ensures simplicity, clear distinction, and compatibility with the computational model, while also reflecting both heterogeneity and localization. We have revised and polished all corresponding parts of the main text accordingly.

10) ORIGINAL COMMENT: Line 351-57 – which blaCTX-M group or did you look at multiple groups? Major sequence differences between members of different blaCTX-M groups (Group 1, Group 9 etc) are not due to mutations, but because the source genes are from different Kluyvera species (there are also hybrids between Group 1 and Group 9 genes). Minor changes between blaCTX-M genes within each group are probably explained by mutations that have occurred since capture.

RESPONSE: ...this part of our analysis was performed using raw genomic data to assess the heterogeneity status of a specific β -lactamase gene in each strain across a large dataset. In practice, we first identify the target gene through homology search (BLAST), and then determine heterogeneity by analyzing sequence differences among all raw reads contributing to the assembly. Therefore, whether the reference used for BLAST belongs to a Group 1 or Group 9 blaCTX-M gene does not affect our ability to first locate a specific blaCTX-M genotype within the genome. For our analysis of blaCTX-M, we used blaCTX-M-65 as the reference sequence. As long as a blaCTX-M gene is present in the genome (even one from a divergent group such as Group 1), its presence can still be identified. We then analyze whether heterogeneity exists among all reads assembled into that gene.

NEW COMMENT: I still think that giving a combined % here for all blaCTX-M genes is not ideal. It might be interesting to focus on e.g. blaCTX-1 group and 9 groups (the most common) and see whether the level of heterogeneity differs (although hybrids might interfere with this).

Response: We sincerely thank the reviewer for this valuable suggestion. After careful consideration, we fully agree with your recommendation to focus on the specific β -lactamase gene subtypes. The revised results regarding the proportion of multicopy heterogeneity are now all based on specific subtypes, including *bla*_{SHV-12}, *bla*_{TEM-1}, *bla*_{CTX-M-15/65}, *bla*_{NDM-1}, *bla*_{KPC-2} and *bla*_{OXA-48}, and the corresponding changes have been made in the main text, figures and tables. Following the reviewer's recommendation concerning *bla*_{CTX-M} subgroups, we re-analyzed the raw read data separately using *bla*_{CTX-M-15} (representing the *bla*_{CTX-M-1}-group) and *bla*_{CTX-M-65} (representing the *bla*_{CTX-M-9}-group) as reference sequences. Compared with the previous analysis, the difference this time is that we identified a total of 130 isolates exhibiting *bla*_{CTX-M} intragenic heterogeneity. To distinguish intra-group from inter-group heterogeneity, we performed assembly on these 130 isolates using Illumina raw genomic data: (1) Intra-group heterogeneity (SNPs/indels within the same group) was collapsed into a single consensus contig; (2) Inter-group heterogeneity produced two clearly distinct contigs that could not be merged. After assembling the 130 isolates with *bla*_{CTX-M} heterogeneity and performing resistance gene identification on the resulting contigs, the following pattern was observed: 42 (32.3%) showed heterogeneity exclusively within *bla*_{CTX-M-1}-group alleles, 79 (60.8%) within *bla*_{CTX-M-9}-group alleles, and 9 (6.9%) displayed two distinct contigs corresponding to simultaneous Group 1 and 9 alleles.

We have added a detailed description of *bla*_{CTX-M} heterogeneity in the revised manuscript. To avoid confusion with other β -lactamase gene subtypes, we have retained a separate subpanel in the updated Figure 7 to illustrate *bla*_{CTX-M} heterogeneity. The number of heterogeneous strains is presented as the union of results obtained from comparisons between the *bla*_{CTX-M-15} and *bla*_{CTX-M-65} groups. Table 2 also continues to present only the overall number of 130 isolates with *bla*_{CTX-M} heterogeneity, without subgroup-specific breakdown, to maintain clarity and

readability. Additional methodological details addressing this point have been incorporated into the Methods section.

11) General comments to streamline text.

“p” indicates a plasmid, so using “plasmid pXXX” or “the pXXX plasmid” is not necessary.

Cloning something in implies a vector. e.g. Lines 134-9 - “into pCR2.1 to verify” is sufficient, “The CTX-M-65 beta lactamase” etc can just be “CTX-M-65”. Also remove “the” before CTX-M-65 on Lines 168, 169 etc - not needed.

Delete things like “Results showed that” – they just add unnecessary words.

Remove unnecessary descriptions of Methods in Results

Response: Thank you very much for these practical language-related suggestions. We have carefully reviewed and revised the manuscript accordingly. For expressions such as “pICU-3 plasmid”, “plasmid p65G”, “the p65G-249R plasmid”, and “plasmid p249R”, we have removed the redundant word “plasmid” for simplicity. Similarly, “The vector plasmid pCR2.1” and “pET28a(+) vector” have been simplified to “pCR2.1” and “pET28a(+)”, respectively. We also eliminated several redundant phrases such as “Results showed that”. In addition, some method-like descriptions that appeared in the Results section have been removed or moved to the Methods section. For example, some sentences about description of the plasmid construction for experimental models have been moved from the Results to the Methods section with reorganization for better clarity.

12) Examples where wording would improve clarity/accuracy etc

Title – while I appreciate the reason for wanting to include reference to a single plasmid, I think that the title is now too general – only a single gene type has been looked at and only in a single isolate (and all experiments models were in E. coli).

Response: Thank you for your valuable suggestion regarding the title. We fully agree that choosing an appropriate title is both important and challenging, as it needs to clearly define the study’s contribution and scope while avoiding potential misunderstanding for readers.

According to the journal's requirements, the title should be sufficiently general to appeal to a broad readership. However, we also recognize that our previous title was somewhat too general. Therefore, we have reintroduced *bla*_{CTX-M} variants into the title to refocus the emphasis on the *bla*_{CTX-M} genes (although our study also includes a broader evaluation of other β -lactamases). The key finding of our study is the identification of the evolutionary advantage conferred by heterogeneous multicopy *bla*_{CTX-M} variants carried on the same plasmid in a clinically isolated *K. pneumoniae* strain, which we further modeled and validated using both experimental and mathematical approaches. Given the journal's word limit for titles (fewer than 150 characters), we were unable to include more details. Therefore, we have revised the title to: "Heterogeneous multicopy of *bla*_{CTX-M} variants on the same plasmid enhances evolutionary adaptability in clinical *Klebsiella pneumoniae*". Now we believe this version best captures the focus and significance of our study.

Line 45 – "elevated" in relation to what? This may not be the best word here.

Response: We realized that including the word "elevated" here could be misleading. Since "under CZA pressure" already conveys the intended meaning, we have removed "elevated" from the sentence.

Lines 49 – "heterogeneity duplicated within a single plasmid" is confusing.

Response: Here, we intended to convey that heterogeneity in one plasmid (corresponding to the H1P) provides greater stability and a competitive advantage than heterogeneity in two plasmids (corresponding to the H2P), particularly during drug switching. We have therefore revised the sentence to: "Using experimental and mathematical models, we demonstrated that heterogeneity in a single plasmid provides greater stability and competitive advantage than across separate plasmids, particularly during drug switching."

Lines 52-3 – “heterogeneity is widespread among various beta-lactamase genes in clinical isolates” is not clear – maybe something like “many clinical isolates have multiple variants of a beta-lactamase gene” if this is what is meant.

Response: Using the phrase “many clinical isolates have multiple variants” could be misleading.

What we intend to express is that many individual clinical isolates carry multiple copies of β -lactamase genes, and these copies may be heterogeneous. Therefore, we have revised the sentence to: “Re-analysis of large genomic datasets confirmed that many clinical isolates have multicopy heterogeneity of a β -lactamase gene.”.

Line 55- “multidrug resistance” may not be the best phrase here – this is commonly used to mean resistance to multiple classes of antibiotics, while only beta-lactams are being described here.

Response: We have revised this description as “broad β -lactam resistance”.

Lines 61-3 – “Humans have” might be better.

Response: We have revised this sentence as suggested and adjusted the verb tense throughout the entire sentence.

Line 74 – suggest “particularly those caused by”.

Response: We have revised this description as suggested.

Lines 77-9 – sentence gets hard to follow. More strains than what?

Response: This was an imprecise description, and we have removed “more” to make the expression more natural.

Line 80 – I don’t think that “Beyond KPC carbapenemases” or “occurring” are needed here and “within” could be “in”.

Response: These sentences have been reorganized as: “The *bla*_{KPC} they carry has also undergone point-mutations, mostly within the omega loop, similar to those observed in extended-spectrum β -lactamases (ESBLs) that mediate resistance to CZA. However, such omega loop mutations in ESBLs remain rare, with only a few CZA-non-susceptible cases reported, including CMY-172 and CTX-M-219, the latter being a closely related allele of the widely disseminated CTX-M-14.”.

Lines 81-3 – if CMY-172 and CTX-M-219 give CZA resistance than this needs to be clearly stated.

Response: Yes, CMY-172 and CTX-M-219 present non-susceptible to CZA, and we have revised the wording to make this clearer.

Lines 85- - wording could be improved.

Response: We revised this sentence to make it flow from the previous one and adjusted the wording. “Therefore, understanding and predicting the evolution trajectory of bacterial resistance are crucial and significant questions.”

Line 90 – this could just say “However, gene duplications can alleviate trade-offs because”.

Response: We have revised this description as suggested.

Line 116 – suggest “plasmid-borne”.

Response: The description has been revised per your suggestion.

Line 118 – “location”, not “localization”.

Response: The word “localization” has been replaced by “location”.

Lines 130-131 – this could just be “(Table 1). Both isolates belong...”

Response: We have revised this sentence as suggested.

Line 155 - as noted in previous comments, the variants do not have MICs, the antibiotics do.

Response: Thank you for pointing out our imprecise expression. We have revised the sentence according to your suggestion.

Line 168 - “For enzyme inhibition” is not needed here.

Response: We have removed “For enzyme inhibition” as suggested.

Lines 170 – “Thus” could be used instead of “These Results demonstrate” “point mutations” doesn’t need a dash, “enzyme’s” can be deleted

Response: We have revised the sentence as suggested.

Line 193 – which “drug”?

Response: Here “drug” refers to CZA, and we have clarified this in the sentence.

Line 198 – “the blaCTX-M gene near blaKPC-157”

Response: We have revised the sentence as suggested.

Lines 199-200 – the IS26, not the TU, that “bracket” the blaCTX-M gene, which is part of the TU and it is a single TU that is amplified.

Response: Thank you for pointing out this conceptual issue. Indeed, tandem amplification is mediated by the TU structure, where the resistance gene is bracketed by two IS26 elements.

The revised sentence is as follows: “This tandem amplification is probably mediated by IS26 elements bracketing the resistance gene to form a translocatable unit (TU).”.

Line 207 – “In this study” is not needed and “of plasmid-harboring blaCTX-M” does not make sense – Suggest “of plasmids carrying blaCTX-M” or “plasmid-borne blaCTX-M”.

Response: The description has been revised per your suggestion.

Lines 208-9 - Delete “To facilitate observation” and “corresponding” has no meaning in this context.

Response: We have revised this sentence and move it to the Methods section according to previous suggestion.

Line 231 – maybe “The existence of multiple variants of the same gene that confer distinct resistance profiles enables bacteria to withstand the combined effects of multiple drugs”

Response: This sentence has been revised as suggested.

Line 239 – “are encoded”

Response: We have replaced the “were encoded” with “are encoded”.

Lines 253 etc – “anchored” is not the right word here. “When both copies of blaCTX are on the same plasmid” would be simpler and more accurate, also Line 415.

Response: Thank you for your helpful suggestions. Based on your comment, we have revised the sentence accordingly. Furthermore, we have removed the term “anchored” from the Abstract, the “Inferred evolutionary trajectory”, the legend, and the final summary section.

Line 259 – change “supplemented” to “modified”

Response: Here, to distinguish between H1P and H2P in the mixed population, we added an additional fluorescent gene to the plasmid rather than altering its existing fluorescent gene. Therefore, the term “modified” is not appropriate, and we chose to retain the use of “supplemented”.

Line 270 - “surged” is not scientific.

Response: We have replaced the word “surged” with “increased sharply” to make the description more scientific.

Line 281 – “of” here does not make sense.

Response: We reorganized this sentence as: “We developed an ordinary differential equation (ODE) model to simulate plasmid population dynamics, plasmid stability, and competitive advantage.”.

Line 308 – “gained superiority” is unscientific.

Response: We have replaced the phrase “gained superiority” with “became dominant” to make the description more scientific.

Lines 314, 317, 318 – “raw genomic data”

Response: We have revised “genomic raw data” to “raw genomic data”.

Line 316 – “heterogeneity of” and “such as” is not clear – just list all gene types examined.

Response: Referring to the previous response, we have listed all examined gene types, including *bla*_{SHV-12}, *bla*_{TEM-1}, *bla*_{CTX-M-15/65}, *bla*_{NDM-1}, *bla*_{KPC-2} and *bla*_{OXA-48}.

Line 320-1 – “by mapping raw reads” if this is what is meant?

Response: We have revised the sentence as suggested.

Lines 321-2 – all this needs to say is “The estimated copy number distributions of the genes examined” – it has already been made clear that they are bla genes and from *K. pneumoniae*.

Response: We have revised the sentence as suggested.

Lines 324- 5 - *bla*_{SHV} is expected to be in the chromosome of all *K. pneumoniae* and it may or may not additionally be on plasmids – this needs to be properly explained.

Response: We have reorganized this sentence as: “Notably, as *bla*_{SHV} is chromosomally encoded in almost all *K. pneumoniae* and may additionally occur on plasmids, the proportion of isolates showing multicopy heterogeneity of *bla*_{SHV-12} reached 20.15%.”.

Line 325, 331 – sentences shouldn’t start with “And”

Response: We have removed the wording “And” in these sentences.

Line 488-90 – which flow cells/chemistry was used. The assembly is not combined, the reads are combined for the assembly?

Response: After confirming the sequencing methods, we found that Illumina sequencing was performed using the NovaSeq 6000 S4 Reagent Kit v1.5, and Nanopore sequencing was conducted using the MinION Flow Cell R9.4.1. These details have been added or revised in the Methods section. We used the term “combined” to indicate that Unicycler assembly integrates both short and long reads—first assembling large contigs with Illumina reads and then bridging them with Nanopore long reads to obtain complete genomes. We have revised the description of the assembly method accordingly.

Lines 494-5 – needs rewording.

Response: We revised this sentence as: “The number of Single Nucleotide Polymorphisms (SNPs) between *K. pneumoniae* isolates ICU-3 and ICU-18 was identified and assessed with Snippy v4.4.5 and snp-dists v0.6.3.”.

Line 507 – “Colonies were selected”, “Positive” is not needed.

Response: This sentence has been revised as suggested.

Line 510 - “...sites, cloned..”

Response: The description has been revised per your suggestion.

Line 528 – what is meant by “the low-affinity substrate”?

Response: For substrates with low-affinity (such as certain antibiotics), enzyme kinetics cannot be represented by K_m/K_{cat} values. Therefore, a direct competition method was used, with the parameter K_i employed to characterize the enzyme kinetic properties of these substrates.

Line 557 - a *psiB* gene is found on many plasmids, it is not specific to pICU-3 and there seem to be two genes annotated as *psiB* (*psiB1* and *psiB2*) in the pICU-3 sequence in PX405700, but only ~57% identical). The *psiB* primers and probe in Table S3 match *psiB*, but this needs to be explained more accurately.

Response: We thank the reviewer for pointing out the annotation and specificity issues regarding *psiB*. We have thoroughly re-examined the sequences and annotations and performed experimental validation as follows: (1) We reanalyzed all plasmids in the ICU-3 strain and confirmed that *psiB* is present only on pICU-3; no other plasmids carry this gene. (2) In the sequence of pICU-3 (PX405700), two alleles, *psiB_1* and *psiB_2*, were identified, sharing ~57% nucleotide similarity. The reference gene selected for our digital qPCR assay was *psiB_1*. Sequence alignment showed that the primers and probe used in the digital qPCR were a perfect match to *psiB_1* (100% identity), but poorly matched *psiB_2*. Specifically, the forward primer (*psiB_FW*, 18 bp) contained three mismatches, and the reverse primer (*psiB_RV*, 20 bp) contained eight mismatches with *psiB_2*, including mismatches in the critical 3' regions. The probe (*psiB-P*, 22 bp) also exhibited ten mismatches with *psiB_2*. Previous studies have demonstrated that mismatches in fluorescent probes drastically reduce binding efficiency. The figure below illustrates the alignment between the primers/probe and the two *psiB* alleles. (3) Using the same primers as in the digital qPCR assay, we performed PCR amplification and Sanger sequencing of the ICU-3 genome. The resulting chromatograms confirmed that the primers specifically amplified *psiB_1*, as shown in the figure below. (4) Table S3 and the

corresponding section in the manuscript have been updated. The primers and probe are now renamed psiB_1-FW, psiB_1-RV, and psiB_1-P, respectively.

Forward Primer(18 bp)

Reverse Primer(20 bp)

Probe(22 bp)

Sanger Sequencing Read, Forward Primer(*psiB_1-F*):

Sanger Sequencing Read, Reverse Primer(*psiB_1-R*):

Lines 580-581 – “12 dilutions covering”, “16 dilutions covering”?

Response: The description has been revised per your suggestion.

Line 589 - “grown” not “growth”.

Response: We have changed “growth” to “grown” as suggested.

Line 602 – it's the plasmids that have the fitness costs, not the strains?

Response: Yes, the correct expression should be “the plasmids imposed fitness costs on the strains”. To clarify, we conducted an assessment of the strains’ fitness and have revised the sentence accordingly.

Lines 704-5 – suggest “The pICU-3 and pICU-18 sequences are available under GenBank accession numbers PX405700 and pX405 701, respectively”.

Response: We have revised the sentence as suggested.

13) Discussion

The Discussion is still long, and some parts need rewording to improve accuracy. Also, I think that it is important not to generalise results beyond at least bla genes, maybe not beyond blaCTX-M. See comments on Fig. 7 below.

Response: We sincerely appreciate all the valuable suggestions you have provided. While addressing the following questions, we have continuously streamlined and simplified the Discussion section, including removing sentences that resembled those in the Introduction and relocating some that were more appropriate for the Results section. We fully agree that the study’s findings should not be generalized beyond β -lactamase genes. However, we also believe that the significance of multicopy heterogeneity is not limited to *bla*_{CTX-M}, and we hope that our conclusions will encourage clinical researchers to pay attention to this potential resistance feature that may occur in broader β -lactamases.

Lines 337-45 – this information still seems like it repeats and/or should be in the Introduction.

Response: We have removed some redundant sentences, particularly those already introduced in the Introduction section (e.g., “In hospital-acquired infections, *K. pneumoniae* strains producing ESBLs and carbapenemases are highly prevalent, leading to widespread resistance to commonly used cephalosporins and carbapenems.”, “To address this issue, CZA has been introduced worldwide and shows activity against CRKP strains producing KPC enzymes.”,

“Currently, most reports on CZA resistance mechanisms attribute it to MBLs, point-mutations or high expression of *bla_{KPC}*, with few studies of resistance caused by ESBLs mutations. Existing reports include the CMY-172 and a CTX-M-219.”). In addition, we reorganized several sentences and merged the first and second paragraphs of the Discussion section.

Line 350 – what is meant by “ESBL mutations” here? In which gene(s)? CMY-172 and CTX-M-219 are already mentioned in the Introduction.

Response: This sentence has been removed because it has already been described in the Introduction section.

Line 351 – needs rewording - a clonal strain is not a variant, either “trade-offs” or “a trade-off phenomenon” - see previous comments. AST is done on isolates/strains, enzyme kinetics uses proteins, neither is done with genes.

Response: We revised this sentence as: “As a closely related allele of *bla_{CTX-M-65}*, the phenotypic changes conferred by *bla_{CTX-M-249}* demonstrated trade-offs, since the carrying strain gained CZA resistance while losing resistance to CTX.”.

Line 354 – change “explored” – see previous comments.

Response: The use of the word “explored” has been revised, and the sentence has been merged and condensed with the following one.

Line 357, 360, 421 - genes are not encoded on plasmids, they are e.g. carried by plasmids, found on plasmids – see previous comments.

Response: The phrase “gene encoded on plasmids” has been corrected to “carried” in multiple locations throughout the main text.

Line 358 - “In the subsequent study” does not make sense here.

Response: The phrase “In the subsequent study” has been removed and replaced with the single word “Then”.

Line 362 – the “locus encoding CTX-M-249” is the blaCTX-M-249 gene - this term can be used. A gene can't undergo an “increase in plasmid copy number”.

Response: This sentence contained expression errors. We have reorganized and condensed it as follows: “Furthermore, we found that under drug pressure, the plasmid can undergo an increase in copy number or transient tandem multicopy events mediated by IS26 flanking *bla*_{CTX-M-249}, which could revert once the drug pressure is removed.”.

Line 367 - it is really the blaCTX-M-65 gene that is widespread.

Response: Here, our primary aim was to convey that CTX-M-65 is one of the most widely reported ESBLs. Accordingly, we have revised the original statement to reflect this.

Line 370 – “the novel” not needed.

Response: We have removed “the novel” as suggested.

Lines 371-32 – the enzymes hydrolyse, but it is the strains encoding them that shows resistance.

Response: We have changed the “enzymes” by “CTX-M-247-producing strain” or “CTX-M-249-producing strain”. And these sentences have been reorganized as: “While CTX-M-247-producing strain shows no resistance enhancement to CAZ, CTX and CZA, and CTX-M-248 only increases CAZ hydrolysis, CTX-M-249-producing strain, which harbors both substitutions, retains CAZ resistance and confers higher resistance to CZA.”.

Line 377 – it is multicopy of essentially a single gene (based on blaCTX-M-65) not multiple genes

Response: We have revised this description with “multiple closely related alleles of a resistance gene”.

Lines 377-8 - what “functioned as a bet hedging strategy”?

Response: We have revised this sentence as: “representing paralogous genes that execute a bet-hedging evolutionary strategy.”.

Lines 382-4 – “simultaneously carrying blaTEM-1 and blaTEM-12” and state differences in sequences and phenotypes, “trade-off losses”, not “trade-offs losses”.

Response: We have restructured this sentence, revised the description according to your suggestion, and split it into two separate sentences. The revised version is as follows: “In 2018, a study artificially constructed bacteria simultaneously carrying *bla*_{TEM-1} and *bla*_{TEM-12}, which are closely related alleles but exhibit different resistance profiles. The authors first proposed that such heterogeneity could avoid trade-offs, conferring a bet-hedging effect, which they experimentally confirmed.”.

Line 385 – change “brought by” to “due to”

Response: We have changed “brought by” to “due to” as suggested.

Line 386 – “in a clinical strain” – only one strain studied here showed this effect.

Response: We have revised this phrase as suggested.

Line 393 – no dash needed in “single antibiotic”

Response: We have reorganized this sentence as: Through the comparison of our constructed “Heterogeneity in Two Plasmids (H2P)”, we found that the “Heterogeneity in One Plasmid (H1P)” strain shows reduced fitness without drug pressure.

Line 398 – “genes split between” doesn’t really make sense here – “genes on” would be fine.

Response: We have revised this phrase as suggested.

Line 415 etc – “anchored in” is not the best way to say this. “on”.

Response: We have removed several instances of the term “anchored” throughout the manuscript.

Lines 416-7 – unclear - do you mean that the first step was for the plasmid to acquire a single copy of blaCTX-M-65?

Response: Yes, that is exactly our intended meaning. We reworded the expression to make the meaning more direct. The revised version is as follows: “We hypothesize that the acquisition of a single *bla*_{CTX-M-65} by the plasmid represents an early stage in the evolutionary trajectory in this clinical strain.”.

Lines 441-2 – the wording here could still be made clearer.

Response: We reworded the expression to make the meaning more direct and clearer. The revised version is as follows: “In contrast, multicopy heterogeneity refers to the coexistence of distinct allelic variants of a resistance gene within the same strain or plasmid, resulting in varied resistance profiles and potentially enhancing bacterial adaptation under diverse antibiotic pressures.”.

14) ORIGINAL COMMENT: Lines 471-5 – is this really true for long-read if multiple copies of the gene are in different genetic locations, as for copies of blaCTX-M-65 (or variants) both in pICU-3 here?

RESPONSE: ...when multiple copies of a gene are located in different genetic contexts—as with blaCTX-M-65 (or its variants) present in both regions of pICU-3—the relatively lower base accuracy of long-read sequencing makes it difficult to resolve which specific variant resides in which repeat region.

NEW COMMENT: I don’t really agree with this. From my own use of read mapping with ONT reads (R10), it can be pretty easy to distinguish real nt differences (occur in almost all ONT reads) from errors (each occurs in only one or two sequences).

Response: Thank you very much for the valuable comments. Our intention here was mainly to emphasize our concern that the phenomenon of multicopy heterogeneity is often overlooked. In clinical genomic epidemiology studies, most analyses are based on short-read sequencing using the Illumina platform, where heterogeneously paralogous genes are almost certainly missed. Even for the minority of isolates sequenced using long-read platforms, accurate identification of such heterogeneity from assembled genomes is still difficult without an understanding of the multicopy heterogeneity mechanism described in this study—highlighting the significance of our work. We have confirmed that the sequencing data used in this study (for strains ICU-3 and ICU-18) were generated with the Nanopore R9.4.1 flow cell, which is the similar version used for most Nanopore-based sequences available in GenBank. To verify the identity of the obtained raw reads, we randomly selected 20 *bla*_{CTX-M}-containing reads from strain ICU-3 and aligned them to *bla*_{CTX-M-65}, revealing an average identity of approximately 90%. We acknowledge that the ONT R10 has greatly improved the base-calling accuracy of Nanopore sequencing, and we hope that with the advancement of such technologies, the phenomenon of multicopy heterogeneity will receive increasing attention from clinical researchers.

15) References

ORIGINAL COMMENT: Ref 8 needs page numbers or equivalent. Something seems to be wrong with page numbers in Ref 49. Journal titles need to be correctly abbreviated. Ref 4, 12 etc., remove Title Case. Species and gene names etc. need to be correctly formatted.

RESPONSE: and carefully checked all reference entries, correcting errors and inconsistencies in formatting. Species and gene names have also been italicized accordingly.

NEW COMMENT: There are still a few problems.

Ref 4 has no page numbers or equivalent

Refs 4,11 – check if in vitro/vivo are in italics in these papers.

Refs 8, 23, 28, 55– check if beta symbols are used in these papers.

Refs 13,15, 53, 54, 55 – page number equivalents look odd.

Ref 19 – probably should be “Typhimurium”, not in italics and *rrn* in italics.

Ref 37 – “26” of IS26 should probably be in italics.

Ref 61 – “a tool”

Response: Thank you for your detailed suggestions. We have reorganized and reviewed all the references and have made all the revisions as you advised, including page numbers, italics, wording, and so on.

16) Figures

The figures and their organisation/distribution between the main manuscript and Supplementary is improved, but more consistency in colours etc across figures/parts of figures would improve clarity. For example, in the main manuscript, the bar chart at the bottom of Fig. 1 Part II uses different colours for blaCTX-M-65 and the three variants from the diagrams in Part I. What appear to be the same graphs in Fig. 1 Part I and Fig. 2 b and c use different colours again, as do Fig. 3 and Fig. 7. In Fig. 7 (a) blue is used for blaCTX-M-65 and a dark red colour is used for blaCTX-M-249, but in (b) the ancestral allele (blaCTX-M-65 in part a) is in a dark red and the evolved allele (blaCTX-M-249 in part a) is in blue. Also, in Fig. 1 Part III the colours of the blaCTX-M genes in the models are hard to see.

Maybe start with green for GFP, red for mCherry, orange for both and blue for mTagBFP, as these are fixed, then find different, easily distinguishable colours to show blaCTX-M-65 and the three variants and use these consistently in diagrams, graphs etc consistently throughout, where possible.

Response: We thank the reviewer for this very helpful suggestion and fully agree that colour consistency greatly improves clarity. Exactly as the reviewer kindly suggested, the fluorescent protein genes themselves are always coloured green for eGFP, red for mCherry, and blue for mTagBFP (Fig. 3, Fig. S2a, Fig. S4a); strains carrying only eGFP are shown in green, strains carrying only mCherry in red, dual-fluorescent (eGFP + mCherry) strains in orange, and triple-fluorescent (eGFP + mCherry + mTagBFP) strains in purple-blue (Fig. 5, Fig. 6). Independently,

the *bla*_{CTX-M-65} and the three variants are consistently represented as yellow for *bla*_{CTX-M-65}, fuchsia for *bla*_{CTX-M-247}, russet for *bla*_{CTX-M-248}, and purple for *bla*_{CTX-M-249} (Fig. 1 Part I-II, Fig. 2c, Fig. 3a, Fig. 8a, Fig. S1c, Fig. S2a, Fig. S4a). Additionally, the colours in the bar chart at the bottom of Fig. 1 Part II have been updated to match those used in Fig. 2b. To improve visibility of the gene colours in the plasmid schematics of Fig. 1 Part III, the backbones have been substantially narrowed so that the coloured arrows now clearly stand out.

Figure legends often need some rewriting to improve accuracy/calclity

Fig. 1 legend

Lines 881-2 – is ICU-18 really a “clone”?

Response: We have revised the “clone” to “strain”.

Line 883 – cloning also identified the different variants

Response: We have revised this phrase to “cloning confirmed distinct variants and their resistance phenotypes”.

Line 884 – “Plasmid location”

Response: “Plasmid localization” has been changed to “Plasmid location”.

Lines 886-7 - “different multicopy heterogeneity states”, “heterogeneity stability” can be described more clearly with less jargon.

Response: We have reorganized the language to make the description of Part III more concise and clearer. The revised sentence is as follows: “Plasmid models carrying *bla*_{CTX-M} variants were constructed to simulate and compare the advantages of multicopy heterogeneity, particularly under multiple antibiotic combination pressures, plasmid stability under drug pressure, and competitive ability during drug switching. All findings were further validated using mathematical modeling and large-scale genomic data.”.

Lines 889-90 – “multiple minor variants of various bla genes in a single isolate from SRA data”

Response: We have rephrased this sentence, simplifying the description of specific details and instead outlining the general framework of the work. The revised sentence is as follows: “All findings were further validated using mathematical modeling and large-scale genomic data.”.

Fig. 2 legend

Line 892 – “diagram” not needed”.

Response: The figure title has been simplified by removing the redundant word “diagram”.

Line 893 – pICU-18 is not shown in the diagram.

Response: pICU-18 is actually already shown in the diagram. We used pICU-3 as the reference sequence, and pICU-18 was aligned against it. In the plasmid circular map, the inner blue ring represents pICU-3, while the outer pink ring represents pICU-18. To make the presence of pICU-18 completely unambiguous, we have adjusted the diagram and revised the figure legend. The three small circular markers in the legend are now explicitly labeled as: pICU-3 (pink), pICU-18 (blue), and GC content (black).

Line 894 – “elements” not needed – they are IS. The IS26 form the TUs, and all of the tandem TUs could be shown on the diagram – see previous comment.

Response: First, we deleted the term “elements”. Regarding the issue of tandem TUs, in our study strains, the number of tandem TUs within the bacterial population actually varies among individual cells, ranging from 1 to 7 copies (as stated in the main text). We verified this by examining the third-generation raw reads, which confirmed the presence of TUs in different copy numbers (e.g., 3 or 7 TUs, etc.). We chose the case with 3 tandem TUs as an example to display in the figure, as its sequence length is more consistent and suitable for the scale of the figure. We have also clarified in the legend that the tandem TUs shown are based on a representative Nanopore-based raw read.

Line 895 – “ONT read”? Relative to what?

I think that it would be helpful in part c to make it clear that these copy numbers are due to a combination of things i.e., every plasmid would have a copy of blaCTX-M-65 (non-amplified copy flanked by inverted IS26) and the % of blaCTX-M-249 is a combination of both mutation and amplification, at least under some conditions – see previous comment relating to this.

Response: We thank the reviewer for pointing this out. To avoid any ambiguity, the phrase “raw read” in the legend of Fig. 2a has been revised to “Nanopore-based long read”. The reference for each relative copy number is already indicated on the x-axis labels of Figure 2b: “vs. chromosome” means divided by the single-copy chromosomal housekeeping gene *pgi* of *K. pneumoniae*, and “vs. pICU-3” means divided by the pICU3-borne single-copy marker *psiB_I*. We have added in the description of panel c that the changes in the proportion of each allele arise from variations in both the plasmid copy number they reside in and the number of copies within tandem TUs.

Fig. 3

p15A_ori could be removed from the diagrams at the top to simplify them (it’s in all of the plasmids) and note that pACYCDuet1 has this replicon in Methods (Lines 563-71).

Response: We have removed the “p15A” label from all plasmid schematics to reduce redundancy. The replication origin itself is retained as a simple “ori” arrow to indicate the replication start site. The type of replicon is explicitly described in the Methods section.

Legend

Lines 900-1 – title could be improved.

Response: We have revised the fig title to “Construction of heterogeneity models and selection under multidrug combinations”.

Text can be simplified and condensed – see comments above and avoid explaining things that are obvious from the figure.

e.g. Lines 901-2 “Model 1 (strain 65G) has p65G (blaCTX-M-65-GFP) only, Model 2 (strain 249R) has p249R (blaCTX-M-249-mCherry), Model 3 is a 1:1 mixture of 65G and 249R (population level heterogeneity, PLH). Model 4 has p65G and p249R in the same cell (2P) and Model 5 has p65G-249R, (blaCTX-M-65-GFP+blaCTX-M-249-mCherry) on a single plasmid (1P), using alternative model names suggested above.

Response: Thank you very much for your valuable suggestions. We have simplified these descriptions as per your recommendations.

Line 970 – “results for the five models.

Response: The comment most likely refers to line 907 in the previous version of the manuscript. Following the reviewer’s suggestion, we have rephrased the sentence as: “Checkerboard susceptibility assay results for the five models”.

Line 980 – “co-selection by”

Response: The comment most likely refers to line 908 in the previous version of the manuscript. Following the reviewer’s suggestion, the phrase “co-selection of” has been changed to “co-selection by” in the revised manuscript.

Lines 909-10 “represent”, not “represented”

Response: The verb tense has been corrected from “represented” to “represents” in the revised manuscript.

Line 911-2 – “in each cell represent the” and “values of bacterial... concentration” are not needed.

Response: We have simplified this sentence as suggested.

Fig. 4

Part (a) - suggest stating % of what on graph, rather than in legend.

Response: The label now consistently reads "XX/total cells (%)" to ensure readers immediately understand the basis of the relative percentage calculation.

Part (B) – not clear why a dotted line is used for “all” but a continuous line for CTX resistant. Might be better if all lines are dotted.

Response: We thank the reviewer for the comment. We have now changed all lines to continuous (solid) lines for consistency and clarity.

Legend

Title, Lines 914-5 – probably doesn't need “during serial passages”.

Response: The phrase “during serial passages” has been removed from the title. The revised figure title now reads: Stability of heterogeneity in the H1P and H2P models under CTX or CZA pressure.

Lines 916-7 – probably not needed - see above (but should be “is plotted”)

Response: This sentence describes the method for plotting the line graph. We have retained it for the sake of clarity, and have also changed “was plotted” to “is plotted”.

Lines 917-9, 922-3 – legends should describe what is in the figure, not repeat Results.

Response: We have removed these sentences.

Line 920 – suggest “by plating serial dilutions on CTX or CAZ and counting CFU”, Lines 920-22 are then not needed.

Response: We have revised this sentence to “by plating serial dilutions on CTX or CAZ and counting CFU at days 0, 3, and 6” and removed the last sentence.

Fig. 5

Suggest stating relative % of what on graph, rather than in legend. Maybe colours used could match the fluorescence colours? Also, 65G is used elsewhere as a strain name, but here “65G” is really the 2 plasmid strain that has lost p249R, and conversely “249R” is really the 2 plasmid strain that has lost p65G. Also “SPD” is used here, not SPD-B, as in the legend. Some simple explanatory diagrams might be useful here – showing original DPD, SPD-B and loss of one plasmid or another to give “65G” and “249R”.

Response: We appreciate the reviewer’s detailed feedback on Fig. 5, and we have implemented the following revisions: (1) The label now consistently reads “XX/total cells (%)” to ensure readers immediately understand the basis of the relative percentage calculation; (2) We have updated the color scheme for the entire Fig. 5 to match the fluorescence colours, enhancing the intuitive understanding of the data; (3) We have added a simple explanatory diagram as Fig. 5a to visually illustrate the plasmid composition of the original H2P and H1P-B strains, and how the loss of one plasmid leads to the 65G and 249R derivatives. (4) To ensure consistency and accuracy, the strain name “H1P-B” has been uniformly applied to both Fig. 5 and its corresponding legend. In addition, “65G” refers to the strain carrying only the p65G. When the H2P strain loses p249R, it converts into strain 65G, and vice versa. “249R” refers to the strain carrying only the p249R; when the H2P strain loses p65G, it becomes strain 249R.

Legend

Line 926 – suggest “(additional blue fluorescence)”.

Response: The description has been changed to “H1P-B (additional blue fluorescence)” as suggested.

Fig. 6

See comments above on blaCTX-M genes.

Response: We re-analyzed the raw read data separately using *bla*_{CTX-M-15} (representing the *bla*_{CTX-M-1}-group) and *bla*_{CTX-M-65} (representing the *bla*_{CTX-M-9}-group) as reference sequences. This re-analysis identified a total of 130 isolates exhibiting *bla*_{CTX-M} intragenic heterogeneity.

For concise presentation, the results of these 130 isolates are displayed together as “*bla*_{CTX-M-15/65}” in Fig. 7 (previously Fig. 6).

Line 934 – delete “such as”, “-like” should not be subscript.

Response: The sentence has been revised to: “Several common β -lactamase genes (*bla*_{SHV-12}, *bla*_{TEM-1}, *bla*_{NDM-1}, *bla*_{KPC-2}, *bla*_{CTX-M-15/65}, and *bla*_{OXA-48}) were evaluated for copy number and mutation heterogeneity by analyzing raw read mapping coverage in *K.pneumoniae*.”.

Line 935 – is this really mapping to a reference not “alignment”?

Response: “alignment” has been corrected to “mapping”.

Lines 937-8 – it’s not really the strains that are “(non)heterogeneous”, they have multiple variants of the gene of interest.

Response: The sentence has been revised to: “The circular charts show the proportions of strains carrying resistance genes that exhibit heterogeneous and non-heterogeneous in human versus non-human sources.”.

Fig. 7

This scheme may be generalisable to e.g. other *bla*_{CTX-M} gene families (e.g. group 1) or other *bla* genes and beta-lactam antibiotics (where one mutation can have a drastic effect on phenotype) but might not apply to other types of antibiotic resistance genes.

Response: We thank the reviewer for the valuable suggestion. In revised Fig. 8a (previously Fig.7a), this is an inferred evolutionary trajectory schematic illustrating *bla*_{CTX-M} heterogeneity states in the clinical strain. Thus, the colours for *bla*_{CTX-M} genes have been fully unified with the rest of the manuscript. Fig. 8b (previously Fig.7b) is a general conceptual model applicable to the broad β -lactamase genes. Therefore, to preserve its universality and avoid restricting its application to specific *bla*_{CTX-M} alleles, the original color scheme, which distinguishes different

conceptual components, has been intentionally retained unchanged. We have accordingly added some qualifiers in the main text and the legend.

See comments above on colours used for blaCTX-M genes. What do pink and blue parts signify?

Response: In the revised Fig. 8a (previously Fig. 7a), the colours for *bla*_{CTX-M} genes have now been fully unified across all figures. The pink and blue backgrounds in Fig. 8b (previously Fig. 7b) represent the drug pressure environment: blue background = cells under antibiotic A pressure; pink background = cells under antibiotic B pressure.

Suggest flipping the direction of the curved arrows on the two left hand diagrams, to make it clearer that these parts are “moving away from” the central diagram.

Response: In response to your suggestion, the curved arrows in the two left-hand diagrams of Fig. 8b (previously Fig. 7b) have been flipped to point away from the center, effectively illustrating the “moving away from” relationship.

Legend

Line 942 -suggest “blaCTX-M gene heterogeneity in ICU-3” if that is what is meant?

Response: The legend has been revised to: “A hypothetical evolutionary trajectory schematic illustrating *bla*_{CTX-M} heterogeneity in the clinical ICU-3.”.

Line 945 – suggest “single mutation or two mutations”

Response: The sentence has been revised as suggested.

Line 946 - suggest “...pressure transient tandem copies of the mutated gene may be generated”

Response: The sentence has been revised to: “Under antibiotic pressure, transient tandem copies of the mutated gene may be generated.” as suggested.

Line 947 – suggest “of heterogeneity in blaCTX-M genes carried on the same plasmid”

Response: Following the reviewer's suggestion to make the wording more precise while preserving general applicability, the legend has been revised to: "Schematic of five models illustrating the adaptive advantage of multicopy heterogeneity of a β -lactamase gene on the same plasmid."

Line 948-53 – I would suggest trying to simplify the text, e.g. "non-heterogeneity" = single gene variant, "plasmid-level multi-copy heterogeneity" = both gene variants on the same plasmid.

Response: To be honest, we don't believe the suggested phrasing provides a better description. In fact, heterogeneity is a key concept in our study — we aim to highlight multicopy heterogeneity more prominently to draw greater attention from clinical researchers.

17) Supplementary

Table S1 - column heading says "inhabitants", should be "inhibitors".

Response: We have corrected "inhabitants" to the proper spelling, "inhibitors", in the revised manuscript.

Table S2, 3rd column heading use "allele(s)"?

Response: We have revised the heading of the 3rd column in Table S2 to use "Allele(s)".

Table S3 – "probes, not "Probes" in heading, suggest making it clear where CTX-M primers are only relevant for blaCTX-M group 9 genes.

Response: "Probes" was corrected to "probes" in the table heading. Furthermore, clear statement has been added in the "Purpose" column for all *bla*_{CTX-M} primers/probes.

Table S4 – shorten and reword titles – suggest "BioProjects containing Illumina sequence reads for *K. pneumoniae* isolated from humans" etc.

Response: Following the reviewer's recommendation, we have revised the titles of the two parts of Supplementary Table 4 (now Supplementary Table 5) to make them immediately clear: Supplementary Table 5a BioProjects containing Illumina sequence reads for *K. pneumoniae* isolated from humans; Supplementary Table 5b BioProjects containing Illumina sequence reads for *K. pneumoniae* isolated from non-human.

Supplementary figures – check for accuracy and condense if possible - see comments on legends for main figures.

Response: All supplementary figures and their legends have been carefully checked for accuracy and consistency with the main figures.